# GENERATOR MATCHING: GENERATIVE MODELING WITH ARBITRARY MARKOV PROCESSES

**Peter Holderrieth**[1,†], **Marton Havasi**[2], **Jason Yim**[1], **Neta Shaul**[2,3], **Itai Gat**[2],
**Tommi Jaakkola**[1], **Brian Karrer**[2], **Ricky T. Q. Chen**[2], **Yaron Lipman**[2]
[1]MIT CSAIL, [2]FAIR, Meta, [3]Weizmann Institute of Science
[†]Work done during an internship at FAIR, Meta.

## ABSTRACT

We introduce *Generator Matching*, a modality-agnostic framework for generative modeling using arbitrary Markov processes. Generators characterize the infinitesimal evolution of a Markov process, which we leverage for generative modeling in a similar vein to flow matching: we construct conditional generators which generate single data points, then learn to approximate the marginal generator which generates the full data distribution. We show that Generator Matching unifies various generative modeling methods, including diffusion models, flow matching and discrete diffusion models. Furthermore, it expands the design space to new and unexplored Markov processes such as jump processes. Finally, Generator Matching enables the construction of superpositions of Markov generative models and enables the construction of multimodal models in a rigorous manner. We empirically validate our method on image and multimodal generation, e.g. showing that superposition with a jump process improves performance.

## 1 INTRODUCTION

Early deep generative models—like VAEs (Kingma, 2013) and GANs (Goodfellow et al., 2014) generated samples in a single forward pass. With denoising diffusion models (DDMs) (Song et al., 2020; Ho et al., 2020), a paradigm shift happened were *step-wise* updates are used to transform noise into data. Similarly, scalable training of continuous normalizing flows (CNFs; Chen et al. 2018) via flow matching (Lipman et al., 2022; Liu et al., 2022; Albergo et al., 2023) allowed for high-quality and fast generative modeling by simulating an ODE. Since then, similar constructions based on diffusion and flows have also been applied to other modalities such as discrete data (Campbell et al., 2022; Gat et al., 2024) or data on manifolds (De Bortoli et al., 2022; Huang et al., 2022; Chen & Lipman, 2024) leading to a variety of models for different data types.

The single common property of the aforementioned generative models is their iterative step-wise nature: starting with a sample $X_0 \sim p_{\text{simple}}$ from an easy-to-sample distribution $p_{\text{simple}}$, they iteratively construct samples $X_{t+h}$ of the next time step depending only on the current state $X_t$. Mathematically speaking, this means that they are all *Markov processes*. In this work, we develop a generative modeling framework that relies on that Markov property. At the core of our framework is the concept of a *generator* that describes the infinitesimal change of the distribution of a Markov process. We show that one can easily learn a *generator* through a family of scalable training objectives—a framework we coin *Generator Matching (GM)*.

Generator Matching unifies many existing generative modeling techniques across data modalities such as denoising diffusion models (Song et al., 2020), flow matching (Lipman et al., 2022), stochastic interpolants (Albergo et al., 2023), discrete diffusion models (Campbell et al., 2022; Gat et al., 2024; Lou et al., 2024a), among many others (see sec. 8). Most importantly, GM gives rise to new, unexplored models, and allows us to combine models across different classes of Markov processes. We make the following contributions:

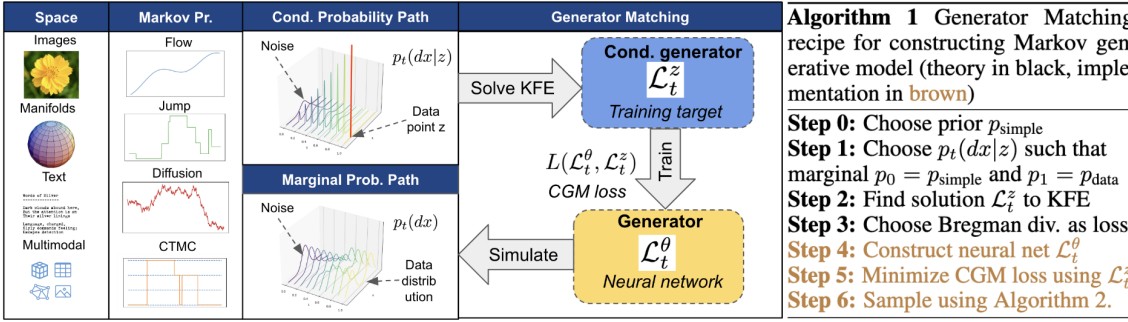

Figure 1: Overview of the Generator Matching (GM) framework to construct generative models. GM works on any state space (including multi-modal) and Markov processes. *Flower image source:* `vecteezy.com`

1. **Generator Matching:** We present *Generator Matching* (GM), a framework for generative modeling with Markov processes on arbitrary state spaces. This framework unifies a diversity of prior generative modeling methods into a common framework that is modality-agnostic.

2. **Novel models:** On discrete and Euclidean spaces, we universally characterize the space of Markovian generative models identifying jump models as an unexplored model class for $\mathbb{R}^d$.

3. **Model combinations:** We show how GM allows to combine models in 2 ways: (1) *Markov superpositions* enable to construct ensembles of generative models; and (2) *Multimodal generative models* can be constructed by combining GM models built for single data modalities.

4. **Experiments:** On image and multimodal protein generation experiments, we show that jump models and Markov superpositions allow us to achieve competitive results.

## 2 GENERATIVE MODELING VIA PROBABILITY PATHS

Let $S$ be a **state space**. Important examples are $S = \mathbb{R}^d$ (*e.g.*, images, vectors), $S$ discrete (*e.g.*, language), $S$ a Riemannian manifold (*e.g.*, geometric data) or their products for multimodal generation. In generative modeling, we are given samples $x_1, \ldots, x_N \sim p_{\text{data}}$ from a distribution $p_{\text{data}}$ on $S$ and our goal is to generate novel samples $z \sim p_{\text{data}}$. GM works for arbitrary distributions, in particular those that do not have densities (*e.g.*, with discrete support). For general probability measures $p$, we use the notation $p(dx)$ where "$dx$" is a *symbolic* expression denoting integration with respect to $p$ in a variable $x$. If for a distribution $p$ a density exists with respect to a reference measure $\nu$ on $S$, we write $\frac{dp}{d\nu}(x)$ for its density.

A fundamental paradigm of recent state-of-the-art generative models is that they prespecify a transformation of a simple distribution $p_{\text{simple}}$ (e.g. a Gaussian) into $p_{\text{data}}$ via probability paths. Specifically, a **conditional probability path** is a set of time-varying probability distributions $(p_t(dx|z))_{0 \leq t \leq 1}$ depending on a data point $z \in S$. Together with the data distribution $p_{\text{data}}$, this induces a corresponding **marginal probability path** via the hierarchical sampling procedure:

$$z \sim p_{\text{data}}, x \sim p_t(dx|z) \quad \Rightarrow \quad x \sim p_t(dx) \tag{1}$$

i.e. first sample a data point $z \sim p_{\text{data}}$ and then sample $x \sim p_t(dx|z)$ from the conditional path. As we will see, this makes training scalable. The conditional probability path is usually chosen such that $p_0(dx|z) = p_{\text{simple}}$ and $p_1(dx|z) = \delta_z$ where $\delta_z$ is the Dirac delta distribution at $z$ (i.e. the trivial, deterministic distribution returning $z$ every draw). The associated marginal probability path interpolates between $p_{\text{simple}}$ and $p_{\text{data}}$, leading to the first design principle of GM:

**Principle 1**: Given a data distribution $p_{\text{data}}$, choose a prior $p_{\text{simple}}$ and a conditional probability path such that its marginal probability path $(p_t)_{0 \leq t \leq 1}$ fulfills $p_{\text{simple}} = p_0$ and $p_{\text{data}} = p_1$.

Two common constructions are mixtures (for arbitrary $S$) and geometric averages (for $S = \mathbb{R}^d$):

$$p_t(dx|z) = (1 - \kappa_t) \cdot p_{\text{simple}}(dx) + \kappa_t \cdot \delta_z(dx) \quad \Leftrightarrow \quad x_t \sim \begin{cases} z & \text{with prob } \kappa_t \\ x_0 & \text{with prob } (1 - \kappa_t) \end{cases} \quad \blacktriangleright \text{ mixture} \qquad (2)$$

$$p_t(dx|z) = \mathbb{E}_{x_0}[\delta_{\sigma_t x_0 + \alpha_t z}] \qquad\qquad\qquad \Leftrightarrow \quad x_t = \sigma_t x_0 + \alpha_t z \qquad \blacktriangleright \text{ geometric average} \qquad (3)$$

where $x_t \sim p_t(\cdot|z), x_0 \sim p_{\text{simple}}, z \sim p_{\text{data}}$, and $\alpha_t, \sigma_t, \kappa_t \in \mathbb{R}_{\geq 0}$ are differentiable functions satisfying $\kappa_0 = \alpha_0 = \sigma_1 = 0$ and $\kappa_1 = \alpha_1 = \sigma_0 = 1$ and $0 \leq \kappa_t \leq 1$.

**Remark.** Note that the above includes specific ways of constructing probability paths, e.g. via a forward process in diffusion models (Song et al., 2020) or an interpolant function (Albergo et al., 2023) (see app. A.4). GM works for all these cases including if one conditions on more general latent variables $z$ (Tong et al., 2023; Pooladian et al., 2023). Note also that in the diffusion literature, time is inverted ($t = 0$ corresponds to data).

## 3 MARKOV PROCESSES

We briefly define time-continuous Markov processes, a fundamental concept in this work (Ethier & Kurtz, 2009). For $t \in [0, 1]$, let $X_t \in S$ be a random variable. We call $(X_t)_{0 \leq t \leq 1}$ a *Markov process* if it fulfills the following condition for all $0 \leq t_1 < t_2 < \cdots < t_n < t_{n+1} \leq 1$ and $A \subseteq S$ (measurable):

$$\mathbb{P}[X_{t_{n+1}} \in A | X_{t_1}, X_{t_2}, \ldots, X_{t_n}] = \mathbb{P}[X_{t_{n+1}} \in A | X_{t_n}] \qquad \blacktriangleright \text{ Markov assumption} \qquad (4)$$

Informally, the above condition says that the process has no memory. If we know the present, knowing the past will not influence our prediction of the future. In table 1, we give an overview over important classes of Markov processes. Each Markov process has a **transition kernel** $(k_{t+h|t})_{0 \leq t < t+h \leq 1}$ that assigns every $x \in S$ a probability distribution $k_{t+h|t}(\cdot|x)$ such that $\mathbb{P}[X_{t+h} \in A | X_t = x] = k_{t+h|t}(A|x)$. Due to the Markov assumption, there is a 1:1 correspondence between a Markov process and a transition kernel paired with an initial distribution $p_0$. We impose loose regularity assumptions on $X_t$ listed in app. A.2.

In the context of GM, we use a Markov process as follows: Given a marginal path $(p_t(dx))_{0 \leq t \leq 1}$ (see sec. 2), we want to train a model that allows to simulate a Markov process such that $X_0 \sim p_{\text{simple}} \Rightarrow X_t \sim p_t$ for all $0 \leq t \leq 1$. That is, if **initializing state at** $t = 0$ **with** $X_0 \sim p_0$**, the marginals of** $X_t$ **will be** $p_t$ for all $0 \leq t \leq 1$. Once we have found such a Markov process, we can simply generate samples from $p_1 = p_{\text{data}}$ by sampling $X_0 \sim p_0$ and simulating $X_{t+h} \sim k_{t+h|t}(\cdot|X_t)$ step-wise up to time $t = 1$. The challenge with such an approach is that an arbitrary general kernel $k_{t+h|t}$ is hard to parameterize in a neural network. One of the key insights in the development of diffusion models was that for small $h > 0$, the kernel $k_{t+h|t}$ can be closely approximated by a simple parametric distribution like Gaussians (Sohl-Dickstein et al., 2015; Ho et al., 2020). One can extend this idea to Markov processes leading to the concept of the *generator*.

## 4 GENERATORS

Let us consider the transition kernel $k_{t+h|t}$ for small $h > 0$. Specifically, we consider an *informal* 1st-order Taylor approximation in $t$ with an error term $o(h)$:

$$\text{"}k_{t+h|t} = k_{t|t} + h\mathcal{L}_t + o(h)\text{"}, \quad \mathcal{L}_t := \frac{d}{dh}\Big|_{h=0} k_{t+h|t}, \quad k_{t|t}(\cdot|x) = \delta_x \qquad (5)$$

We call the 1st-order derivative $\mathcal{L}_t$ the *generator* of $k_{t+h|t}$ (Ethier & Kurtz, 2009; Rüschendorf et al., 2016). Similar to derivatives, generators are first-order *linear* approximations and, as we will see, easier to parameterize than $k_{t+h|t}$. Diffusion, flow, and other generative models can all be seen as algorithms to learn the generator of a Markov process (see table 1). However, as a probability measure is not a standard function, equation 5 is not well-defined yet. We will make it rigorous using *test functions*.

**Test functions.** Test functions are a way to "probe" a probability distribution. They serve as a theoretical tool to handle distributions as if they were real-valued functions. Specifically, we use a family $\mathcal{T}$ of bounded,

| Name | Flow | Diffusion | Jump process | Continous-time Markov chain |
|---|---|---|---|---|
| Space $S$ | $S = \mathbb{R}^d$ | $S = \mathbb{R}^d$ | $S$ arbitrary | $|S| < \infty$ |
| Parameters | $u_t(x) \in \mathbb{R}^d$ | $\sigma_t^2(x) \in \mathbb{R}^{d \times d}$ 
 $\sigma_t^2$ pos. semi-def. | Jump measure $Q_t(dy; x)$ | $Q_t \in \mathbb{R}^{S \times S}, 1^T Q_t = 0$ 
 $Q_t(x'; x) \geq 0 \, (x' \neq x)$ |
| Sampling | $X_{t+h} = X_t + h u_t(X_t)$ | $X_{t+h} = X_t + \sqrt{h\sigma_t^2(X_t)}\epsilon_t$ 
 $\epsilon_t \sim \mathcal{N}(0, I)$ | $X_{t+h} = X_t$ with prob. $1 - h\int Q_t(dy; x)$ 
 $X_{t+h} \sim \frac{Q_t(dy;x)}{\int Q_t(dy;x)}$ with prob. $h\int Q_t(dy;x)$ | $X_{t+h} \sim (I + hQ_t)(\cdot; X_t)$ |
| Generator $\mathcal{L}_t$ | $\nabla f^T u_t$ | $\frac{1}{2}\nabla^2 f \cdot \sigma_t^2$ | $\int [f(y) - f(x)]Q_t(dy; x)$ | $f^T Q_t^T$ |
| KFE (Adjoint) | Continuity Equation: 
 $\partial_t p_t = -\nabla \cdot [u_t p_t]$ | Fokker-Planck Equation: 
 $\partial_t p_t = \frac{1}{2}\nabla^2 \cdot [p_t \sigma_t^2]$ | Jump Continuity Equation: $\partial_t \frac{dp_t}{d\nu}(x) =$ 
 $\int Q_t(x; x')\frac{dp_t}{d\nu}(x') - Q_t(x'; x)\frac{dp_t}{d\nu}(x)v(dx')$ | Mass preservation: 
 $\partial_t p_t = Q_t p_t$ |
| Marginal | $\mathbb{E}_{z \sim p_{1|t}(\cdot|x)}[u_t(x|z)]$ | $\mathbb{E}_{z \sim p_{1|t}(\cdot|x)}[\sigma_t^2(x|z)]$ | $\mathbb{E}_{z \sim p_{1|t}(\cdot|x)}[Q_t(dx'; x|z)]$ | $\mathbb{E}_{z \sim p_{1|t}(\cdot|x)}[Q_t(x'; x|z)]$ |
| CGM Loss (Example) | $\|u_t(x|z) - u_t^\theta(x)\|^2$ | $\|\sigma_t^2(x|z) - [\sigma_t^\theta]^2(x)\|_2^2$ | $(\int Q_t^\theta(x'; x)v(dx')$ 
 $-Q_t(x'; x|z)\log Q_t^\theta(x'; x)v(dx'))$ | $(\sum_{x' \neq x} Q_t^\theta(x'; x)$ 
 $-Q_t(x'; x|z)\log Q_t^\theta(x'; x))$ |

Table 1: Examples of Markov models that can be learnt with GM. Derivations are in app. A.5. For diffusion, we assume zero drift. KFE is listed in its adjoint version, i.e. assumes jump kernel $Q_t(y; x)$ and density $\frac{dp_t}{d\nu}(x)$ exists with respect to reference $\nu$. For Lebesgue measure $\nu$, we write $\frac{dp_t}{d\nu}(x) = p_t(x)$.

integrable functions $f : S \to \mathbb{R}$ that *characterize* probability distributions fully, *i.e.*, two probability distributions $\mu_1, \mu_2$ are equal if and only if $\mathbb{E}_{x \sim \mu_1}[f(x)] = \mathbb{E}_{x \sim \mu_2}[f(x)]$ for all $f \in \mathcal{T}$. Generally speaking, one chooses $\mathcal{T}$ to be as "nice" (or regular) as possible. For example, if $S = \mathbb{R}^d$, the space $\mathcal{T} = C_c^\infty$ of infinitely differentiable functions with compact support fulfills that property. We define the action of the marginal $p_t$ and transition kernels $k_{t+h|t}$ for all $f \in \mathcal{T}$ via the linear function defined via

$$\langle p_t, f \rangle \stackrel{\text{def}}{=} \int f(x)p_t(dx) = \mathbb{E}_{x \sim p_t}[f(x)] \qquad \blacktriangleright \text{ marginal action} \qquad (6)$$

$$\langle k_{t+h|t}, f \rangle(x) \stackrel{\text{def}}{=} \langle k_{t+h|t}(\cdot|x), f \rangle = \mathbb{E}[f(X_{t+h})|X_t = x] \qquad \blacktriangleright \text{ transition action} \qquad (7)$$

where the marginal action maps each test function $f$ to a scalar $\langle p_t, f \rangle \in \mathbb{R}$, while the transition action maps a real-valued function $x \mapsto f(x)$ to a another real-valued function $x \mapsto \langle k_{t+h|t}, f \rangle(x)$. The tower property implies that $\langle p_t, \langle k_{t+h|t}, f \rangle \rangle = \langle p_{t+h}, f \rangle$. We note that the above is only a "symbolic" dot product but becomes a "proper" dot product if a density $\frac{dp_t}{d\nu}$ exists, i.e. $\langle p_t, f \rangle = \int f(x)\frac{dp_t}{d\nu}(x)\nu(dx)$.

**Generator definition.** Let us revisit equation 5 and define the derivative of $k_{t+h|t}$. With the test function perspective in mind, we can take derivatives of $\langle k_{t+h|t}, f \rangle(x)$ per $x \in S$ and define

$$\frac{d}{dh}\bigg|_{h=0} \langle k_{t+h|t}, f \rangle(x) = \lim_{h \to 0} \frac{\langle k_{t+h|t}, f \rangle(x) - f(x)}{h} \stackrel{\text{def}}{=} [\mathcal{L}_t f](x). \qquad (8)$$

We call this action the *generator* $\mathcal{L}_t$ (and define it for all $f$ for which the limit exists uniformly in $x$ and $t$, see app. A.1 ). In table 1, there are several examples of generators listed with derivations in app. A.5. With this definition, the Taylor series in equation 5 has the, now well-defined, form as $\langle k_{t+h|t}, f \rangle = f + h\mathcal{L}_t f + o(h)$.

Under mild regularity assumptions, there is a unique correspondence between the generator and the Markov process (see (Ethier & Kurtz, 2009; Pazy, 2012)). This allows us to parameterize a Markov process:

**Principle 2**: Parameterize a Markov process via a parameterized generator $\mathcal{L}_t^\theta$.

Of course, it is hard to parameterize a linear operator $\mathcal{L}_t$ on function spaces directly via a neural network. A simple solution is to restrict ourselves to certain subclasses of Markov processes and parameterize it linearly with a neural network (see app. A.6 for details and examples). For example, flow matching restricts itself to generators of the form $\mathcal{L}_t f = \nabla f(x)^T u_t^\theta(x)$ which correspond to flows. However, as we will show now, we can in fact fully characterize generators on specific spaces.

**Theorem 1** (Universal characterization of generators). *Under regularity assumptions (see app. A.2), the generators of a Markov processes $X_t$ ($0 \leq t \leq 1$) take the form:*

1. ***Discrete*** $|S| < \infty$***:*** *The generator is given by a rate transition matrix $Q_t$ and the Markov process corresponds to a continuous-time Markov chain (CTMC).*

2. ***Euclidean space*** $S = \mathbb{R}^d$***:*** *The generator has a representation as a sum of components described in table 1, i.e.,*

$$\mathcal{L}_t f(x) = \underbrace{\nabla f(x)^T u_t(x)}_{\text{flow}} + \underbrace{\frac{1}{2} \nabla^2 f(x) \cdot \sigma_t^2(x)}_{\text{diffusion}} + \underbrace{\int [f(y) - f(x)] \, Q_t(dy; x)}_{\text{jump}} \tag{9}$$

*where $u : [0,1] \times \mathbb{R}^d \to \mathbb{R}^d$ is a **velocity field**, $\sigma : [0,1] \times \mathbb{R}^d \to S_d^{++}$ the **diffusion coefficient** ($S_d^{++}$=positive semi-definite matrices), and $Q_t(A|x)$ is a finite measure called **jump measure**. $\nabla^2 f(x)$ describes the Hessian of $f$ and $\nabla^2 f(x) \cdot \sigma_t^2(x)$ describes the Frobenius inner product.*

The proof adapts a known result in the mathematical literature and can be found in app. C.1. This result allows us to not only characterize a wide class of Markov process models but to characterize the design space *exhaustively* for $S = \mathbb{R}^d$ or $S$ discrete. In Euclidean space, people have considered learning the flow parts of the generator and for diffusion models, using a fixed $\sigma_t$ for a diffusion. Learning $\sigma_t$ or jump models on non-discrete spaces have not (or rarely) been considered. A general recipe to sample from a Markov process with a universal generator is presented in alg. 2. For $S = \mathbb{R}^d$, we can therefore simplify Principle 2:

**Principle 2** ($S = \mathbb{R}^d$): Parameterize a Markov process (*e.g.*, using a neural network) via a generator $\mathcal{L}_t$ that is composed of (a subset of) velocity $u_t$, diffusion coefficient $\sigma_t^2$, and jump measure $Q_t$.

## 5 KOLMOGOROV FORWARD EQUATION AND MARGINAL GENERATOR

Beyond parameterizing a Markov process, the generator has a further use-case in the Generator Matching framework: checking if a Markov process generates a desired probability path $p_t$. We discuss the latter now using the **Kolmogorov Forward Equation (KFE)**. Specifically, the evolution of the marginal probabilities $p_t$ of a Markov process $X_t$ are governed by the generator $\mathcal{L}_t$, as can be seen by computing:

$$\partial_t \langle p_t, f \rangle = \frac{d}{dh}\Big|_{h=0} \langle p_{t+h}, f \rangle = \left\langle p_t, \frac{d}{dh}\Big|_{h=0} \langle k_{t+h|t}, f \rangle \right\rangle \stackrel{(8)}{=} \langle p_t, \mathcal{L}_t f \rangle \tag{10}$$

where we used that the $\langle p_t, \cdot \rangle$ operation is linear to swap the derivative, and the fact that $\langle p_t, \langle k_{t+h|t}, f \rangle \rangle = \langle p_{t+h}, f \rangle$. This shows that given a generator $\mathcal{L}_t$ of a Markov process $X_t$ we can recover its marginal probabilities via their infinitesimal change,

$$\partial_t \langle p_t, f \rangle = \langle p_t, \mathcal{L}_t f \rangle \qquad \blacktriangleright \text{ Kolmogorov Forward Equation (KFE)} \tag{11}$$

Conversely, if a generator $\mathcal{L}_t$ of a Markov process $X_t$ satisfies the above equation, then $X_t$ generates the probability path $(p_t)_{0 \leq t \leq 1}$, i.e. initializing $X_0 \sim p_0$ will imply that $X_t \sim p_t$ for all $0 \leq t \leq 1$ (see app. A.2) (Rogers & Williams, 2000). Therefore, the key challenge of Generator Matching is:

**Principle 3\***: Given a marginal probability path $(p_t)_{0 \leq t \leq 1}$, find a generator satisfying the KFE.

**Remark - Adjoint KFE.** We note that the above version of the KFE determines the evolution of expectations of test functions $f$. Whenever a probability density $\frac{dp_t}{d\nu}(x)$ exists, one can use the *adjoint KFE* (see table 1 for examples and app. A.3). In this form, the KFE generalizes many equations used to develop generative models such as the Fokker-Planck or the continuity equation (Song et al., 2020; Lipman et al., 2022).

We now show how to find a generator that generates a marginal probability path $p_t$ with conditional path $p_t(\cdot|z)$. Assume that for every data point $z \in S$, we found a generator $\mathcal{L}_t^z$ that generates $p_t(\cdot|z)$. We call $\mathcal{L}_t^z$ **conditional generator**. This allows us to construct a generator for the marginal path (**marginal generator**):

**Proposition 1.** *The marginal probability path* $(p_t)_{0 \leq t \leq 1}$ *is generated by a Markov process* $X_t$ *with generator*

$$\mathcal{L}_t f(x) = \mathbb{E}_{z \sim p_{1|t}(\cdot|x)}[\mathcal{L}_t^z f(x)] \tag{12}$$

*where* $p_{1|t}(dz|x)$ *is the posterior distribution (i.e. the conditional distribution over data* $z$ *given an observation* $x$). *For* $S = \mathbb{R}^d$ *and the representation in eq.* (9), *we get a marginal representation of* $\mathcal{L}_t f(x)$ *given by:*

$$\nabla f(x)^T \mathbb{E}_{z \sim p_{1|t}(\cdot|x)}[u_t(x|z)] + \frac{\nabla^2 f(x)}{2} \cdot \mathbb{E}_{z \sim p_{1|t}(\cdot|x)}[\sigma_t^2(x|z)] + \int [f(y) - f(x)] \mathbb{E}_{z \sim p_{1|t}(\cdot|x)}[Q_t(dy; x|z)]$$

*Generally, an identity as in eq.* (12) *holds for any linear parameterization of the generator (see app. A.6).*

The proof relies on the linearity of the KFE (see app. C.2). Proposition 1 immensely simplifies the construction of a Markov process that generates a desired probability path. To find the right training target, we only need to find a solution for the KFE for the conditional path. This simplifies Principle 3* to:

> **Principle 3**: Derive a *conditional* generator $\mathcal{L}_t^z$ satisfying the KFE for the conditional path $p_t(\cdot|z)$.

In denoising diffusion models, the strategy to find solutions to a KFE is to construct a probability path via a forward noising process and then use a time-reversal of that process as a solution to the KFE (we illustrate this in app. H.2). Here, we illustrate two novel solutions for the KFE for common conditional probability paths on $\mathbb{R}^d$ in fig. 2. We discuss them here for $d = 1$ (in sec. 7.2 it is discussed how to easily extend it to $d > 1$).

**Example 1 - Jump solution to geometric average.** Current state-of-the-art models use a geometric average probability path given by $p_t(\cdot|z) = \mathcal{N}(tz, (1-t)^2)$ called CondOT path (Lipman et al., 2022). We ask the question: are there other Markov processes that follow the same probability path? As derived in app. E.7, another solution is given by a jump model with rate kernel $Q_t : \mathbb{R} \times \mathbb{R} \to \mathbb{R}$:

$$Q_t(x'; x|z) = \frac{[k_t(x)]_+[-k_t(x')]_+ p_t(x'|z)}{(1-t)^3 \int [-k_t(\tilde{x})]_+ p_t(\tilde{x}|z)d\tilde{x}}, \quad k_t(x) = x^2 - (t+1)xz - (1-t)^2 + tz^2 \tag{13}$$

where $[x]_+ := \max(x, 0)$. In fig. 2, we illustrate how a jump model trained with this conditional rate has the same *marginal probability* path as common flow models but with significantly different *sample* paths.

**Example 2 - Pure diffusion solution to mixture path.** GM allows to learn the diffusion coefficient $\sigma_t^2$ of an SDE. We illustrate this for the mixture path $p_t(dx|z) = \kappa_t \delta_z + (1 - \kappa_t)\mathrm{Unif}_{[a_1, a_2]}$. We introduce a solution that we call **"pure diffusion"** (see app. E.3). The corresponding Markov process is given by an SDE with no drift (*i.e.*, no vector field) and diffusion coefficient given by

$$\sigma_t^2(x|z) = 2\dot{\kappa}_t \frac{a_2 - a_1}{1 - \kappa_t} \left( \frac{1}{2} \frac{(z - a_1)^2}{a_2 - a_1} + [x - z]_+ - \frac{1}{2} \frac{(x - a_1)^2}{a_2 - a_1} \right) \tag{14}$$

We add an additional reflection term at the boundaries of the data support (see app. E.3). Note the striking feature of this model: It only specifies how much noise to add to the current state. Still, it is able to generate data (see fig. 2). This is strictly different than "denoising diffusion models" because they *corrupt* data (as opposed to *generating*) with a diffusion process and their $\sigma_t^2(x) = \sigma_t^2$ is state-independent and usually fixed.

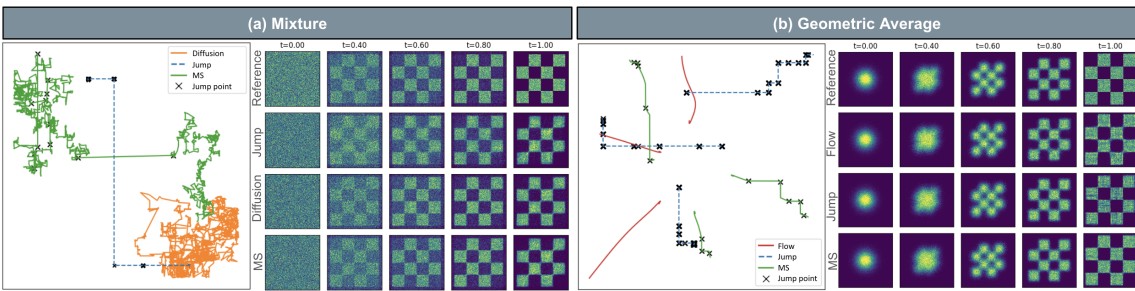

Figure 2: Illustration of Markov models trained with different KFE solutions for the same probability path. The paths for individual samples are plotted *across* time in one plot. 2d histograms of generated samples are plotted per time point. Although the individual sample paths look very different, the marginal probability path (histogram) are the same up to approximation error (Geometric average ~ example 1, mixture ~ example 2).

## 6   GENERATOR MATCHING

We now discuss how to train a parameterized generator $\mathcal{L}_t^\theta$ to approximate the "true" marginal generator $\mathcal{L}_t$. In practice, $\mathcal{L}_t^\theta$ is linearly parameterized by a neural network $F_t^\theta : S \times [0,1] \to \Omega$ where $\Omega \subset V$ is a convex subset of some vector space $V$ with inner product $\langle \cdot, \cdot \rangle$ (see app. A.6 for details). Our goal is to approximate the ground truth parameterization $F_t : S \times [0,1] \to \Omega$ of $\mathcal{L}_t$. For example, $F_t = u_t$ for flows, $F_t = \sigma_t^2$ for diffusion, or $F_t = Q_t$ for jumps (see table 1). We train the neural network $F_t^\theta$ to approximate $F_t$. As a distance function on $\Omega$, we consider **Bregman divergences** defined via a convex function $\phi : \Omega \to \mathbb{R}$ as

$$D(a,b) = \phi(a) - [\phi(b) + \langle a - b, \nabla\phi(b) \rangle], \quad a, b \in \Omega \tag{15}$$

which are a general class of loss functions including many examples such as MSE or the KL-divergence (see app. C.3.1). We use $D$ to measure how well $F_t^\theta$ approximates $F_t$ via the **Generator Matching loss** defined as

$$L_{\text{gm}}(\theta) \overset{\text{def}}{=} \mathbb{E}_{t \sim \text{Unif}, x \sim p_t} \left[ D(F_t(x), F_t^\theta(x)) \right] \qquad \blacktriangleright \text{ Generator Matching} \tag{16}$$

Unfortunately, the above training objective is intractable as we do not know the marginal generator $\mathcal{L}_t$ and also no parameterization $F_t$ of the marginal generator. To make training tractable, let us set $F_t^z$ to be a linear parameterization of the conditional generator $\mathcal{L}_t^z$ with data point $z$ (see app. A.6). For clarity, we reiterate that by construction, we know $F_t^\theta, F_t^z, p_t(\cdot|z), D$ as well as can draw data samples $z \sim p_{\text{data}}$ but the shape of $F_t$ is unknown. By proposition 1, we can assume that $F_t$ has the shape $F_t(x) = \int F_t^z(x) p_{1|t}(dz|x)$. This enables us to define the **conditional Generator Matching loss** as

$$L_{\text{cgm}}(\theta) \overset{\text{def}}{=} \mathbb{E}_{t \sim \text{Unif}, z \sim p_{\text{data}}, x \sim p_t(\cdot|z)} \left[ D(F_t^z(x), F_t^\theta(x)) \right] \qquad \blacktriangleright \text{ Conditional Generator Matching} \tag{17}$$

This objective is tractable and scalable. It turns out that we can use it to minimize the desired objective.

**Proposition 2.** *For any Bregman divergence, the GM loss $L_{gm}$ has the same gradients as the CGM loss $L_{cgm}$, i.e. $\nabla_\theta L_{gm}(\theta) = \nabla_\theta L_{cgm}(\theta)$. Therefore, minimizing the CGM loss with Stochastic Gradient Descent will also minimize the GM loss. Further, for this property to hold, $D$ must necessarily be a Bregman divergence.*

See app. C.3 for a proof. Note the significance of proposition 2: we can learn $\mathcal{L}_t$ with a scalable objective. Further, we can **universally characterize the space of loss functions**, including unexplored and new loss functions for diffusion models. In table 1, we list examples of several CGM loss functions. Often it is also possible to derive losses that give lower bounds on the model log-likelihood (ELBO bounds, see app. D).

> **Principle 4**: Train $\mathcal{L}_t^\theta$ by minimizing the CGM loss with a Bregman divergence.

With this, we arrived at the last principle of GM. In alg. 1, we summarize the Generator Matching recipe for constructing generative models.

## 7 APPLICATIONS OF GENERATIVE MATCHING THEORY

GM provides a unifying framework for many existing generative models (see sec. 8), as well as gives rise to new models. Beyond that, the generality of GM in itself has several use cases that we discuss in this section.

### 7.1 COMBINING MODELS

The generator is a linear operator and the KFE $\partial_t \langle p_t, f \rangle = \langle p_t, \mathcal{L}_t f \rangle$ is a linear equation. These two properties enable us to combine generative models for the same state space $S$ in different ways.

**Proposition 3** (Combining models). *Let $p_t$ be a marginal probability path, then the following generators solve the KFE for $p_t$ and consequently define a generative model with $p_t$ as marginal:*

1. ***Markov superposition:*** *$\alpha_t^1 \mathcal{L}_t + \alpha_t^2 \mathcal{L}_t'$, where $\mathcal{L}_t, \mathcal{L}_t'$ are two generators of Markov processes solving the KFE for $p_t$, and $\alpha_t^1, \alpha_t^2 \geq 0$ satisfy $\alpha_t^1 + \alpha_t^2 = 1$. We call this a **Markov superposition**.*

2. ***Divergence-free components:*** *$\mathcal{L}_t + \beta_t \mathcal{L}_t^{div}$, where $\mathcal{L}_t^{div}$ is a generator such that $\langle p_t, \mathcal{L}_t^{div} f \rangle = 0$ for all $f \in \mathcal{T}$, and $\beta_t \geq 0$. We call such $\mathcal{L}_t^{div}$ **divergence-free**.*

3. ***Predictor-corrector:*** *$\alpha_t^1 \mathcal{L}_t + \alpha_t^2 \bar{\mathcal{L}}_t$, where $\mathcal{L}_t$ is a generator solving the KFE for $p_t$ in forward-time and $\bar{\mathcal{L}}_t$ is a generator solving the KFE in backward time, and $\alpha_t^1, \alpha_t^2 \geq 0$ with $\alpha_t^1 - \alpha_t^2 = 1$.*

A proof can be found in app. C.4. Markov superpositions can be used to combine generative models of different classes, *e.g.*, one could combine a flow and a jump model. These can be 2 networks trained separately or we can train two models in one network simultaneously. We illustrate Markov superpositions in fig. 2. To find divergence-free components, one can use existing Markov-Chain Monte-Carlo (MCMC) algorithms - such as Hamiltonian Monte Carlo, Langevin dynamics, or approaches based on detailed balance - all of these algorithms are general recipes to find divergence-free components.

### 7.2 MULTIMODAL AND HIGH-DIMENSIONAL GENERATIVE MODELING

GM allows us to easily combine generative models from two state spaces $S_1, S_2$ into the product space $S_1 \times S_2$ in a rigorous, principled, and simple manner. This has two advantages: (1) we can design a joint multi-modal generative model easily and (2) we can often reduce solving the KFE in high dimensions to the one-dimensional case. We state here the construction informally and provide a rigorous treatment in app. C.5.

**Proposition 4** (Multimodal generative models - Informal version). *Let $q_t^1(\cdot|z_1), q_t^2(\cdot|z_2)$ be two conditional probability paths on state spaces $S_1, S_2$. Define the conditional factorized path on $S_1 \times S_2$ as $p_t(\cdot|z_1, z_2) = q_t^1(\cdot|z_1) q_t^2(\cdot|z_2)$. Let $p_t(dx)$ be its marginal path.*

1. ***Conditional generator:*** *To find a solution to the KFE for the conditional factorized path, we only have to find solutions to the KFE for each $S_1, S_2$. We can combine them component-wise.*

2. ***Marginal generator:*** *The marginal generator of $p_t(dx)$ can be parameterized as follows: (1) parameterize a generator on each $S_i$ but make it values depend on all dimensions; (2) During sampling, update each component independently as one would do for each $S_i$ in the unimodal case.*

3. ***Loss function:*** *We can simply take the sum of loss functions for each $S_i$.*

As a concrete example, let us consider joint image-text generation with a joint flow and discrete Markov model with $S_1 = \mathbb{R}^d, S_2 = \{1, \dots, N\}$. To build a multimodal model, we can simply make the vector field $u_t(x_t^1, x_t^2) \in \mathbb{R}^d$ depend on both modalities $x_t^1, x_t^2$ but update the flow part via $X_{t+h}^1 = X_t^1 + h u_t(X_t^1, X_t^2)$. Similarly, the text updates depend on both $(X_t^1, X_t^2)$. In app. F, we give another example for jump models.

## 8 RELATED WORK

GM unifies a diversity of previous generative modeling approaches. We discuss here a selection for $S = \mathbb{R}^d$ and $S$ discrete. App. H includes an extended overview and models for other $S$ (e.g. manifolds, multimodal).

**Denoising Diffusion and Flows in $\mathbb{R}^d$.** From the perspective of GM, a "denoising diffusion model" is a flow model that is learnt using the CGM loss with the mean squared error. During sampling, a divergence-free component given via Langevin dynamics (Flow + SDE) can be be added for stochastic sampling (see proposition 3). If we set the weight of that component to 0, we recover the probability flow ODE (Song et al., 2020). To the best of our knowledge, it has not been explored in the literature yet whether one could *learn* a state-dependent diffusion coefficient $\sigma_t(x)$ as opposed to fixing it. Our framework allows for that as we illustrate in fig. 2. Flow matching and rectified flows (Lipman et al., 2022; Liu et al., 2022) are immediate instances of Generator Matching leveraging the flow-specific versions of the KFE given by the continuity equation (see table 1). Stochastic interpolants (Albergo et al., 2023) extend general flow-based models by learning an additional divergence-free Langevin dynamics component separately (see proposition 3 (b)) and showcase the advantages of adding it both theoretically and practically.

**Discrete models and LLMs.** In discrete spaces, Generator Matching recovers generative modeling via continuous-time Markov chains (Campbell et al., 2022; Santos et al., 2023; Gat et al., 2024), often coined "discrete diffusion models". These models use a version of proposition 4 using factorized probability paths to make the generator (=rate matrix $Q_t$) update each dimension independently. SEDD (Lou et al., 2024b) use the same Bregman divergence but with a different linear parameterization of the generator, namely via the discrete score. Theoretically, by using auto-regressive probability paths and fixing the jump times via high jump intensities, one can also recover common language model training as an edge case of GM.

**Markov generative modeling.** The most closely-related work to ours is Benton et al. (2024) that focus on recovering existing denoising diffusion models into a common framework. Here, we try to fully characterize the design space of Markov generative models as a whole and identify novel parts - *e.g.*, by introducing jump models on $\mathbb{R}^d$, Markov superpositions, universal characterizations of the space of generators, novel solutions to the KFE, Bregman divergence losses as the natural loss classes, among others.

## 9 EXPERIMENTS

The design space of the GM framework is extraordinarily large. At the same time, single classes of models (*e.g.*, diffusion and flows) have already been optimized over many previous works. Therefore, we choose to focus on 3 aspects of GM: (i) Jump models as a novel class of models (ii) The ability of combining different model classes into a single generative model (iii) The ability to design models for multiple data modalities.

**New models - Jump model.** We first study jump models as a novel model class in Euclidean space. We use the jump model defined in eq. (13) and extend it to multiple dimensions using proposition 4. The jump kernel is parameterized with a U-Net architecture (see app. F for details). We use the loss from table 1. In app. D, we show that this corresponds to an ELBO loss. We apply the model on CIFAR10 and the ImageNet32 (blurred faces) datasets. As jump models do not have yet an equivalent of classifier-free guidance, we focus on unconditional generation. A challenge for a fair comparison is that flow models can use higher-order ODE samplers, while sampling for jump models in $\mathbb{R}^d$ is only done with Euler sampling so far.

| Method | CIFAR10 | ImageNet |
|---|---|---|
| DDPM (Ho et al., 2020) | 3.17 | 6.99 |
| VP-SDE (Song et al., 2020) | 3.01 | 6.84 |
| EDM (Karras et al., 2022) | 1.98 | – |
| Flow model (Euler) | 2.94 | 4.58 |
| Jump model (Euler) | 4.23 | 7.66 |
| Jump + Flow MS (Euler) | **2.49** | **3.47** |
| Flow model (2nd order) | 2.48 | 3.59 |
| Jump + Flow MS (mixed) | **2.36** | **3.33** |

Table 2: Experimental results for image generation. FID scores are listed. MS=Markov superposition. Euler: euler sampling. 2nd order: 2nd order ODE sampler. Mixed: Flow uses 2nd order sampler and jump uses Euler sampling.

Hence, we ablate over this choice. As one can see in fig. 4, the jump model can generate realistic images of high quality. In table 2, we show quantitative results. While lacking behind current state-of-the art models, the jump model shows very promising results as a first version of an unexplored class of models.

**Combining models - Markov superposition.** Next, we train a flow and jump model in the same architecture. We validate that the flow part achieves the state-of-the-art results as before. We then combine both models via a Markov superposition. As one can see in table 2, a Markov superposition of a flow and jump model boosts the performance of each other. For Euler sampling, we see significant improvements. We can also combine 2nd order samplers for flows with Euler sampling for jumps in a "mixed" sampling method (see table 2) leading to improvements of the current SOTA by flow models. We anticipate that with further improvements of the jump model, the increased performance via Markov superposition will be even more pronounced.

**Multimodal state spaces - Protein experiments.** GM allows us to design models for arbitrary and complex state spaces. To illustrate this, we show that GM allows to easily improve *Multi-Flow*, a state-of-the-art model for joint protein structure and amino acid sequence generation (Campbell et al., 2024b), without even re-training the model. Specifically, we derive a novel solution $\mathcal{L}_t^z$ to the KFE on $S = SO(3)$ with a jump model (app. G.1). We then use proposition 4 to make it multi-dimensional and combine it with a

| Method | Multimodal | | Unimodal | |
| --- | --- | --- | --- | --- |
| | Div. | Nov. | Div. | Nov. |
| RFdiffusion (Watson et al., 2023) | N/A | | 0.4 | 0.37 |
| FrameFlow (Yim et al., 2024) | N/A | | 0.39 | 0.39 |
| FoldFlow (Bose et al., 2023) | N/A | | 0.24 | 0.32 |
| Protpardelle (Chu et al., 2024) | 0.1 | 0.4 | 0.12 | **0.41** |
| ProteinGenerator (Lisanza et al., 2023) | 0.09 | 0.31 | 0.19 | 0.35 |
| MultiFlow (Campbell et al., 2024b) | 0.38 | 0.39 | 0.52 | 0.39 |
| w/ $SO(3)$ jumps (ours) | **0.48** | **0.41** | **0.63** | **0.41** |
| w/ $SO(3)$ jumps + flow (ours) | 0.47 | 0.4 | 0.59 | 0.40 |

Table 3: Protein generation results. Diversity (Div) is the share of *unique* proteins passing a quality check called designability, Novelty (Nov) is the average inverse similarity of each protein passing designability.

flow model on $\mathbb{R}^d$ and a discrete Markov model on $\{1, \ldots, n\}^d$ for $n = 20$ (# amino acids), i.e. the state space becomes $S = \mathbb{R}^d \times SO(3)^d \times \{1, \ldots, 20\}^d$. Using the pre-trained MultiFlow without any fine-tuning, we "pseudo-marginalize" the conditional jumps by predicting $x_1 \in SE(3)$ and then taking a conditional step with $\mathcal{L}_t^z$. In fig. 3, we can see examples of generated proteins. We benchmark our results following Yim et al. (2023a). Table 3 shows our results of incorporating jumps with MultiFlow compared to baselines. In multimodal setting, both sequence and structure state spaces are sampled jointly while in the unimodal setting only the structure is sampled. We see that **including a jump model results in state-of-the-art performance while greatly increases the diversity metric**. See App. G.4 for more results and experiment details.

**Additional small-scale experiments.** We include a systematic study of the design space of GM on toy data in app. E. In app. I, we show that new Bregman divergences can improve existing flow and diffusion models.

## 10 DISCUSSION

We introduced Generator Matching, a general framework for scalable generative modeling on arbitrary state spaces via Markov generators. The generator abstraction offers key insights into the fundamental equations governing Markov generative models: Generators are *linear* operators, the KFE is a *linear* equation, and Bregman divergences are *linear* in the training target. Therefore, any minimization we do *conditionally* on a data point, implicitly minimizes the training target *marginalized* across a distribution of data points. These principles allowed us to unify a diversity of prior generative modeling methods such as diffusion models, flow matching, or discrete diffusion models. Further, we could universally characterize the space of Markov models and loss functions. GM allows us to combine generative models of different classes on the same state space (Markov superpositions) and to easily build multimodal generative models. Future work could further explore the design space of GM. For example, we showed how one can learn a diffusion coefficient $\sigma_t$ of a diffusion model. In addition, jump models on Euclidean space offer a large class of models that we could only study here in its simple instances. In addition, future work can explore better samplers or distillation. To conclude, Generator Matching provides both a rigorous theoretical foundation and opens up a large practical design space to advance generative modeling across a diverse range of applications.

ACKNOWLEDGMENTS

P.H. would like to thank Yann Ollivier for his insightful comments and for sharing his mathematical expertise throughout the project. Further, P.H. would like to thank Andreas Eberle for mathematical support and advice as well as for sharing excellent teaching materials for Markov process theory. Further, we would like to thank Maurice Weiler, Ishaan Chandratreya, Alexander Sauer, and Krunoslav Lehman Pavasovic for extensive feedback on early drafts of the work. We also thank Anuroop Sriram and Daniel Severo for feedback and help along the way. Finally, we would like to thank Heli Ben-Hamu for sharing code to implement Mises-Fisher distributions.

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

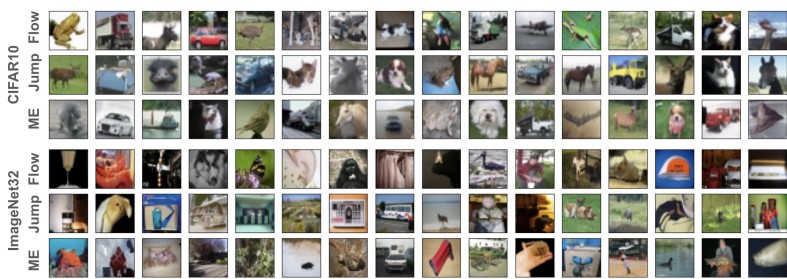

Figure 4: . Examples of generated images on CIFAR10 (top) and ImageNet32 (bottom).

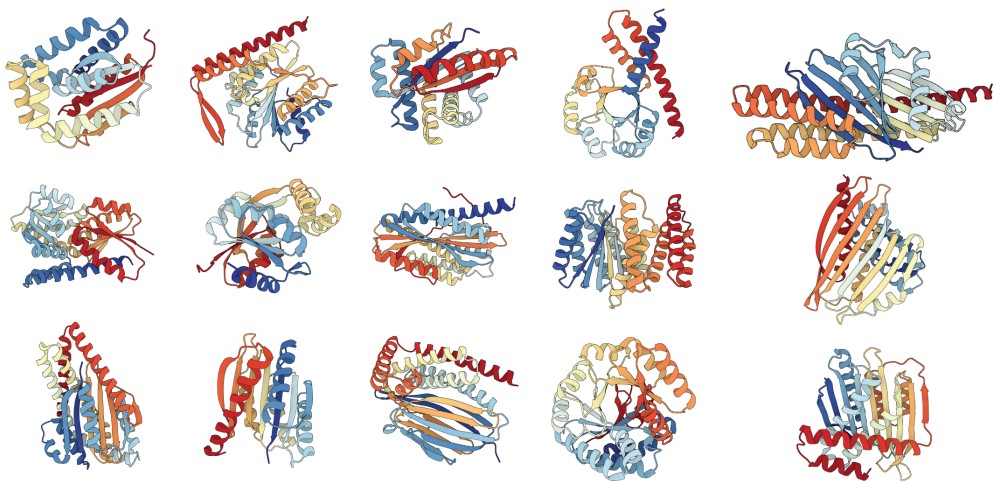

Figure 3: Examples of generated proteins with $SO(3)$ jumps model and MultiFlow. Each protein passes the designability filter check and is structurally unique.

---

**Algorithm 1** Generator Matching recipe for constructing Markov generative model (theory in black, implementation in brown)

---

**Step 0:** Choose prior $p_{\text{simple}}$
**Step 1:** Choose $p_t(dx|z)$ such that marginal $p_0 = p_{\text{simple}}$ and $p_1 = p_{\text{data}}$
**Step 2:** Find solution $\mathcal{L}_t^z$ to KFE
**Step 3:** Choose Bregman div. as loss
**Step 4:** Construct neural net $\mathcal{L}_t^\theta$
**Step 5:** Minimize CGM loss using $\mathcal{L}_t^z$
**Step 6:** Sample using Algorithm 2.

---

**Algorithm 2** Euler sampling for $S = \mathbb{R}^d$

---

**Given:** $u_t, \sigma_t, Q_t, p_{\text{simple}}$, step size $h > 0$
**Init:** $X_0 \sim p_{\text{simple}}$
  1: **for** $t$ in linspace$(0, 1, 1/h)$ **do**
  2:      Jump intensity $\lambda_t(X_t) = \int Q_t(dy; X_t)$
  3:      $\bar{X}_{t+h} \sim Q_t(\cdot; X_t)/\lambda_t(X_t)$
  4:      $m \sim \text{Bernoulli}(h\lambda_t(X_t)), \epsilon_t \sim \mathcal{N}(0, 1)$
  5:      $\tilde{X}_{t+h} = X_t + hu_t(X_t) + \sqrt{h}\sigma_t(X_t)\epsilon_t$
  6:      $X_{t+h} = m\bar{X}_{t+h} + (1 - m)\tilde{X}_{t+h}$
  7: **end for**
**Return:** $X_1$

---

## A OVERVIEW OF MARKOV PROCESSES AND THEIR GENERATORS

### A.1 SETUP AND DEFINITIONS

**State space.** Throughout this work, we assume $(S, d)$ is a Polish metric space, *i.e.*, $S$ is a set and there is a metric $d : S \times S \to \mathbb{R}_{\geq 0}$ defined on $S$ such that $(S, d)$ is complete (*i.e.*, any Cauchy sequence converges) and separable (*i.e.*, it has a countable dense subset). We endow $S$ with its Borel $\sigma$-algebra $B(S)$ and consider a set $A \subset S$ as measurable if $A \in B(S)$. Any function $f : S \to \mathbb{R}$ considered in this work is assumed to be measurable. Throughout this work, we assume that $\mathcal{T}$ is a set of functions $f : S \to \mathbb{R}$ on $S$ such two probability measures $\mu_1, \mu_2$ are equal if and only if $\mathbb{E}_{x \sim \mu_1}[f(x)] = \mathbb{E}_{x \sim \mu_2}[f(x)]$ for all $f \in \mathcal{T}$. We call elements in $\mathcal{T}$ test functions. Throughout this work, $\nu$ denote a reference measure on $S$ (e.g. the Lebesgue measure on $\mathbb{R}^d$ or the count measure on discrete spaces). We write $\frac{dp}{d\nu}(x)$ for a density of a probability measure $p$ with respect to the reference measure $\nu$.

**Markov process.** Let $(\Omega, \mathcal{F}, \mathbb{P})$ be a probability space. A Markov process $(X_t)_{0 \leq t \leq 1}$ is a collection of integrable random variables $X_t : \Omega \to S$ such that

$$\text{Markov assumption: } \mathbb{P}[X_{t_{n+1}} \in A | X_{t_1}, X_{t_2}, \ldots, X_{t_n}] = \mathbb{P}[X_{t_{n+1}} \in A | X_{t_n}] \tag{18}$$

$$\text{for all } 0 \leq t_1 < t_2 < \cdots < t_n < t_{n+1} \leq 1, A \subseteq S \text{ measurable} \tag{19}$$

We denote by $k_{t+h|t}(A|x) = \mathbb{E}[X_{t+h} \in A | X_t = x]$ the transition kernel of $X_t$.

**Semigroup.** We define the action of marginals $p_t$ and of transition kernels $k_{t+h|t}$ on test functions $f$ as in the main paper via:

$$\langle p_t, f \rangle = \int f(y) p_t(dy) = \mathbb{E}\left[f(X_t)\right] \qquad \blacktriangleright \text{ marginal action} \tag{20}$$

$$\left\langle k_{t+h|t}, f \right\rangle(x) = \int f(y) k_{t+h|t}(dy|x) = \mathbb{E}\left[f(X_{t+h}) | X_t = x\right] \qquad \blacktriangleright \text{ transition action} \tag{21}$$

where the marginal action maps each test function $f$ to a scalar $\langle p_t, f \rangle \in \mathbb{R}$, while transition action maps a real-valued function $x \mapsto f(x)$ to a another real-valued function $x \mapsto [\langle k_{t+h|t}, f \rangle](x)$. The tower property implies that $p_t \langle k_{t+h|t}, f \rangle = \langle p_{t+h}, f \rangle$. Considering $k_{t+h|t}$ as a linear operator as above, we know by the Markov assumption and the tower property that there are two fundamental properties that hold:

$$k_{t|t} = \text{Id}, \quad \langle p_{s|u}, f \rangle = \langle p_{t|u}, \langle p_{s|t}, f \rangle \rangle, \quad 0 \leq u < t < s \leq 1 \qquad \blacktriangleright \text{ composition} \tag{22}$$

$$\| \langle p_{s|t}, f \rangle \|_\infty \leq \|f\|_\infty, \quad 0 \leq t < s \leq 1 \qquad \blacktriangleright \text{ contraction} \tag{23}$$

where $\| \cdot \|_\infty$ describes the supremum norm.

### A.1.1 DEFINITIONS FOR TIME-HOMOGENEOUS MARKOV PROCESSES

A complication in the theory we develop here is that we (need to) consider time-inhomogeneous Markov processes, while most of the mathematical theory and literature resolves around time-homogeneous Markov processes. Therefore, we first give the definitions for time-homogeneous Markov processes and then explain how this can be translated to the time-inhomogeneous case.

A time-homogeneous Markov process is a Markov process $\bar{X}_t$ such that $k_{t+h|t} = k_{h|0}$ for all $t, h \geq 0$ - i.e. the evolution is constant in time. This implies the semigroup property

$$k_{t|0} = \text{Id}, \quad k_{s|0} \circ k_{t|0} = k_{s+t|0} \tag{24}$$

Let $C_0(S)$ be the space of continuous functions $f : S \to \mathbb{R}$ that vanish at infinity, i.e. for all $\epsilon > 0$ there exists a compact set $K \subset S$ such that $|f(x)| < \epsilon$ for all $x \in S \setminus K$.

**Feller process.** We call $\bar{X}_t$ a *Feller* process if it holds that

1. **Operators on continuous functions:** For any $f \in C_0(S)$ and $t \geq 0$, also $k_{t|0}f \in C_0(S)$.
2. **Strong continuity:** For any $f \in C_0(S)$:
$$\lim_{t \downarrow 0} \|k_{t|0}f - f\|_\infty = 0 \tag{25}$$

**Generator.** We define the generator $\mathcal{L}$ of a Feller process $\bar{X}_t$ as follows: for any $f \in C_0(S)$ such that the limit

$$\lim_{t \downarrow 0} \frac{k_{t|0}f - f}{t} \to \mathcal{L}f \tag{26}$$

exists uniformly in $S$ (i.e., in $\|\cdot\|_\infty$) and we define $\mathcal{L}f$ as the limit above. We define the **core** $D(\mathcal{L})$ as all $f$ for which that limit exists. It holds that $D(\mathcal{L})$ is a dense subspace of $C_0(S)$ (Pazy, 2012) and $\mathcal{L}$ is a linear operator.

### A.1.2 DEFINITIONS FOR TIME-INHOMOGENEOUS MARKOV PROCESSES

For a general time-inhomogeneous Markov process $X_t$, one can extend the definitions to a two-parameter semigroup, see (Rüschendorf et al., 2016) for example. Another approach is to simply consider the corresponding time-homogeneous Markov process $\bar{X}_t = (X_t, t)$ on extended state space $S \times [0, 1]$. The transition kernel on extended state space is then given for $A \subset S \times [0, 1]$ via:

$$\bar{p}_{h|0}(A, (x, t)) = k_{t+h|t}(A_{t+h}|x), \quad A_{t+h} := \{y|(y, t + h) \in A\} \tag{27}$$

and the action is given via

$$\bar{p}_{h|0}f(x, t) = \mathbb{E}[f(X_{t+h}, t + h)|X_t = x] \tag{28}$$

**Feller process.** The Feller assumption is equivalent to:

1. **Operators on continuous functions:** For any $f \in C_0(S \times [0, 1])$ and $t \geq 0$, also $k_{t|0}f \in C_0(S \times [0, 1])$.
2. **Strong continuity:** For any $f \in C_0(S \times [0, 1])$:
$$\lim_{t \downarrow 0} \|k_{t|0}f - f\|_\infty = 0 \tag{29}$$

where here the supremum norm $\|\cdot\|_\infty$ is taken across $S$ and time $[0, 1]$.

**Time-dependent generator.** In the above case, we can reshape the generator via

$$\mathcal{L}f(x, t) = \lim_{h \to 0} \frac{\mathbb{E}[f(X_{t+h}, t + h) - f(X_t, t)|X_t = x]}{h} \tag{30}$$

$$= \lim_{h \to 0} \frac{\mathbb{E}[f(X_{t+h}, t + h) - f(X_t, t + h) + f(X_t, t + h) - f(X_t, t)|X_t = x]}{h} \tag{31}$$

$$= \lim_{h \to 0} \left[\frac{\mathbb{E}[f(X_{t+h}, t + h) - f(X_t, t + h)|X_t = x]}{h}\right] + \lim_{h \to 0} \frac{f(x, t + h) - f(x, t)}{h} \tag{32}$$

$$= \underbrace{\lim_{h \to 0} \left[\frac{\mathbb{E}[f(X_{t+h}, t + h) - f(X_t, t + h)|X_t = x]}{h}\right]}_{=:\mathcal{L}_t f^t(x)} + \partial_t f(x, t) \tag{33}$$

$$= \mathcal{L}_t f^t(x) + \partial_t f(x, t) \tag{34}$$

where $f^t : S \to \mathbb{R}, x \mapsto f(x, t)$ describes the restriction of $f$ on $S$ and we defined the time-dependent generator $\mathcal{L}_t$ as an operator acting on spatial components, i.e. functions in $C_0(S)$, via

$$\mathcal{L}_t g(x) := \lim_{h \to 0} \frac{\mathbb{E}[g(X_{t+h}) - g(X_t)|X_t = x]}{h} \tag{35}$$

for any $g : S \to \mathbb{R}$ such that the limit exists. Note that however that when we define the time-dependent generator $\mathcal{L}_t f$, we also need to specify the direction operator $\partial_t$ (i.e. in what direction of time it goes). This is always implicitly assumed. Further, we remark that in the above derivation, we have assumed that $t \mapsto \mathcal{L}_t f^t$ is continuous (in supremum norm) in $t$ for any function in the domain of $\mathcal{L}_t$. We will state this now again.

## A.2 REGULARITY ASSUMPTIONS.

Throughout this work, we make the following regularity assumptions:

1. **Assumption on semigroup:** The Markov process $X_t$ is a Feller process in the sense defined in app. A.1.2.

2. **Assumption on sample paths:** In every time interval $[s, t]$, the expected number of discontinuities of $t \mapsto X_t$ is finite.

3. **Assumptions on test functions:** There exists a subspace of test functions $\mathcal{T}$ that is dense in $C_0(S)$ such that $\mathcal{T} \subset \text{dom}(\mathcal{L}_t)$ and the function $t \mapsto \mathcal{L}_t f$ is continuous for any $f \in \mathcal{T}$. Further, two probability distributions $\mu_1, \mu_2$ on $S$ are equal if and only if for all $f \in \mathcal{T}$ it holds that $\mathbb{E}_{x \sim \mu_1}[f(x)] = \mathbb{E}_{x \sim \mu_2}[f(x)]$.

4. **Assumptions on probability path:** Any probability path $(p_t)_{0 \leq t \leq 1}$ considered fulfils that the function $t \mapsto \langle p_t, f \rangle$ is continuous in $t$ for all $f \in \mathcal{T}$.

5. **KFE is sufficient to check probability path:** Let $(p_t)_{0 \leq t \leq 1}$ be a probability path on $S$. Then:

$$\begin{aligned} X_0 &\sim p_0 & &\text{(start with right initial distribution)} & (36)\\ \partial_t \langle p_t, f \rangle &= \langle p_t, \mathcal{L}_t f \rangle \text{ for all } f \in \mathcal{T} & &\text{(fulfill KFE)} & (37)\\ \Rightarrow X_t &\sim p_t \text{ for all } 0 \leq t \leq 1 & &\text{(marginals of } X_t \text{ are given by } p_t) & (38) \end{aligned}$$

i.e. if $X_t$ is initialized with the right initial distribution, its marginals will be given by $p_t$.

**Remark regarding assumption 5.** We note that assumption 5 is true under relatively weak assumptions and there is a diversity of mathematical literature on showing uniqueness of the solution of the KFE in $p_t$ for different settings. However, to the best of our knowledge, there is no known result that states regularity assumptions for general state spaces and Markov processes, which is why simply state it here as an assumption. For the machine learning practitioner, this assumption holds for any state space of interest. To illustrate this, we point the rich sources in the mathematical literature that show uniqueness that list the regularity assumptions for specific spaces and classes of Markov processes:

1. Flows in $\mathbb{R}^d$ and manifolds: (Villani et al., 2009, Mass conservation formula, page 15), (DiPerna & Lions, 1989), (Ambrosio, 2004)

2. Diffusion in $\mathbb{R}^d$ and manifolds: (Villani et al., 2009, Diffusion theorem, page 16)

3. General Ito-SDEs in $\mathbb{R}^d$: (Figalli, 2008, Theorem 1.3 and 1.4), (Kurtz, 2011, Corollary 1.3), (Bogachev et al., 2022)

4. Discrete state spaces: Here, the KFE is a linear ODE, which has a unique solution under the assumption that the coefficients are continuous.

**Remark regarding assumption 1.** In order for a generator of a Markov process to be defined, property 2 in the definition of a Feller process must hold (see section A.1.2). Property 1 in the definition of a Feller process (see section A.1.2), might not be stricly necessary. However, we use it because (i) most of the mathematical literature (including the one we cite) uses it and (ii) any function on a computer is defined on a compact set.

**Motivation for assumption 2.** Our framework would not break necessarily if we allowed for infinite number of discontinuities. We included this assumption because simulating an infinite number of discontinuities with a finite number of simulation steps would necessarily induce uncontrollable simulation error upon sampling and therefore most likely be not interesting for the purposes of generative modeling.

**Motivation for assumption 4.** As every probability path generated by a Feller process fulfils this assumption, it is reasonable to only consider probability paths that have that property as well.

### A.3 ADOINT KFE

The version of the KFE in eq. (11) determines the evolution of expectations of test functions $f$. This is necessary if we use probability distributions that do not have a density. If a density exists, a more familiar version of the KFE can be used that directly prescribe the change of the probability densities. To present it, we introduce the **adjoint generator** $\mathcal{L}_t^*$, which acts on probability densities $\frac{dp_t}{d\nu}(x)$ with respect to a reference measure $\nu$, namely $[\mathcal{L}_t^* p_t](x)$ is implicitly defined by the identity

$$\left\langle \frac{dp_t}{d\nu}, \mathcal{L}_t f \right\rangle_\nu = \left\langle \mathcal{L}_t^* \frac{dp_t}{d\nu}, f \right\rangle_\nu, \qquad \forall f \in \mathcal{T} \tag{39}$$

$$\Leftrightarrow \quad \int \frac{dp_t}{d\nu}(x) \mathcal{L}_t f(x) \nu(dx) = \int \mathcal{L}_t^* \frac{dp_t}{d\nu}(x) f(x) \nu(dx) \qquad \forall f \in \mathcal{T} \tag{40}$$

Now, equation 39 applied to the KFE (equation 11) we get

$$\int \partial_t \frac{dp_t}{d\nu}(x) f(x) \nu(dx) \tag{41}$$

$$= \partial_t \int \frac{dp_t}{d\nu}(x) f(x) \nu(dx) \tag{42}$$

$$= \partial_t \langle p_t, f \rangle \tag{43}$$

$$= \langle p_t, \mathcal{L}_t f \rangle \tag{44}$$

$$= \left\langle \frac{dp_t}{d\nu}, \mathcal{L}_t f \right\rangle_\nu \tag{45}$$

$$= \left\langle \mathcal{L}_t^* \frac{dp_t}{d\nu}, f \right\rangle_\nu \tag{46}$$

$$= \int [\mathcal{L}_t^* \frac{dp_t}{d\nu}](x) f(x) \nu(dx) \tag{47}$$

where "$v(dx)$" denotes the integration with respect to the reference measure $\nu$ on $S$. As this holds for all test functions $f$, we can conclude that this is equivalent to

$$\partial_t \frac{dp_t}{d\nu}(x) = [\mathcal{L}_t^* \frac{dp_t}{d\nu}](x) \text{ for all } x \in S \qquad \blacktriangleright \text{ Adjoint KFE} \tag{48}$$

which is an equivalent version of the KFE that we call **adjoint KFE**. In this form, the KFE generalizes many famous equations used to develop generative models such as the Continuity Equation or the Fokker-Planck Equation (Song et al., 2020; Lipman et al., 2022) (see table 1). Whenever a probability density exists, we use the adjoint KFE. We will derive several examples of adjoint generators and adjoint KFEs in app. A.5.

### A.4 TIME-REVERSAL

In diffusion models (Song et al., 2020), the notion of a time-reversal of a diffusion process plays a crucial role in the construction of the generative model. We discuss here the idea of time-reversal of a stochastic process and explain why we build our framework based on probability paths and solutions to the Kolmogorov Forward Equation (KFE) and how this includes the diffusion construction.

**Probability paths vs noising/forward process.** In the context of diffusion models, one usually corrupts (or "noises") data via a "forward process". With our time parameterization (where $t = 1$ corresponds to data), this is represented by a Markov process $(\bar{X}_t)_{0 \le t \le 1}$ running backwards in time. Every such Markov process also defines a conditional probability path $p_t(\cdot|z) = \bar{k}_{t|1}(\cdot|z)$ via its transition kernel $\bar{k}$ and the marginal probability $p_t$ corresponds to the marginals, i.e. $X_t \sim p_t$. **Therefore, diffusion models specify a probability path via the forward process/data corruption process**. We use probability path as opposed to a noising process because (1) it is often easier to define a probability path as opposed to a full transition kernel (2) it is the only thing that is used during training (only the marginals of the probability path are used during training, see eq. (17) - this includes diffusion model training as a special example). Further, we show in this work how one probability path can have different Markov processes that generate the same probability path (see fig. 2 for examples). However, an explicit data corruption process via a Markov process can be still a useful heuristic to find good probability paths.

**The notion of the time-reversal by Anderson (1982) is stronger than the one needed for generative modeling.** To explain this, we briefly define two notions of time-reversals. Let $(X_t)_{0 \le t \le 1}$ be a Markov process running in forward time and $(\bar{X}_t)_{0 \le t \le 1}$ a Markov process running in backward time. The Markov process $X_t$ is called a *time-reversal* of $\bar{X}_t$ if the joint distributions coincide, i.e. if

$$\mathbb{P}[X_{t_1} \in A_1, \ldots, X_{t_n} \in A_n] = \mathbb{P}[\bar{X}_{t_1} \in A_1, \ldots, \bar{X}_{t_n} \in A_n] \tag{49}$$

$$\text{for all } 0 \le t_1, \ldots, t_n \le 1, A_1, \ldots, A_n \subset S \text{ measurable} \tag{50}$$

A famous example is the time-reversal of stochastic differential equations by Anderson (1982). Next, we introduce here the notion of a *weak time-reversal*. We call $X_t$ a weak time-reversal of $\bar{X}_t$ if the marginals coincide, i.e. if

$$\mathbb{P}[X_t \in A] = \mathbb{P}[\bar{X}_t \in A] \text{ for all } 0 \le t \le 1, A \subset S \text{ measurable} \tag{51}$$

Obviously, any strong time-reversal gives us a weak time-reversal but not vice versa necessarily. For the purposes of generative modeling, we often only use the final point $X_1$ of the Markov process (e.g. as a generated image) and discard earlier time points. Therefore, whether a Markov process is a "true" time-reversal or weak time-reversal does not matter for most applications. A famous example of a weak time reversal that is not a "strong" time-reversal is the *probability flow ODE* in diffusion models (Song et al, 2020). The probability flow ODE does *not* constitute a time-reversal of a diffusion process in the sense of Anderson (1982) but it constitutes a weak time-reversal (i.e. it has the same marginals). As the probability flow ODE is currently the state-of-the-art method for low NFEs, it illustrates that **finding "true" time-reversals is a harder mathematical problem to solve but (often) not necessary to solve for the purposes of generative modeling and might give suboptimal results**. Therefore, we restrict ourselves to weak time-reversals in this work.

**Using the KFE to find weak time-reversals.** Let $(\bar{X}_t)_{0 \le t \le 1}$ be a Markov process running in backward time. How can we find or check whether a Markov process $(X_t)_{0 \le t \le 1}$ running in forward time is a weak time reversal of $(\bar{X}_t)_{0 \le t \le 1}$? Let $(p_t)_{0 \le t \le 1}$ be the marginals of $\bar{X}_t$, then $p_t$ defines a probability path. Further, by definition $(X_t)_{0 \le t \le 1}$ is a weak time reversal of $(\bar{X}_t)_{0 \le t \le 1}$ if and only if $X_t$ generates the probability path $(p_t)_{0 \le t \le 1}$. By the KFE (see eq. (11)), this holds if and only if $X_0 \sim p_0$ and

$$\langle p_t, \mathcal{L}_t f \rangle = \partial_t \langle p_t, f \rangle = - \langle p_t, \bar{\mathcal{L}}_t f \rangle \quad \text{for all } f \in \mathcal{T}, 0 \le t \le 1 \tag{52}$$

where in the second equation we use the KFE in reverse-time (we have to flip signs as time is running backwards). Therefore, finding a time-reversal reduces to finding a generator $\mathcal{L}_t$ such that

$$\langle p_t, \mathcal{L}_t f \rangle = -\langle p_t, \bar{\mathcal{L}}_t f \rangle \quad \text{for all } f \in \mathcal{T}, 0 \leq t \leq 1 \tag{53}$$

This allows us to find weak time-reversal for many Markov processes. We illustrate this in app. H.2 on the example of diffusion processes. However, actual time-reversals like the one by Anderson (1982) can still be helpful for our purposes as a way to find a solution to the KFE. If $p_t$ is a probability path given by the marginals of a Markov a process $\bar{X}_t$, its strong time-reversal is also a weak time-reversal - so we can use it to find a solution to the KFE. Therefore, **finding time-reversals can of Markov processes can be a tool to find solutions to the KFE**. We avoid using time-reversal as it imposes additional mathematical complexity that is not needed for the vanilla generative modeling task.

## A.5 PROPERTIES OF IMPORTANT MARKOV PROCESSES (DERIVATIONS FOR TABLE 1)

In this section, we include the definition of important classes of Markov processes and their properties including their generators.

### A.5.1 FLOWS

**Definition.** Let $S = \mathbb{R}^d$ and $u : \mathbb{R}^d \times \mathbb{R} \to \mathbb{R}^d, (x, t) \mapsto u_t(x)$ be a time-dependent vector field. Then a flow $\phi_{t,s}(x)$ is the solution to the ODE:

$$\frac{d}{dt}\phi_{t,s}(x) = u_t(\phi_{t,s}(x), t), \quad \phi_{s,s}(x) = x \tag{54}$$

It is clear that $\phi_{t,s}$ is a deterministic Markov transition kernel $p_{t|s}$, i.e. $p_{t|s}(\cdot; x)$ is a delta distribution for every $x \in \mathbb{R}^d$. Euler sampling of the ODE corresponds to

$$X_{t+h} = X_t + h u_t(X_t) + o(h) \tag{55}$$

**Generator.** Let $\mathcal{T} = C_c^\infty(\mathbb{R}^d)$ be the space of infinitely differentiable and smooth functions with compact support (test function in $\mathbb{R}^d$). Then we can compute the generator via

$$[\mathcal{L}_t f](x) = \lim_{h \to 0} \frac{\mathbb{E}\left[f(X_t + h u_t(X_t) + o(h)) | X_t = x\right] - f(x)}{h} \tag{56}$$

$$= \lim_{h \to 0} \frac{h \nabla f(x)^T u_t(x) + o(h)}{h} = \nabla f(x)^T u_t(x), \tag{57}$$

where we used a first-order Taylor approximation and the limit is uniform by the fact that $f$ is in $C_c^\infty(\mathbb{R}^d)$.

**Adjoint and adjoint KFE.** Let's assume that $p_t$ has a density $\frac{dp_t}{d\nu}(x)$ with respect to the Lebesgue measure that is bounded and continuously differentiable. In the following paragraph, we write $p_t(x) = \frac{dp_t}{d\nu}(x)$ for its density. Then we can compute the adjoint generator $\mathcal{L}_t^*$ via

$$\langle p_t, \mathcal{L}_t f \rangle = \mathbb{E}_{x \sim p_t}[\mathcal{L}_t f(x)] \tag{58}$$

$$= \int \mathcal{L}_t f(x) p_t(x) dx \tag{59}$$

$$= \int \nabla f(x)^T u_t(x) p_t(x) dx \tag{60}$$

$$= \int f(x)[-\nabla \cdot [u_t(x) p_t(x)]] dx \tag{61}$$

$$= \int f(x)[\mathcal{L}_t^* p_t](x) dx \tag{62}$$

by partial integration. Therefore, the adjoint generator is given by $\mathcal{L}_t^* p_t = -\nabla \cdot [u_t(x)p_t(x)]$. Using the adjoint KFE, we get the well-known **continuity equation**

$$\partial_t p_t(x) = -\nabla \cdot [u_t p_t](x) \tag{63}$$

### A.5.2 DIFFUSION

**Definition.** Let $S = \mathbb{R}^d$ and $\sigma_t : \mathbb{R}^d \times \mathbb{R} \to \mathbb{R}^{d \times d}, (x, t) \mapsto \sigma_t(x)$ be a time-dependent function mapping to symmetric positive semi-definite matrices $\sigma_t$ in a continuous fashion. A diffusion process with diffusion coefficient $\sigma_t$ is defined via the infinitesimal sampling procedure:

$$X_{t+h} = X_t + \sqrt{h}\sigma_t(X_t)\epsilon_t \tag{64}$$

where $\epsilon_t \sim \mathcal{N}(0, I)$. For a more formal definition, see (Oksendal, 2013).

**Generator.** Let $\mathcal{T} = C_c^\infty(\mathbb{R}^d)$ be the space of infinitely differentiable and smooth functions with compact support (test function in $\mathbb{R}^d$). Then we can compute the generator via

$$[\mathcal{L}_t f](x) = \lim_{h \to 0} \frac{\mathbb{E}\left[f(X_t + \sqrt{h}\sigma_t(X_t)\epsilon_t + o(h))|X_t = x\right] - f(x)}{h} \tag{65}$$

$$= \lim_{h \to 0} \frac{\mathbb{E}[f(x) + \nabla f(x)^T \sqrt{h}\sigma_t(x)\epsilon_t + \frac{1}{2}h[\sigma_t(x)\epsilon_t]^T \nabla^2 f(x)[\sigma_t(x)\epsilon_t] - f(x)]}{h} \tag{66}$$

$$= \lim_{h \to 0} \frac{\nabla f(x)^T \sqrt{h}\sigma_t(x)\mathbb{E}[\epsilon_t] + \mathbb{E}[\frac{1}{2}h[\sigma_t(x)\epsilon_t]^T \nabla^2 f(x)[\sigma_t(x)\epsilon_t]]}{h} \tag{67}$$

$$= \frac{1}{2}\mathbb{E}[\epsilon_t^T [\sigma_t(x)]^T \nabla^2 f(x)[\sigma_t(x)]\epsilon_t]] \tag{68}$$

$$= \frac{1}{2}\text{Trace}\left(\sigma_t(x)^T \nabla^2 f(x)\sigma_t(x)\right) \tag{69}$$

$$= \frac{1}{2}\text{Trace}[\sigma_t(x)\sigma_t(x)^T \nabla^2 f(x)] \tag{70}$$

$$= \frac{1}{2}\sigma_t^2(x) \cdot \nabla^2 f(x) \tag{71}$$

where we used a 2nd order Taylor approximation (2nd order because $\mathbb{E}[\|\sqrt{h}\epsilon_t\|^2] \propto h$) and the symmetry of $\sigma_t^2$. Further, we use $A \cdot B = \text{trace}(AB)$ to denote the matrix inner product.

**Adjoint and adjoint KFE.** Let's assume that $p_t$ has a density $\frac{dp_t}{d\nu}(x)$ with respect to the Lebesgue measure that is bounded and continuously differentiable. In the following paragraph, we write $p_t(x) = \frac{dp_t}{d\nu}(x)$ for its density. We can compute the adjoint generator $\mathcal{L}_t^*$ via

$$\langle p_t, \mathcal{L}_t f \rangle = \mathbb{E}_{x \sim p_t}[\mathcal{L}_t f(x)] \tag{72}$$

$$= \int \mathcal{L}_t f(x)p_t(x)dx \tag{73}$$

$$= \frac{1}{2}\int \sigma_t^2(x) \cdot \nabla^2 f(x)p_t(x)dx \tag{74}$$

$$= \frac{1}{2}\int f(x)\nabla^2 \cdot [\sigma_t^2 p_t](x)dx \tag{75}$$

$$= \int f(x)[\mathcal{L}_t^* p_t](x)dx \tag{76}$$

by partial integration. Using the fact that this holds for all test functions, we can convert the KFE to the adjoint KFE recovering the well-known **Fokker-Planck equation**

$$\partial_t p_t(x) = \nabla^2 \cdot [\sigma_t^2 p_t](x) \tag{77}$$

### A.5.3 Jumps

Let us assume that we consider a jump process defined by a time-dependent kernel $Q_t(dy; x)$, i.e. for every $0 \le t \le 1$ and every $x \in S$, $Q_t(dy; x)$ is a positive measure over $S \setminus \{x\}$. The idea of a jump process is that the total volume assigned to $S$

$$\lambda_t(x) = \int Q_t(dy; x) \tag{78}$$

gives the **jump intensity**, i.e. the infinitesimal likelihood of jumping. Further, if $\lambda_t(x) > 0$, we can assign a **jump distribution** by normalizing $Q_t$ to a probability kernel

$$J_t(dy; x) = \frac{Q_t(dy; x)}{\lambda_t(x)} \tag{79}$$

The infinitesimal sampling procedure is as follows:

$$X_{t+h} = \begin{cases} X_t & \text{with probability } 1 - h\lambda_t(X_t) + o(h) \\ \sim J_t(dy; X_t) & \text{with probability } h\lambda_t(X_t) + o(h) \end{cases} \tag{80}$$

We derive the generator here in an informal way. For a rigorous treatment of jump processes, see for example (Davis, 1984). Up to $o(h)$-approximation error, the generator is then given by

$$\mathcal{L}_t f(x) = \lim_{h \to 0} \frac{\mathbb{E}[f(X_{t+h}) - f(X_t)|X_t = x]}{h} \tag{81}$$

$$= \lim_{h \to 0} \frac{\mathbb{E}[f(X_{t+h}) - f(X_t)|X_t = x, \text{Jump in } [t, t+h]]\mathbb{P}[\text{Jump in } [t, t+h]]}{h} \tag{82}$$

$$+ \lim_{h \to 0} \underbrace{\frac{\mathbb{E}[f(X_{t+h}) - f(X_t)|X_t = x, \text{No jump in } [t, t+h]]\mathbb{P}[\text{No jump in } [t, t+h]]}{h}}_{=0} \tag{83}$$

$$= \lim_{h \to 0} \frac{\mathbb{E}_{y \sim J_t(dy; x)}[f(y) - f(x)]h\lambda_t(x)}{h} \tag{84}$$

$$= \mathbb{E}_{y \sim J_t(dy; x)}[f(y) - f(x)]\lambda_t(x) \tag{85}$$

$$= \int (f(y) - f(x))Q_t(dy; x) \tag{86}$$

where we have used that if $X_t$ does not jump in $[t, t+h]$, then $X_{t+h} = X_t$.

**Adjoint and adjoint KFE.** Let's assume that $p_t$ has a density $\frac{dp_t}{d\nu}(x)$ with respect to the Lebesgue measure that is bounded and continuously differentiable. In the following paragraph, we write $p_t(x) = \frac{dp_t}{d\nu}(x)$ for its density. Let's assume that the jump measures $Q_t(dy; x)$ is given via a kernel $Q_t : S \times S \to \mathbb{R}_{\ge 0}, (y, x) \mapsto Q_t(y; x)$ such that

$$\int f(y)Q_t(dy; x) = \int f(y)Q_t(y; x)\nu(dy) \tag{87}$$

where $\nu(dy)$ denotes the integration with respect to the reference measure $\nu$ on $S$. Then we can derive the adjoint generator as follows:

$$\langle p_t, \mathcal{L}_t f \rangle (x) = \int \int (f(y) - f(x)) Q_t(y; x) \nu(dy) p_t(x) \nu(dx) \tag{88}$$

$$= \int \int f(y) Q_t(y; x) p_t(x) \nu(dy) \nu(dx) - \int \int f(x) Q_t(y; x) p_t(x) \nu(dy) \nu(dx) \tag{89}$$

$$= \int \int f(x) Q_t(x; y) p_t(y) \nu(dy) \nu(dx) - \int \int f(x) Q_t(y; x) p_t(x) \nu(dy) \nu(dx) \tag{90}$$

$$= \int f(x) \underbrace{\left[ \int Q_t(x; y) p_t(y) - Q_t(y; x) p_t(x) \nu(dy) \right]}_{=: \mathcal{L}_t^* p_t} \nu(dx) \tag{91}$$

$$\tag{92}$$

where we have seen that $\mathcal{L}_t^*$ describes the adjoint generator. With this, we get the **jump continuity equation** as adjoint KFE

$$\partial_t p_t(x) = \int \underbrace{Q_t(x; y) p_t(y)}_{\text{inflow}} - \underbrace{Q_t(y; x) p_t(x)}_{=\text{outflow}} \nu(dy) \tag{93}$$

### A.5.4 CONTINUOUS-TIME MARKOV CHAIN (CTMC)

Let us consider a continuous-time Markov chain $X_t$ on a discrete space $S$ with $|S| < \infty$. We define this to be a jump process as defined in app. A.5.3. However, we can find a convenient parameterization for the jump process. Specifically, we can convert integrals into sums and see that there is a jump kernel $Q_t \in \mathbb{R}_{\geq 0}^{S \times S}$ given such that for all $x \in S$ it holds that

$$\mathcal{L}_t f(x) = \sum_{y \in S} [f(y) - f(x)] Q_t(y; x) = \sum_{y \neq x} [f(y) - f(x)] Q_t(y; x) \tag{94}$$

A natural convention people follow is to set $Q_t(x; x) = - \sum_{y \neq x} Q_t(y; x)$. This gives us the rate of staying at $x$ (However, note that $Q_t(y; x)$ does not describe a kernel of a positive measure anymore.) With this constraint, we get

$$\mathcal{L}_t f(x) = \sum_{y \in S} f(y) Q_t(y; x) = f^T Q_t \tag{95}$$

where we consider $f = (f(x))_{x \in S}$ as a row vector. The adjoint is simply given by the adjoint multiplication $p_t \mapsto Q_t p_t$ and we get that the adjoint KFE is given by

$$\partial_t p_t(x) = \sum_{y \in S} Q_t(x; y) p_t(y) \tag{96}$$

To sample the next time step given $X_t = x$, we get

$$X_{t+h} = \begin{cases} \sim \frac{Q_t(y;x)}{\sum\limits_{y \neq x} Q_t(y;x)} & \text{with probability } h \sum\limits_{y \neq x} Q_t(y;x) + o(h) \\ x & \text{with probability } 1 - h \sum\limits_{y \neq x} Q_t(y;x) + o(h) \end{cases} \tag{97}$$

$$= \begin{cases} \sim \frac{Q_t(y;x)}{\sum\limits_{y \neq x} Q_t(y;x)} & \text{with probability } h \sum\limits_{y \neq x} Q_t(y;x) + o(h) \\ x & \text{with probability } 1 + hQ_t(x;x) + o(h) \end{cases} \tag{98}$$

$$= \begin{cases} y & \text{with probability } hQ_t(y;x) + o(h) \text{ (for all } y \in S) \\ x & \text{with probability } 1 + hQ_t(x;x) + o(h) \end{cases} \tag{99}$$

$$\sim (I + hQ_t)(\cdot;x) \tag{100}$$

where we simply sample from the stochastic matrix $I + hQ_t$ (a matrix whose columns sum to 1).

### A.6 LINEAR PARAMETERIZATION OF GENERATORS

We describe here what we understand under a linear parameterization of a generator. This is the basis for parameterizing a generator in a neural network that can be implemented in a computer. For simplicity, we fix a $t \in [0,1]$. As before, $\mathcal{T}$ be the space of test functions and $B(S)$ be the space of bounded functions on $S$. Then the space $A(\mathcal{T}) := \{\mathcal{L} : \mathcal{T} \to B(S) | \mathcal{L} \text{ is linear}\}$ of linear operators on $\mathcal{T}$ is itself again a vector space. Let $W \subset A(\mathcal{T})$ be a subspace. Then a **linear parameterization** of $W$ is given by 2 components: (1) a convex closed set $\Omega \subset V$ that is a subset of a vector space with an inner product $\langle \cdot, \cdot \rangle_V$ and (2) a linear operator $\mathcal{K} : \mathcal{T} \to C(S; V)$ such that every $\mathcal{L}_t \in W$ can be written as

$$\mathcal{L}_t f(x) = \langle \mathcal{K}f(x); F_t(x) \rangle_V \tag{101}$$

for a continuous function $F_t : S \to \Omega$. We list several examples to make this abstract definition more concrete.

1. **Flows in $S = \mathbb{R}^d$:** $\mathcal{T} = C_c^\infty(\mathbb{R}^d)$ and $\Omega = V = \mathbb{R}^d$. Let $W$ be the space of linear operators given via generators of flows, i.e.

$$\mathcal{L}_t f = \nabla f^T u_t, \quad u_t : \mathbb{R}^d \to \mathbb{R}^d \tag{102}$$

Setting $\mathcal{K}f = \nabla f$ and $F_t = u_t$ we recover the shape of eq. (101).

2. **Diffusion in $S = \mathbb{R}^d$:** $\mathcal{T} = C_c^\infty(\mathbb{R}^d)$ and $\Omega = S_d^{++} \subset \mathbb{R}^{d \times d} = V$, where $S_d^{++}$ denotes the set of all positive semi-definite matrices. Let $W$ is the space of linear operators given via generators of diffusion, i.e.

$$\mathcal{L}_t f = \nabla^2 f \cdot \sigma_t^2, \quad \sigma_t : \mathbb{R}^d \to S_d^{++} \tag{103}$$

Setting $\mathcal{K}f = \nabla^2 f$ and $F_t = \sigma_t^2$ we recover the shape of eq. (101).

3. **Jumps in $S = \mathbb{R}^d$:** $\mathcal{T} = C_c^\infty(\mathbb{R}^d)$ and $\Omega = \{a : \mathbb{R}^d \to \mathbb{R}_{\geq 0}, a \text{ continuous}\} \subset C^1(\mathbb{R}^d, \mathbb{R}) = V$. On $V$, a dot product is defined via $\langle a, b \rangle_V = \int a(x)b(x)dx$. Let $W$ be the space of linear operators given via generators of jumps, i.e.

$$\mathcal{L}_t f(x) = \int f(y) - f(x)Q_t(y;x)dy = \langle Df(x), Q_t(\cdot;x) \rangle_V \tag{104}$$

where we set $\mathcal{K}f(x)$ as the function $y \mapsto f(y) - f(x)$. Setting $F_t = Q_t$ we recover the shape of eq. (101).

4. **Continuous-time Markov chains:** Let $S$ be discrete and $Q_t \in \mathbb{R}^{S \times S}$ be a rate matrix of a continuous-time Markov chain. Then for any $f \in \mathbb{R}^S$

$$\mathcal{L}_t f(x) = f^T Q_t(\cdot; x) = \langle f, Q_t(\cdot; x) \rangle_V \tag{105}$$

where $V = \mathbb{R}^S$ and $Df = f$ and $\langle \cdot, \cdot \rangle_V$ is the standard Euclidean dot product. With this, we recover the shape of eq. (101).

## B  SAMPLING WITH UNIVERSAL REPRESENTATION (ALG. 2)

In alg. 2, we summarize a sampling procedure to sample a generator for $S = \mathbb{R}^d$ with universal representation described in theorem 1. We briefly want to derive that alg. 2 is in fact a method a sampling procedure that simulates a Markov process with the correct generator up to $o(h)$ approximation error. Let us assume that $X_t$ is a Markov process that is obtained by sampling as in alg. 2 (limit process for $h \to 0$). Then its generator is given via:

$$\mathcal{L}_t f(x) \tag{106}$$

$$= \lim_{h \to 0} \frac{1}{h} \left( \mathbb{E}[f(X_{t+h}) - f(X_t)|X_t = x] \right) \tag{107}$$

$$= \lim_{h \to 0} \frac{\mathbb{E}[f(X_{t+h}) - f(X_t)|X_t = x, \text{Jump in } [t, t+h)]\mathbb{P}[\text{Jump in } [t, t+h)]}{h} \tag{108}$$

$$+ \lim_{h \to 0} \underbrace{\frac{\mathbb{E}[f(X_{t+h}) - f(X_t)|X_t = x, \text{No jump in } [t, t+h)]\mathbb{P}[\text{No jump in } [t, t+h)]}{h}}_{=0} \tag{109}$$

$$= \lim_{h \to 0} \frac{\mathbb{E}_{y \sim Q_t(dy;x)/\lambda_t(x)}[f(y) - f(x)]h\lambda_t(x)}{h} \tag{110}$$

$$+ \lim_{h \to 0} \frac{[\mathbb{E}[f(X_t + hu_t(X_t) + \sqrt{h}\sigma_t(X_t)\epsilon_t) - f(X_t)|X_t = x, \text{No jump in } [t, t+h)]](1 - h\lambda_t(x))}{h} \tag{111}$$

$$= \int (f(y) - f(x))Q_t(dy;x) \tag{112}$$

$$+ \lim_{h \to 0} \frac{\mathbb{E}[f(X_t + hu_t(X_t) + \sqrt{h}\sigma_t(X_t)\epsilon_t) - f(X_t)|X_t = x, \text{No jump in } [t, t+h)]}{h} \tag{113}$$

$$= \int (f(y) - f(x))Q_t(dy;x) \tag{114}$$

$$+ \lim_{h \to 0} \frac{\mathbb{E}[h\nabla f(x)^T[u_t(X_t) + \sqrt{h}\sigma_t(X_t)\epsilon_t)] + \frac{1}{2}h^2[u_t(X_t) + \sqrt{h}\sigma_t(X_t)\epsilon_t)]^T\nabla^2 f(x)[u_t(X_t) + \sqrt{h}\sigma_t(X_t)\epsilon_t)]]}{h} \tag{115}$$

$$= \int (f(y) - f(x))Q_t(dy;x) \tag{116}$$

$$+ \nabla f(x)^T u_t(x) + \lim_{h \to 0} \frac{\mathbb{E}[\frac{1}{2}[\sigma_t(X_t)\epsilon_t)]^T\nabla^2 f(x)[\sigma_t(X_t)\epsilon_t)]]}{h} \tag{117}$$

$$= \int (f(y) - f(x))Q_t(dy;x) + \nabla f(x)^T u_t(x) + \frac{1}{2}\text{Trace}\left[\sigma_t(X_t)\nabla^2 f(x)\sigma_t(X_t)\right] \tag{118}$$

$$= \int (f(y) - f(x))Q_t(dy;x) + \nabla f(x)^T u_t(x) + \frac{1}{2}\sigma_t^2(X_t) \cdot \nabla^2 f(x) \tag{119}$$

This finishes the proof.

## C  PROOFS

### C.1  PROOF OF THEOREM 1

Let $(X_t)_{0 \leq t \leq 1}$ be a continuous-time Markov process. For now, we assume that $X_t$ is time-homogeneous. Let $\mathcal{L}$ be the generator of $X_t$ defined via

$$\mathcal{L}f(x) = \lim_{h \to 0} \frac{\mathbb{E}[f(X_{t+h}) - f(X_t)|X_t = x]}{h} \tag{120}$$

We know that $\mathcal{L}$ has the following property:

$$f(x) = 0, f \geq 0 \quad \Rightarrow \quad \mathcal{L}f(x) \geq 0 \quad \text{(``almost positive'')} \tag{121}$$

An operator having the above property is called **almost positive**. The theory of almost positive operators is well-established. Specifically, we can use universal representations of almost positive operators as established in (von Waldenfels, 1965, Satz 1). Related theorems can be found in (Courrege, 1965) for characterizing differential operators satisfying the absolute maximum principle (that is closely related to the almost positive principle) and in weaker form for Markov processes (Feller, 1955). Here we use (von Waldenfels, 1965, Satz 1). Specifically, we know that $\mathcal{L}$ must have the form

$$\mathcal{L}f(x) = -c(x)f(x) + u(x)^T \nabla f(x) + \frac{1}{2}\sigma^2(x) \cdot \nabla^2 f(x) \tag{122}$$

$$+ \int\limits_{y \neq x} \left[ f(y) - f(x) - y^T \nabla f(x) \mathbf{1}_{\|y-x\|<1} \right] Q(dy; x) \tag{123}$$

where $c(x) \geq 0, u(x) \in \mathbb{R}^d$ are continuous functions and $\sigma^2(x)$ is a positive semi-definite matrix continuous in $x$. For every $x$, $Q(dy; x)$ is a measure on $\mathbb{R}^d \setminus \{x\}$. The term $c(x)$ corresponds to a process that is "dying". As we assume that our process runs from $t = 0$ to $t = 1$, we can drop $c(x)$. Further, using the assumption that there only exists a finite number of discontinuities (see app. A.2), we can discard the term $y^T \nabla f(x) \mathbf{1}_{\|y-x\|<1}$ (alternatively, one can redefine the drift to $\bar{u}(x) = u(x) - \int y \mathbf{1}_{\|y-x\|<1} Q(dy; x)$). Therefore, we can rewrite the above formula as:

$$\mathcal{L}f = u(x)^T \nabla f(x) + \frac{1}{2}\sigma^2(x) \cdot \nabla^2 f(x) + \int\limits_{y \neq x} (f(y) - f(x))Q(dy; x) \tag{124}$$

### C.1.1  TIME-INHOMOGENOUS CASE

Let $(X_t)_{0 \leq t \leq 1}$ be a continuous-time Markov process. Now, we allow $X_t$ to be time-inhomogenous. To make it time-homogenous, we define $\bar{X}_t = (X_t, t)$. Let $\bar{\mathcal{L}}$ be the generator of $\bar{X}_t$. Then for a test function $f : \mathbb{R}^d \times \mathbb{R} \to \mathbb{R}$ we get

$$\bar{\mathcal{L}}f = \bar{u}(x,t)^T \nabla_{x,t} f(x,t) + \frac{1}{2}\bar{\sigma}^2(x,t) \cdot \nabla_{x,t}^2 f(x,t) + \int (f(y,s) - f(x,s))Q(dy, ds; x, t) \tag{125}$$

where $\bar{u}, \bar{\sigma}^2, Q_t$ are operators over the extended space $\mathbb{R}^d \times \mathbb{R}$, i.e. $\bar{u}(x,t) = (\bar{u}_x(x,t), \bar{u}_t(x,t))$ and

$$\bar{\sigma}(x,t) = \begin{pmatrix} \bar{\sigma}_{x,x}(x,t) & \bar{\sigma}_{x,t}(x,t) \\ \bar{\sigma}_{t,x}(x,t) & \bar{\sigma}_{t,t}(x,t) \end{pmatrix} \tag{126}$$

However, note that since marginal process in $t$ is deterministic and has derivative one, it must necessarily hold (by uniqueness of the representation) that $\bar{u}_t(x,t) = 1$ and $\bar{\sigma}^2(x,t)_{t,x} = \bar{\sigma}^2(x,t)_{x,t} = \bar{\sigma}^2(x,t)_{t,t} = 0$ and $Q(\cdot,\cdot;x,t)$ is supported over $\mathbb{R}^d \times \{t\}$ - i.e. is a time-dependent kernel $Q_t(dy;x)$. Therefore, we can rewrite the above equation as:

$$\bar{\mathcal{L}}f = \frac{\partial}{\partial t}f(x,t) + u_t(x)^T \nabla_x f(x,t) + \frac{1}{2}\sigma_t^2(x) \cdot \nabla_x^2 f(x,t) + \int (f(y,t) - f(x,t))Q_t(dy;x) \quad (127)$$

Rewriting $\bar{\mathcal{L}}$ as a time-dependent generator $\mathcal{L}_t$ (see app. A.2), we get for a time-independent test function $f : \mathbb{R}^d \to \mathbb{R}$

$$\mathcal{L}_t f = u_t(x)^T \nabla_x f(x) + \frac{1}{2}\sigma_t^2(x) \cdot \nabla_x^2 f(x) + \int (f(y) - f(x))Q_t(dy;x) \quad (128)$$

This finishes the proof.

## C.2 PROOF OF PROPOSITION 1

Let $k_{t+h|t}(\cdot|x,z)$ be the conditional transition kernel for the conditional Markov process $X_t^z$ with conditional generator $\mathcal{L}_t^z$ that solves the KFE for the conditional probability path $p_t(dx|z)$. Then we can compute for the marginal probability path $p_t$ and a test function $f$ that:

$$\partial_t \langle p_t, f \rangle = \lim_{h \to 0} \frac{1}{h} \langle p_{t+h}, f \rangle - \langle p_t, f \rangle \quad (129)$$

$$= \lim_{h \to 0} \frac{1}{h} [\mathbb{E}_{z \sim p_{\text{data}}, x' \sim p_{t+h}(\cdot|z)}[f(x')] - \mathbb{E}_{z \sim p_{\text{data}}, x \sim p_t(\cdot|z)}[f(x)]] \quad (130)$$

$$= \lim_{h \to 0} \frac{1}{h} [\mathbb{E}_{z \sim p_{\text{data}}, x \sim p_t(\cdot|z), x' \sim k_{t+h|t}(\cdot|x,z)}[f(x')] - \mathbb{E}_{z \sim p_{\text{data}}, x \sim p_t(\cdot|z)}[f(x)]] \quad (131)$$

$$= \lim_{h \to 0} \frac{1}{h} [\mathbb{E}_{z \sim p_{\text{data}}, x \sim p_t(\cdot|z), x' \sim k_{t+h|t}(\cdot|x,z)}[f(x') - f(x)]] \quad (132)$$

$$= \lim_{h \to 0} \frac{1}{h} \mathbb{E}_{x \sim p_t(x)} \left[ \mathbb{E}_{z \sim p_t(dz|x)} \left[ \left[ \mathbb{E}_{x' \sim k_{t+h|t}(\cdot|x,z)}[f(x')] - f(x) \right] \right] \right] \quad (133)$$

$$= \mathbb{E}_{x \sim p_t(x)} \left[ \mathbb{E}_{z \sim p_t(dz|x)} \left[ \lim_{h \to 0} \frac{1}{h} \left[ \mathbb{E}_{x' \sim k_{t+h|t}(\cdot|x,z)}[f(x')] - f(x) \right] \right] \right] \quad (134)$$

$$= \mathbb{E}_{x \sim p_t(x)} [\underbrace{\mathbb{E}_{z \sim p_t(dz|x)}[\mathcal{L}_t^z f(x)]}_{=: \mathcal{L}_t f(x)}] \quad (135)$$

$$= \mathbb{E}_{x \sim p_t}[\mathcal{L}_t f(x)] \quad (136)$$

$$= \langle p_t, \mathcal{L}_t f \rangle \quad (137)$$

Therefore, we see that the marginal generator $\mathcal{L}_t$ defined as above solves the KFE for the marginal probability path $p_t$.

**Proof that proposition 1 also holds for any linear parameterization of the generator.** Let us assume that the conditional generator has a linear parameterization in the sense defined in app. A.6 given via

$$\mathcal{L}_t^z(x) = \langle \mathcal{K}f(x); F_t^z(x) \rangle_V \quad (138)$$

for a function $F_t^z : S \to V$. The proof proceeds exactly as above to get

$$\partial_t \langle p_t, f \rangle = \mathbb{E}_{x \sim p_t(x)}[\mathbb{E}_{z \sim p_t(dz|x)}[\mathcal{L}_t^z f(x)]] \tag{139}$$

$$= \mathbb{E}_{x \sim p_t(x)}[\mathbb{E}_{z \sim p_t(dz|x)}[\langle \mathcal{K}f(x); F_t^z(x) \rangle_V]] \tag{140}$$

$$= \mathbb{E}_{x \sim p_t(x)}[\langle \mathcal{K}f(x); \underbrace{\mathbb{E}_{z \sim p_t(dz|x)}[F_t^z(x)]}_{=:F_t(x)} \rangle] \tag{141}$$

$$= \langle p_t, \mathcal{L}_t f \rangle \tag{142}$$

Therefore, we can see that also the marginal generator has linear parameterization given via $F_t : S \to \Omega$. Note that it holds that $F_t(x) \in \Omega$ because $\Omega$ is convex and closed.

### C.3   PROOF OF PROPOSITION 2

#### C.3.1   EXAMPLE OF BREGMAN DIVERGENCES

We list two motivating examples of an illustration for Bregman divergences (see eq. (15)):

1. **Mean squared-error:** Setting $\phi : \mathbb{R}^d \to \mathbb{R}, x \mapsto \|x\|^2$ leads to

$$D(x, y) = \phi(x) - \phi(y) - \langle x - y, \nabla\phi(y) \rangle \tag{143}$$

$$= \|x\|^2 - \|y\|^2 - \langle x - y, 2y \rangle \tag{144}$$

$$= \|x\|^2 - 2\langle x, y \rangle + \|y\|^2 \tag{145}$$

$$= \|x - y\|^2 \tag{146}$$

2. **KL-divergence:** Define the probability simplex as $\Delta_{d-1} = \{x \in \mathbb{R}_{\geq 0}^d | \sum_{i=1}^d x = 1\}$. Then define

$$\phi : \Delta_{d-1} \to \mathbb{R}, x \mapsto \sum_{i=1}^d x_i \log x_i \tag{147}$$

$$\Rightarrow \quad D(x, y) = \phi(x) - \phi(y) - \langle x - y, \nabla\phi(y) \rangle \tag{148}$$

$$= \sum_{i=1}^d x_i \log x_i - \sum_{i=1}^d y_i \log y_i - \sum_{i=1}^d (x_i - y_i)(1 + \log x_i) \tag{149}$$

$$= \sum_{i=1}^d y_i \log \frac{y_i}{x_i} \tag{150}$$

#### C.3.2   BREGMAN DIVERGENCE AS SUFFICIENT CONDITION FOR PROPOSITION 2

First, we show that $\nabla_\theta L_{\mathrm{gm}}(\theta) = \nabla_\theta L_{\mathrm{cgm}}(\theta)$ for any Bregman divergence. Let $V$ be an arbitrary vector space, $\Omega \subset V$ be a convex subset, and $D : \Omega \times \Omega \to \mathbb{R}_{\geq 0}$ be a Bregman divergence on it defined via

$$D(a, b) = \phi(a) - [\phi(b) + \langle a - b, \nabla\phi(b) \rangle], \quad a, b \in \Omega \tag{151}$$

for a convex function $\phi : \Omega \to \mathbb{R}$. We rewrite $D$ as

$$D(a, b) = A(a) + \langle a, B(b) \rangle + C(b) \tag{152}$$

with functions $A(a) = \phi(a), B(b) = \nabla\phi(b), C(b) = -\phi(b) + \langle b, \nabla\phi(b) \rangle$.

With this, we get:

$$L_{\text{gm}}(\theta) = \mathbb{E}_{t \sim \text{Unif}, x \sim p_t} \left[ D(F_t(x), F_t^\theta(x)) \right] \tag{153}$$

$$= \mathbb{E}_{t \sim \text{Unif}, x \sim p_t} \left[ A(F_t(x)) + \left\langle F_t(x), B(F_t^\theta(x)) \right\rangle + C(F_t^\theta(x)) \right] \tag{154}$$

$$= \mathbb{E}_{t \sim \text{Unif}, x \sim p_t} \left[ A(F_t(x)) + \left\langle \int F_t^z(x) p_t(dz|x), B(F_t^\theta(x)) \right\rangle + C(F_t^\theta(x)) \right] \tag{155}$$

$$= \mathbb{E}_{t \sim \text{Unif}, x \sim p_t} \left[ A(F_t(x)) + \int \left\langle F_t^z(x), B(F_t^\theta(x)) \right\rangle p_t(dz|x) + C(F_t^\theta(x)) \right] \tag{156}$$

$$= \mathbb{E}_{t \sim \text{Unif}, x \sim p_t} \left[ \int \left[ A(F_t^z(x)) + \left\langle F_t^z(x), B(F_t^\theta(x)) \right\rangle + C(F_t^\theta(x)) \right] p_t(dz|x) \right] + \text{const} \tag{157}$$

$$= \mathbb{E}_{t \sim \text{Unif}, x \sim p_t, z \sim p_t(\cdot|x)} \left[ D(F_t^z(x), F_t(x)) \right] + \text{const} \tag{158}$$

$$= \mathbb{E}_{t \sim \text{Unif}, z \sim p_{\text{data}}, x \sim p_t(\cdot|z)} \left[ D(F_t^z(x), F_t(x)) \right] + \text{const} \tag{159}$$

$$= L_{\text{cgm}}(\theta) + \text{const} \tag{160}$$

i.e. both losses only different by a constant in $\theta$. Hence, their gradients with respect to $\theta$ are the same. This proves proposition 2 for any Bregman divergence $D$ eq. (162).

### C.3.3 Bregman divergence as necessary condition for proposition 2

We now show that the Bregman divergence property is a necessary condition for $\nabla_\theta L_{\text{gm}}(\theta) = \nabla_\theta L_{\text{cgm}}(\theta)$ to hold for arbitrary probability paths and data distributions.

To do so, we first prove an equivalent characterization of Bregman divergences. For simplicity and because any representation used in a computer will be finite-dimensional, we restrict ourselves here to fine-dimensional vector spaces, i.e. we set $V = \mathbb{R}^k$ for some $k \in \mathbb{N}$.

**Lemma 1.** *Let $\Omega \subset \mathbb{R}^k$ to be a convex closed subset such that its interior $\Omega^\circ$ is convex and dense in $\Omega$. Let $D : \Omega \times \Omega \to \mathbb{R}_{\geq 0}$ be an arbitrary cost function, here defined as smooth function such that $D(a, b) = 0$ if and only if $a = b$. Then the following statements are equivalent:*

1. ***Bregman divergence:*** *The function $D$ is a Bregman divergence, i.e. there exists a strictly convex function $\phi : \Omega \to \mathbb{R}$ such that*

$$D(a, b) = \phi(a) - [\phi(b) + \langle a - b, \nabla\phi(b) \rangle], \quad \text{for all } a, b \in \Omega \tag{161}$$

2. ***Target-affine loss function:*** *There exist functions $A : \Omega \to \mathbb{R}, B : \Omega \to \mathbb{R}^k, C : \Omega \to \mathbb{R}$ such that*

$$D(a, b) = A(a) + \langle a, B(b) \rangle + C(b) \tag{162}$$

3. ***Gradient in second argument is linear:*** *For every $a_1, a_2 \in \Omega$ and $\lambda_1, \lambda_2 \geq 0$ such that $\lambda_1 + \lambda_2 = 1$ it holds that*

$$\nabla_b D(\lambda_1 a_1 + \lambda_2 a_2, b) = \lambda_1 \nabla_b D(a_1, b) + \lambda_2 \nabla_b D(a_2, b) \tag{163}$$

**Proof of** $(1) \Rightarrow (2)$. This is trivial. Define $A(a) = \phi(a), B(b) = \nabla\phi(b), C(b) = -\phi(b) + \langle b, \nabla\phi(b) \rangle$.

**Proof of** $(2) \Rightarrow (1)$. Let us assume we have

$$D(a, b) = A(a) + \langle a, B(b) \rangle + C(b) \tag{164}$$

The condition that $D$ is a valid differentiable cost function implies that

$$\arg\min_a D(a, b) = b \quad \text{for all } b \in \Omega \tag{165}$$

$$\Rightarrow 0 = \nabla_a D(a, b)_{|a=b} \tag{166}$$

$$= \nabla A(b) + B(b) \text{ for all } b \in \Omega \tag{167}$$

This is equivalent to $B = -\nabla A$. Therefore, we get that

$$D(a, b) = A(a) - \langle a, \nabla A(b) \rangle + C(b) \tag{168}$$

Further, we require that $D(a, a) = 0$ for all $a$, which implies that

$$0 = D(a, a) = A(a) - \langle a, \nabla A(a) \rangle + C(a) \quad \text{for all } a \in \Omega \tag{169}$$

$$\Rightarrow \quad C(a) = \langle a, \nabla A(a) \rangle - A(a) \quad \text{for all } a \in \Omega \tag{170}$$

$$\Rightarrow D(a, b) = A(a) - \langle a, \nabla A(b) \rangle + \langle b, \nabla A(b) \rangle - A(b) \quad \text{for all } a, b \in \Omega \tag{171}$$

$$= A(a) - A(b) - \langle a - b, \nabla A(b) \rangle \quad \text{for all } a \in \Omega \tag{172}$$

Finally, since $D(a, b) \geq 0$ it must hold that

$$A(a) \leq A(b) + \langle a - b, \nabla A(b) \rangle \text{ for all } a, b \in \Omega \tag{173}$$

The above is equivalent to the fact that $A$ is a convex function. Therefore, setting $A(a) = \phi(a)$, we get that $D$ is a Bregman divergence.

**Proof of** $(2) \Rightarrow (3)$. Let us assume that

$$D(a, b) = A(a) + \langle a, B(b) \rangle + C(b) \tag{174}$$

Then:

$$\nabla_b D(\lambda_1 a_1 + \lambda_2 a_2, b) = \langle \lambda_1 a_1 + \lambda_2 a_2, \nabla_b B(b) \rangle + \nabla_b C(b) \tag{175}$$

$$= \lambda_1 \langle a_1, \nabla_b B(b) \rangle + \lambda_2 \langle a_2, \nabla_b B(b) \rangle + \lambda_1 \nabla_b C(b) + \lambda_2 \nabla_b C(b) \tag{176}$$

$$= \lambda_1 \nabla_b D(a_1, b) + \lambda_2 \nabla_b D(a_2, b) \tag{177}$$

**Proof of** $(3) \Rightarrow (2)$. Let us assume that eq. (163) holds. Fix a $b \in \Omega^\circ$. We first show that eq. (163) implies that the function $a \mapsto \nabla_b D(a, b)$ is an affine function. To see this, because of the required condition, the function $F = (F_1, \ldots, F_d) : a \mapsto \nabla_b D(a, b)$ fulfills the condition that each $F_i$ is both convex and concave. In particular, $\nabla^2 F_i$ and $-\nabla^2 F_i$ are both positive semi-definite. This implies that $\nabla^2 F_i = 0$. In turn, this implies that $\nabla F_i$ is a constant function. In turn, this implies that the Jacobian $\nabla F$ is constant. This implies that $F$ is affine, i.e. that

$$\nabla_b D(a, b) = \tilde{B}(b)a + \tilde{C}(b) \tag{178}$$

for a $\tilde{C} : \mathbb{R}^k \to \mathbb{R}^k$ and $\tilde{B} : \mathbb{R}^k \to \mathbb{R}^k \times \mathbb{R}^k$. Further, since $D$ is twice continuously differentiable

$$\nabla_b \nabla_a D(a, b) = \nabla_a \nabla_b D(a, b)^T = \tilde{B}(b)^T \tag{179}$$

which implies that $\tilde{B}(b)^T = \nabla_b B(b)$ for the function $B = \nabla_a D(a, b) : \Omega \to \mathbb{R}^k$. In turn, this implies that $\tilde{C}(b) = \nabla_b C(b)$ for the function $C(b) = D(a, b) - \langle a, B(b) \rangle$. Then we get:

$$\nabla_b D(a, b) = \nabla_b[\langle a, B(b) \rangle + C(b)] \quad \text{for all } a, b \in \Omega \tag{180}$$

$$\Rightarrow D(a, b) = A(a) + \langle a, B(b) \rangle + C(b) \text{ for all } a, b \in \Omega \tag{181}$$

for some function $A : \Omega \to \mathbb{R}$. This finishes the proof.

**Necessity of Bregman divergence property for proposition 2 to hold.** We show the necessity of the Bregman divergence property by giving a counterexample. Fix a $0 \leq t < 1$. Let us assume that the network $F_t^\theta$ is just a constant function given by a vector $\theta \in \Omega$. Now, let us suppose that $D$ is not a Bregman divergence. Then by lemma 1(c), there exist some $a_1, a_2, b \in \Omega$ and $\lambda_1, \lambda_2 \geq 0$ with $\lambda_1 + \lambda_2 = 1$ such that

$$\nabla_b D(\lambda_1 a_1 + \lambda_2 a_2, b) \neq \lambda_1 \nabla_b D(a_1, b) + \lambda_2 \nabla_b D(a_2, b) \tag{182}$$

Then define the data distribution as $p_{\text{data}} = \lambda_1 \cdot \delta_{z_1} + \lambda_2 \cdot \delta_{z_2}$ and the conditional distribution at time $t$ simply as $p_t(\cdot|z) = p_{\text{data}}$, i.e. a distribution that just resamples independently from $p_{\text{data}}$. Then $p_t(dz|x) = p_{\text{data}}(dz)$. Further, let there be a conditional KFE solution parameterized by $F_t^z$ such that $F_t^{z_1}(x) = a_1$, $F_t^{z_2}(x) = a_2$ for all $x$ (note that we can choose an arbitrary parameterization and we can choose $p_{t'}(\cdot|z)$ arbitrarily for any $t < t' < 1$). Finally, set $\theta = b$. Then

$$F_t(x) = \mathbb{E}_{z \sim p_t(dz|x)}[F_t^z(x)] = \mathbb{E}_{z \sim p_{\text{data}}}[F_t^z(x)] = \lambda_1 a_1 + \lambda_2 a_2 \tag{183}$$

and

$$\nabla_\theta L_{\text{gm}}(\theta) = \nabla_\theta \mathbb{E}[D(F_t(z), F_t^\theta(z))] \tag{184}$$
$$= \nabla_\theta D(\lambda_1 a_2 + \lambda_2 a_2, \theta) \tag{185}$$
$$\neq \nabla_\theta[\lambda_1 D(a_1, \theta) + \lambda_2 D(a_2, \theta)] \tag{186}$$
$$= \nabla_\theta \mathbb{E}_{z \sim p_{\text{data}}}[D(F_t^z(z), \theta)] \tag{187}$$
$$= \nabla_\theta \mathbb{E}_{z \sim p_{\text{data}}, x \sim p_t(\cdot|z)}[D(F_t^z(x), \theta)] \tag{188}$$
$$= \nabla_\theta L_{\text{cgm}}(\theta) \tag{189}$$

Therefore, the gradients are not the same and proposition 2 does not hold. This shows that the Bregman divergence property is necessary if we want to desired property to hold for arbitrary data distributions, probability paths, and network parameterizations.

## C.4 PROOF OF PROPOSITION 3

Let $\mathcal{L}_t, \mathcal{L}'_t$ be two generators of two Markov processes that solve the KFE for a probability path $p_t$. Then for $\alpha_t^1, \alpha_t^2 \in \mathbb{R}$ with $\alpha_t^1 + \alpha_t^2 = 1$ it holds that:

$$\langle p_t, (\alpha_t^1 \mathcal{L}_t + \alpha_t^2 \mathcal{L}'_t) f \rangle \tag{190}$$
$$= \alpha_t^1 \langle p_t, \mathcal{L}_t f \rangle + \alpha_t^2 \langle p_t, \mathcal{L}'_t f \rangle \tag{191}$$
$$= \alpha_t^1 \partial_t \langle p_t, f \rangle + \alpha_t^2 \partial_t \langle p_t, f \rangle \tag{192}$$
$$= (\alpha_t^1 + \alpha_t^2) \partial_t \langle p_t, f \rangle \tag{193}$$
$$= \partial_t \langle p_t, f \rangle \tag{194}$$

*i.e.*, $\alpha_t^1 \mathcal{L}_t + \alpha_t^2 \mathcal{L}'_t$ is again a solution of the KFE. A small but important detail is whether $\alpha_t^1, \alpha_t^2$ are positive or negative and whether $\mathcal{L}_t, \mathcal{L}'_t$ correspond to actual Markov processes - and if yes, in forward or backward time. As explained in app. A.5, the generator of a time-inhomogeneous Markov process is given by a spatial generator $\mathcal{L}_t$ operating on spatial components plus an associated time-derivative $\partial_t$ operator (as explained, it is usually ignored but relevant here). Therefore, both the spatial components have to sum to 1 and the time-derivative operators. Therefore, we have to add the time-derivative operators up (Markov superposition), set their weight to zero (divergence-free components), or flip their sign (predictor-corrector) in order to make the KFE hold. This leads to the 3 different use-cases described.

## C.5 PROOF OF PROPOSITION 4

First, we state a formal version of proposition 4. We assume that we have a setup where have a Generator Matching model given for each state space $S_1, S_2$. Specifically, let $q_t^1(\cdot|z_1), q_t^2(\cdot|z_2)$ be two conditional

probability paths on state spaces $S_1, S_2$. Let $\bar{\mathcal{L}}_t^z, \tilde{\mathcal{L}}_t^z$ be solutions to the conditional KFE for $q_t^1(\cdot|z_1), q_t^2(\cdot|z_2)$. In the sense defined as in app. A.6, we assume that there are linear parameterizations $\bar{F}_t^{z_1} : S \to \Omega_1 \subset V_1, \tilde{F}_t^{z_2} : S \to \Omega_1 \subset V_1$ of $\bar{\mathcal{L}}_t^{z_1}, \tilde{\mathcal{L}}_t^{z_2}$ given such that

$$\bar{\mathcal{L}}_t^{z_1} f_1(x_1) = \left\langle \mathcal{K}_1 f_1(x_1), \bar{F}_t^{z_1}(x_1) \right\rangle_{V_1}, \quad \tilde{\mathcal{L}}_t^{z_2} f(x_2) = \left\langle \mathcal{K}_2 f_2(x_2), \tilde{F}_t^{z_2}(x_2) \right\rangle_{V_2} \tag{195}$$

for all test functions $f_1 : S_1 \to \mathbb{R}, f_2 : S_2 \to \mathbb{R}$ and for two linear operators $\mathcal{K}_1, \mathcal{K}_2$. On the product space $V_1 \times V_2$, we define an inner product via

$$\langle (a_1, b_1), (a_2, b_2) \rangle_{V_1 \times V_2} = \langle a_1, a_2 \rangle_{V_1} + \langle b_1, b_2 \rangle_{V_2} \tag{196}$$

Let $X_t^1, X_t^2$ be two marginal Markov processes with corresponding marginal generators $\bar{\mathcal{L}}_t, \tilde{\mathcal{L}}_t$ with parameterization $F_t^1, F_t^2$. We assume that there is a simulation kernels $T^1, T^2$ given that are independent for the Markov process (e.g. Euler sampling in table 1 or an ODE solver), i.e. simulating the Markov process

$$X_{t+h}^1 \sim T_{t+h|t}^1(\cdot|X_t^1, F_t^1(X_t^1)) \tag{197}$$

$$X_{t+h}^2 \sim T_{t+h|t}^2(\cdot|X_t^2, F_t^2(X_t^2)) \tag{198}$$

leads to a valid simulation with the correct generators, i.e. for all test functions $f$:

$$\bar{\mathcal{L}}_t f(x_t) = \lim_{h \to 0} \frac{1}{h} \mathbb{E}_{x_{t+h} \sim T_{t+h|t}^1(\cdot|X_t^1, F_t^1(X_t^1))} [f(x_{t+h}) - f(x_t)] \tag{199}$$

$$\tilde{\mathcal{L}}_t f(x_t) = \lim_{h \to 0} \frac{1}{h} \mathbb{E}_{x_{t+h} \sim T_{t+h|t}^2(\cdot|X_t^2, F_t^2(X_t^2))} [f(x_{t+h}) - f(x_t)] \tag{200}$$

**Proposition 5** (Multimodal generative models - Formal version). *Let there be a data distribution $p_{data}$ be given over $S_1 \times S_2$. Define the **conditional factorized path** as $p_t(\cdot|z_1, z_2) = q_t^1(\cdot|z_1)q_t^2(\cdot|z_2)$ and the **marginal factorized path** via $\mathbb{E}_{(z_1, z_2) \sim p_{data}}[p_t(dx|z_1, z_2)]$. Then:*

1. **Conditional generator:** *The conditional factorized path $p_t(\cdot|z_1, z_2) = q_t^1(\cdot|z_1)q_t^2(\cdot|z_2)$ on $S_1 \times S_2$ admits a KFE solution given by a Markov process $X_t = (X_t^1, X_t^2)$ where $X_t^1, X_t^2$ are independent Markov processes with marginals $q_t^1(\cdot|z_1), q_t^2(\cdot|z_2)$ and generator*

$$\mathcal{L}_t^z f(x_1, x_2) = [\bar{\mathcal{L}}_t^{z_1} f^{x_2}](x_1) + [\tilde{\mathcal{L}}_t^{z_2} f^{x_1}](x_2), \quad z = (z_1, z_2) \tag{201}$$

   *where $f^{x_1}(y) = f(x_1, y)$ describes the restriction of a test function $f : S_1 \times S_2 \to \mathbb{R}$ on $S_2$. In particular, a linear parameterization of $\mathcal{L}_t^z$ is given for $x = (x_1, x_2) \in S_1 \times S_2$ via*

$$\mathcal{L}_t^z f(x_1, x_2) = \langle \mathcal{K} f(x), F_t^z(x) \rangle_{V_1 \times V_2}, \quad \mathcal{K} f(x) = (\mathcal{K}_1 f^{x_2}(x_1), \mathcal{K}_2 f^{x_1}(x_2)) \tag{202}$$

$$F_t^z(x) = (F_t^{z_1}(x_1), F_t^{z_2}(x_2)) \tag{203}$$

   *for all test functions $f : S_1 \times S_2 \to \mathbb{R}$.*

2. **Marginal generator:** *The marginal generator of a multimodal generative model (with conditional generator as in eq. (201)) is given by*

$$\mathcal{L}_t f(x_1, x_2) = \int \bar{\mathcal{L}}_t^{z_1} f^{x_2}(x_1) p_{1|t}(dz_1|x_1, x_2) + \int \tilde{\mathcal{L}}_t^{z_2} f^{x_1}(x_2) p_{1|t}(dz_2|x_1, x_2) \tag{204}$$

$$=: \bar{\mathcal{L}}_t f^{x_2}(x_1, x_2) + \tilde{\mathcal{L}}_t f^{x_1}(x_1, x_2) \tag{205}$$

   *In particular, the above generator can be linearly parameterized by*

$$\mathcal{L}_t f(x) = \langle \mathcal{K} f(x), F_t(x) \rangle_{V_1 \times V_2}, \quad \mathcal{K} f(x) = (\mathcal{K}_1 f^{x_2}(x_1), \mathcal{K}_2 f^{x_1}(x_2)) \tag{206}$$

   *for a function $F_t : S_1 \times S_2 \to \Omega_1 \times \Omega_2$.*

3. **Loss function:** *For two Bregman divergences $D_1 : \Omega_1 \times \Omega_1 \to \mathbb{R}$, $D_2 : \Omega_2 \times \Omega_2 \to \mathbb{R}$, the product*

$$D((a_1, a_2), (b_1, b_2)) = D(a_1, b_1) + D(a_2, b_2), \quad a_1, b_1 \in \Omega_1, a_2, b_2 \in \Omega_2 \tag{207}$$

*is again a Bregman divergence. To train a linear parameterization represented by a neural network $F_t^\theta = (F_{t,1}^\theta, F_{t,2}^\theta) : S_1 \times S_2 \to \Omega_1 \times \Omega_2$ we can train it with the sum of two CGM losses*

$$\mathbb{E}_{t \sim Unif, z \sim p_{data}, x_1 \sim p_t(\cdot|z_1), x_2 \sim p_t(\cdot|z_2)} \left[ D_1(F_t^{z_1}(x_1), F_{t,1}^\theta(x_1, x_2)) + D_2(F_t^{z_2}(x_2), F_{t,2}^\theta(x_1, x_2)) \right] \tag{208}$$

4. **Sampling:** *A valid simulation procedure is given by updating each dimension independently, i.e.*

$$X_{t+h} = \begin{pmatrix} X_{t+h}^1 \\ X_{t+h}^2 \end{pmatrix} \tag{209}$$

$$X_{t+h}^1 \sim T_{t+h|t}^1(\cdot|X_t^1, F_{t,1}^\theta(X_t^1, X_t^2)) \tag{210}$$

$$X_{t+h}^2 \sim T_{t+h|t}^2(\cdot|X_t^2, F_{t,2}^\theta(X_t^1, X_t^2)) \tag{211}$$

*where $X_{t+h}^1, X_{t+h}^2$ are sampled independently, i.e. the above procedure simulates a Markov process with generator as given in eq. (204) for $h \to 0$ (note that each $F_{t,i}^\theta$ depends on both modalities).*

Note that there are several striking features about eq. (204) and eq. (206): (1) each summand depends only on the marginal posterior $p_{1|t}(dz_i|x_1, x_2)$ per single modality ($i = 1, 2$). The dependency of the posterior across modalities does *not* influence the marginal generator $\mathcal{L}_t$. (2) The shape of $\mathcal{L}_t$ can be easily parameterized into a neural network by taking the parameterizations per modality and making the input depend on all modalities jointly (i.e. the dimension of the parameterization scales linearly with the dimension - if we parameterized a transition kernels, it would scale exponentially).

**Proof for conditional generator.** It holds that:

$$\partial_t \langle p_t(\cdot|z_1, z_2), f \rangle \tag{212}$$

$$= \partial_t [\mathbb{E}_{x_1 \sim p_t(\cdot|z_1), x_2 \sim p_t(\cdot|z_2)}[f(x_1, x_2)]] \tag{213}$$

$$= \lim_{h \to 0} \frac{1}{h} \left[ \mathbb{E}_{x_1 \sim p_{t+h}(\cdot|z_1), x_2 \sim p_{t+h}(\cdot|z_2)}[f(x_1, x_2)] - \mathbb{E}_{x_1 \sim p_t(\cdot|z_1), x_2 \sim p_t(\cdot|z_2)}[f(x_1, x_2)] \right] \tag{214}$$

$$= \lim_{h \to 0} \frac{1}{h} (\mathbb{E}_{x_1 \sim p_{t+h}(\cdot|z_1), x_2 \sim p_{t+h}(\cdot|z_2)}[f(x_1, x_2)] - \mathbb{E}_{x_1 \sim p_t(\cdot|z_1), x_2 \sim p_{t+h}(\cdot|z_2)}[f(x_1, x_2)] \tag{215}$$

$$+ \mathbb{E}_{x_1 \sim p_t(\cdot|z_1), x_2 \sim p_{t+h}(\cdot|z_2)}[f(x_1, x_2)] - \mathbb{E}_{x_1 \sim p_t(\cdot|z_1), x_2 \sim p_t(\cdot|z_2)}[f(x_1, x_2)]) \tag{216}$$

$$= \lim_{h \to 0} \frac{1}{h} \mathbb{E}_{x_2 \sim p_{t+h}(\cdot|z_2)}[\mathbb{E}_{x_1 \sim p_{t+h}(\cdot|z_1)}[f^{x_2}(x_1)] - \mathbb{E}_{x_1 \sim p_t(\cdot|z_1)}[f^{x_2}(x_1)]] \tag{217}$$

$$+ \mathbb{E}_{x_1 \sim p_t(\cdot|z_1)} \left[ \lim_{h \to 0} \frac{1}{h} \left[ \mathbb{E}_{x_2 \sim p_{t+h}(\cdot|z_2)}[f^{x_1}(x_2)] - \mathbb{E}_{x_2 \sim p_t(\cdot|z_2)}[f^{x_1}(x_2)]) \right] \right] \tag{218}$$

$$= \mathbb{E}_{x_2 \sim p_t(\cdot|z_2)} \left[ \mathbb{E}_{x_1 \sim p_t(\cdot|z_1)}[\mathcal{L}_t' f^{x_2}(x_1)] \right] + \mathbb{E}_{x_1 \sim p_t(\cdot|z_1)} \left[ \mathbb{E}_{x_2 \sim p_t(\cdot|z_2)}[\tilde{\mathcal{L}}_t f^{x_1}(x_2)] \right] \tag{219}$$

$$= \langle p_t(\cdot|z_1, z_2), \mathcal{L}_t f \rangle \tag{220}$$

where $\mathcal{L}_t$ is defined as in eq. (201).

**Proof of marginal generator shape.**    To show statement 2, we can compute that

$$\mathcal{L}_t f(x_1, x_2) = \int \mathcal{L}_t^z f(x_1, x_2) p_{1|t}(dz_1, dz_2 | x_1, x_2) \tag{221}$$

$$= \int \bar{\mathcal{L}}_t^{z_1} f^{x_2}(x_1) + \tilde{\mathcal{L}}_t^{z_2} f^{x_1}(x_2) p_{1|t}(dz_1, dz_2 | x_1, x_2) \tag{222}$$

$$= \int \bar{\mathcal{L}}_t^{z_1} f^{x_2}(x_1) p_{1|t}(dz_1, dz_2 | x_1, x_2) + \int \tilde{\mathcal{L}}_t^{z_2} f^{x_1}(x_2) p_{1|t}(dz_1, dz_2 | x_1, x_2) \tag{223}$$

$$= \int \bar{\mathcal{L}}_t^{z_1} f^{x_2}(x_1) p_{1|t}(dz_1 | x_1, x_2) + \int \tilde{\mathcal{L}}_t^{z_2} f^{x_1}(x_2) p_{1|t}(dz_2 | x_1, x_2) \tag{224}$$

where we used in the last equation that summand $i = 1, 2$ only depends on $z_i$.

**Proof of Loss shape**    We use the equivalent characterization of Bregman divergences as target-affine loss functions (see lemma 1(2)). Let $D_1(a, b), D_2(c, d)$ are two Bregman divergence functions on $V_1, V_2$ for $a, b \in V_1, c, d \in V_2$ in shape

$$D_1(a, b) = A_1(a) + \langle a, B_1(b) \rangle_{V_1} + C_1(b) \tag{225}$$

$$D_2(c, d) = A_2(c) + \langle c, B_2(d) \rangle_{V_2} + C_2(d) \tag{226}$$

$$\tag{227}$$

Then define:

$$D((a, c), (b, d)) := D_1(a, b) + D_2(c, d) \tag{228}$$

$$= [A_1(a) + A_2(c)] + \langle (a, c), (B_1(b), B_2(d)) \rangle_{V_1 \times V_2} + [C_1(b) + C_2(d)] \tag{229}$$

where the last equation shows that $D$ is again of the shape as outlined in lemma 1(b), i.e. it is again a Bregman divergence. For a parameterization $F_t^\theta(x_1, x_2) = (F_{t,1}^\theta(x_1, x_2), F_{t,2}^\theta(x_1, x_2)) \in V_1 \times V_2$ of the generator, we therefore get the loss given by

$$L_{\text{cgm}}(\theta) = \mathbb{E}_{t \sim \text{Unif}, z \sim p_{\text{data}}, x_1 \sim p_t(\cdot | z_1), x_2 \sim p_t(\cdot | z_2)} \left[ D(F_t^z(x), F_t^\theta(x)) \right] \tag{230}$$

$$= \mathbb{E}_{t \sim \text{Unif}, z \sim p_{\text{data}}, x_1 \sim p_t(\cdot | z_1), x_2 \sim p_t(\cdot | z_2)} \left[ D((F_t^{z_1}(x_1), F_t^{z_2}(x_2)), F_t^\theta(x)) \right] \tag{231}$$

$$= \mathbb{E}_{t \sim \text{Unif}, z \sim p_{\text{data}}, x_1 \sim p_t(\cdot | z_1), x_2 \sim p_t(\cdot | z_2)} \left[ D_1(F_t^{z_1}(x_1), F_{t,1}^\theta(x)) + D_2(F_t^{z_2}(x_2), F_{t,2}^\theta(x)) \right] \tag{232}$$

where $F_t^{z_1}, F_t^{z_2}$ describe the ground truth parameterizations of the conditional generators for each modality. In app. F, we see a concrete example of this loss construction.

**Sampling.**    For readability, we drop the parameter $\theta$ and write $F_t^1, F_t^2$ for two general parameterizations of Markov processes. We assume that eq. (199) and eq. (200) hold. Let $X_{t+h}$ be a Markov process that is simulated as specified in eq. (209). Our goal is to show that $X_{t+h}$ has the desired shape of the generator, i.e. that for a test function $f : S_1 \times S_2 \to \mathbb{R}$ it holds

$$\mathcal{L}_t f(x_1, x_2) = \lim_{h \to 0} \frac{1}{h} \left[ \mathbb{E}_{x_{t+h}^1 \sim T_{t+h|t}^1(\cdot | X_t^1, F_{t,1}^\theta(X_t^1, X_t^2)), x_{t+h}^2 \sim T_{t+h|t}^2(\cdot | X_t^2, F_{t,2}^\theta(X_t^1, X_t^2))} [f(x_{t+h}^1, x_{t+h}^2) - f(x_1, x_2)] \right] \tag{233}$$

We derive

$$\lim_{h \to 0} \frac{1}{h} \left[ \mathbb{E}_{x^1_{t+h} \sim T^1_{t+h|t}(\cdot | X^1_t, F^\theta_{t,1}(X^1_t, X^2_t)), x^2_{t+h} \sim T^2_{t+h|t}(\cdot | X^2_t, F^\theta_{t,2}(X^1_t, X^2_t))} [f(x^1_{t+h}, x^2_{t+h}) - f(x_1, x_2)] \right] \tag{234}$$

$$= \lim_{h \to 0} \frac{1}{h} \left[ \mathbb{E}_{x^1_{t+h} \sim T^1_{t+h|t}(\cdot | X^1_t, F^\theta_{t,1}(X^1_t, X^2_t)), x^2_{t+h} \sim T^2_{t+h|t}(\cdot | X^2_t, F^\theta_{t,2}(X^1_t, X^2_t))} [f(x^1_{t+h}, x^2_{t+h}) - f(x^1_{t+h}, x_2)] \right] \tag{235}$$

$$+ \lim_{h \to 0} \frac{1}{h} \left[ \mathbb{E}_{x^1_{t+h} \sim T^1_{t+h|t}(\cdot | X^1_t, F^\theta_{t,1}(X^1_t, X^2_t))} [f(x^1_{t+h}, x_2) - f(x_1, x_2)] \right] \tag{236}$$

$$= \lim_{h \to 0} \frac{1}{h} \left[ \mathbb{E}_{x^1_{t+h} \sim T^1_{t+h|t}(\cdot | X^1_t, F^\theta_{t,1}(X^1_t, X^2_t)), x^2_{t+h} \sim T^2_{t+h|t}(\cdot | X^2_t, F^\theta_{t,2}(X^1_t, X^2_t))} [f^{x^1_{t+h}}(x^2_{t+h}) - f^{x^1_{t+h}}(x_2)] \right] \tag{237}$$

$$+ \lim_{h \to 0} \frac{1}{h} \left[ \mathbb{E}_{x^1_{t+h} \sim T^1_{t+h|t}(\cdot | X^1_t, F^\theta_{t,1}(X^1_t, X^2_t))} [f^{x_2}(x^1_{t+h}) - f^{x_2}(x_1)] \right] \tag{238}$$

$$= \tilde{\mathcal{L}}^{x_1}_t f^{x_1}(x_1, x_2) + \bar{\mathcal{L}}_t f^{x_2}(x_1, x_2) \tag{239}$$

$$= \mathcal{L}_t f(x_1, x_2) \tag{240}$$

where we used eq. (199) and eq. (200) and the uniform continuity in both arguments $x^1_{t+h}, x^2_{t+h}$. The above derivation shows that the sampling procedure as defined in eq. (209) leads to the correct generator.

# D   KL-DIVERGENCE (ELBO) LOSSES FOR MARKOV PROCESSES

In principle, proposition 2 shows that any Bregman divergence allows us to train a Generator Matching model. However, while the minimum for all Bregman divergences is the same, different Bregman divergences trade-off errors differently. Further, some Bregman divergences can be derived via first principles and have theoretical properties that make it seem preferrable. Here, we consider an ELBO/KL-divergence loss and derive it for jump models, the model that we implement in sec. 9.

## D.1   PATH ELBO LOSS - GENERAL CASE

For completeness, we give here a heuristic derivation of the continuous-time ELBO loss. A similar derivation as below was already used by Sohl-Dickstein et al. (2015) to derive an ELBO loss to train a diffusion model (for discrete time steps). Let us be given a reference Markov process $(X_t)_{0 \leq t \leq 1}$ with transition kernel $k_{t+h|t}$ and a parameterized Markov process $(X^\theta_t)_{0 \leq t \leq 1}$ with transition kernel $k^\theta_{t+h|t}$. Then let us discretize the process in time and let's define grid points $t_i = i/n$ for $i = 0, 1 \ldots, n$ and $h = \frac{1}{n}$. Here, we assume that all considered probability measures are absolutely continuous with respect to a reference measure $\nu$. Let $p, p^\theta$ denote the joint marginals of all grid points, i.e.

$$p^\theta(x_1, x_{\frac{n-1}{n}}, \ldots, x_{\frac{1}{n}}, x_0) = p_0(x_0) \prod_{t=0,1/n,\ldots,(n-1)/n} k^\theta_{t+h|t}(x_{t+h} | x_t) \tag{241}$$

$$p(x_1, x_{\frac{n-1}{n}}, \ldots, x_{\frac{1}{n}}, x_0) = p_0(x_0) \prod_{t=0,1/n,\ldots,(n-1)/n} k_{t+h|t}(x_{t+h} | x_t) \tag{242}$$

Then we can bound the KL-divergence by using the data processing inequality and the chain rule for KL-divergence to get

$$D_{KL}(p_{\text{data}}(x_1)||p_1^\theta(x_1)) \tag{243}$$

$$\leq D_{KL}(p(x_1, x_{\frac{n-1}{n}}, \ldots, x_{\frac{1}{n}}, x_0)||p^\theta(x_1, x_{\frac{n-1}{n}}, \ldots, x_{\frac{1}{n}}, x_0)) \tag{244}$$

$$= D_{KL}(p(x_0)||p^\theta(x_0)) + \sum_{i=0}^{n-1} \mathbb{E}_{x_{t_i} \sim p_{t_i}}[D_{KL}(k_{t_{i+1}|t_i}(\cdot|x_{t_i})||k_{t_{i+1}|t_i}^\theta(\cdot|x_{t_i}))] \tag{245}$$

$$= 0 + \sum_{i=0}^{n-1}(t_{i+1} - t_i)\mathbb{E}_{x_{t_i} \sim p_{t_i}}\left[\frac{D_{KL}(k_{t_{i+1}|t_i}(\cdot|x_{t_i})||k_{t_{i+1}|t_i}^\theta(\cdot|x_{t_i}))}{t_{i+1} - t_i}\right] \tag{246}$$

$$\rightarrow \int_0^1 \int \int p_{\text{data}}(x_1)p_{t|x_1}(x_t|x_1)\frac{\partial}{\partial h}\left[D_{KL}(k_{t+h|t}(x_{t+h}|x_t)||k_{t+h|t}^\theta(x_{t+h}|x_t))\right]_{|h=0} dx_t dx_1 dt \tag{247}$$

$$= \mathbb{E}_{t \sim \text{Unif}, z \sim p_{\text{data}}, x_t \sim p_t(\cdot|z)}\left[\frac{\partial}{\partial h}\left[D_{KL}(k_{t+h|t}(x_{t+h}|x_t)||k_{t+h|t}^\theta(x_{t+h}|x_t))\right]_{|h=0}\right] \tag{248}$$

for $n \rightarrow \infty$. The above shows that

$$D_{KL}(p_{\text{data}}(x_1)||p_1^\theta(x_1)) \tag{249}$$

$$\leq \mathbb{E}_{t \sim \text{Unif}, z \sim p_{\text{data}}, x_t \sim p_t(\cdot|z)}\left[\frac{\partial}{\partial h}\left[D_{KL}(k_{t+h|t}(x_{t+h}|x_t)||k_{t+h|t}^\theta(x_{t+h}|x_t))\right]_{|h=0}\right] \tag{250}$$

### D.2 ELBO LOSS FOR CONTINUOUS-TIME MARKOV CHAIN (CTMC)

We work out an ELBO bound for the case of $S$ discrete using the path ELBO bound given via eq. (249). Let use assume that there are two CTMCs given with rate matrices $Q_t, Q_t^\theta$ and transition kernels $k_{t+h|t}, k_{t+h|t}^\theta$. Then, it holds that:

$$k_{t+h|t}(x_{t+h}|x_t) = \delta_{x_t}(x_{t+h}) + hQ_t(x_{t+h}|x_t) + o(h) \tag{251}$$

$$k_{t+h|t}^\theta(x_{t+h}|x_t) = \delta_{x_t}(x_{t+h}) + hQ_t^\theta(x_{t+h}|x_t) + o(h) \tag{252}$$

Further,

$$D_{KL}(k_{t+h|t}(x_{t+h}|x_t)||k_{t+h|t}^\theta(x_{t+h}|x_t)) \tag{253}$$

$$= \sum_{x_{t+h}} k_{t+h|t}(x_{t+h}|x_t) \log \frac{k_{t+h|t}(x_{t+h}|x_t)}{k_{t+h|t}^\theta(x_{t+h}|x_t)} dx_{t+h} \tag{254}$$

$$= \int [\delta_{x_t}(x_{t+h}) + hQ_t(x_{t+h}|x_t)] \log \frac{\delta_{x_t}(x_{t+h}) + hQ_t(x_{t+h}|x_t) + o(h)}{\delta_{x_t}(x_{t+h}) + hQ_t^\theta(x_{t+h}|x_t) + o(h)} dx_{t+h} \tag{255}$$

$$= (1 + hQ_t(x_t|x_t)) \log \frac{1 + hQ_t(x_t|x_t) + o(h)}{1 + hQ_t^\theta(x_t|x_t) + o(h)} \tag{256}$$

$$+ \sum_{x_{t+h} \neq x_t} hQ_t(x_{t+h}|x_t) \log \frac{hQ_t(x_{t+h}|x_t) + o(h)}{hQ_t^\theta(x_{t+h}|x_t) + o(h)} \tag{257}$$

Ignoring the $o(h)$-terms, we get:

$$D_{KL}(k_{t+h|t}(x_{t+h}|x_t)||k_{t+h|t}^\theta(x_{t+h}|x_t)) \tag{258}$$

$$=(1+hQ_t(x_t|x_t))\log\frac{1+hQ_t(x_t|x_t)}{1+hQ_t^\theta(x_t|x_t)} + \sum_{x_{t+h}\neq x_t} hQ_t(x_{t+h}|x_t)\log\frac{Q_t(x_{t+h}|x_t)}{Q_t^\theta(x_{t+h}|x_t)} \tag{259}$$

It holds that:

$$\frac{\partial}{\partial h}\log\frac{a+hy}{a+hx} = \frac{a+hx}{a+hy}\frac{(a+hx)y-(a+hy)x}{(a+hx)^2} \tag{260}$$

$$= \frac{1}{a+hy}\frac{ay+hxya-xa-hyxa}{a+hx} = \frac{a(y-x)}{(a+hy)(a+hx)} \tag{261}$$

$$\frac{\partial}{\partial h}\log\frac{a+hy}{a+hx}\Big|_{h=0} = \frac{y-x}{a} \tag{262}$$

And therefore,

$$\frac{\partial}{\partial h}D_{KL}(k_{t+h|t}(x_{t+h}|x_t)||k_{t+h|t}^\theta(x_{t+h}|x_t))_{|h=0} \tag{263}$$

$$= \left[Q_t(x_t|x_t)\log\frac{1+hQ_t(x_t|x_t)}{1+hQ_t^\theta(x_t|x_t)} + (1+hQ_t(x_t|x_t))\frac{Q_t(x_t|x_t)-Q_t^\theta(x_t|x_t)}{(1+hQ_t^\theta(x_t|x_t))(1+hQ_t(x_t|x_t))}\right]_{|h=0} \tag{264}$$

$$+ \sum_{x_{t+h}\neq x_t} Q_t(x_{t+h}|x_t)\log\frac{Q_t(x_{t+h}|x_t)}{Q_t^\theta(x_{t+h}|x_t)} \tag{265}$$

$$= Q_t(x_t|x_t) - Q_t^\theta(x_t|x_t) + \sum_{x_{t+h}\neq x_t} Q_t(x_{t+h}|x_t)\log\frac{Q_t(x_{t+h}|x_t)}{Q_t^\theta(x_{t+h}|x_t)} \tag{266}$$

$$= -Q_t^\theta(x_t|x_t) - \sum_{x_{t+h}\neq x_t} Q_t(x_{t+h}|x_t)\log Q_t^\theta(x_{t+h}|x_t) + C \tag{267}$$

$$= \sum_{x_{t+h}\neq x_t} Q_t^\theta(x_{t+h}|x_t) - Q_t(x_{t+h}|x_t)\log Q_t^\theta(x_{t+h}|x_t) + C \tag{268}$$

where the last step assumes that $Q_t$ has been normalized to $\sum_{y\neq x_t} Q_t(y|x_t) = -Q_t(x_t|x_t)$ and $C$ is a constant in $q$. The above gives us a simple way of training the $Q_t$-kernel. Plugging this into the Generator Matching loss (see proposition 2) for a conditional rate matrix $Q_t^z$, this gives us the total loss of:

$$D_{KL}(p_{\text{data}}(x_1)||p_1^\theta(x_1)) \tag{269}$$

$$\leq \mathbb{E}_{z\sim p_{\text{data}}, t\sim\text{Unif}_{[0,1]}, x_t\sim p_t(\cdot|z)}\left[\sum_{\tilde{x}\neq x_t} Q_t^\theta(\tilde{x}|x_t) - Q_t^z(\tilde{x}|x_t)\log Q_t^\theta(\tilde{x}|x_t)\right] + C \tag{270}$$

$$=: L(\theta) \tag{271}$$

The above loss was also found previously in the literature (Opper & Sanguinetti, 2007, equation (3)) as an ELBO bound of the KL-divergence of continuous-space jump processes.

Up to a constant in $\theta$, we can frame the above as a conditional Generator Matching loss with a Bregman divergence by defining the convex function $\phi$ on $\Omega = \mathbb{R}_{\geq 0}^{|S|-1}$ via

$$\phi : \Omega \to \mathbb{R}, \quad x \mapsto \sum_{i=1}^{|S|-1} x_i\log(x_i) - x_i \tag{272}$$

To see this, it holds that

$$\nabla\phi(x) = \log(x), \nabla^2\phi(x) = \mathrm{diag}(1/x) \geq 0 \tag{273}$$

which shows that $\phi$ is a convex function. Further, the corresponding Bregman divergence is given by

$$D(x, y) = \phi(x) - \phi(y) - \langle x - y, \nabla\phi(y) \rangle \tag{274}$$

$$= \sum_{i=1}^{|S|-1} -y_i \log(y_i) + x_i \log(x_i) - (x_i - y_i) - (x_i - y_i) \log(y_i) \tag{275}$$

$$= \sum_{i=1}^{|S|-1} y_i - x_i \log(y_i) + C \tag{276}$$

where $C$ is a constant in $y$. This recovers the above loss.

## E   Systematic Study of Probability Paths and Markov Models

In this section, we introduce and study several new Markov models for various probability paths. This results in a GM model for every combination of {mixture path, CondOT path} × {flow, diffusion, jump, Markov superposition} With this, we hope to provide a systematic study of the design space of the Markov models constructed via generator matching (GM), in particular ablating the effect of the probability path and the Markov model used.

### E.1   Overview and Empirical Study

**Deriving novel Markov models.**   In table 4, one can see an overview over various Markov models introduced in this section for the two most prominent probability paths (mixtures and the CondOT path). For a given probability path, the approach to find a new Markov model is as follows: (1) Define what class of Markov process is used and (2) Find a generator for this Markov process that solves the KFE. If a density is available, we use the adjoint KFE (see table 1). We illustrate this in several examples in this section. We note that throughout this section, we derive equations in $S = \mathbb{R}$ because due to proposition 4, we can easily extend these models to arbitary dimensions. We use standard Bregman divergences to train the model (see proposition 2).

**Empirical study of probability paths and Markov models.**   We implemented the model on a $2d$ checkerboard pattern distribution (Lou et al., 2023; Lipman et al., 2022). As one can see, the performance varies for different models depending of the probability path. From this experiments, we can extract several learnings:

1. **There is no single optimal probability path or Markov model:** In fig. 5 and fig. 6, we plot the marginals generated by the trained Markov models. The flow model performs better on the CondOT path and the jump performs better on the mixture path. **This shows that different probability paths are better or worse depending on the Markov model used and that there is no single best probability path.** There is an intuitive explanation for our results: the CondOT path geometrically moves probability mass, while the mixture path only up- and down-weighs mass but does not move the support of the distribution. In turn, the flow model "transports" probability mass, while the jump model "teleports" probability mass due to its discontinuous evolution. Therefore, it is intuitive that a jump model would perform better on a mixture path while a flow model performs better on a CondOT path. This is also in line with existing result in optimal transport theory connecting the CondOT path with a flow (Ambrosio, 2004).

2. **Discretization error:**   In fig. 7 and fig. 8, we plot the final distribution for various number of function evaluations (NFEs) allowing us to study the discretization errors. The flow model is highly prone to discretization errors in the mixture model, while showing low discretization error for the CondOT path. For a jump model, the opposite is true. This can be explained by the high Lipschitz constant of the learned vector field necessary to learn the transport of mass via a flow in the mixture path. In the context of density sampling, this is a known problem (see e.g. (Máté & Fleuret, 2023)). For the jump model, many small changes/movement (like in the CondOT path) lead to discretization error, while larger movements of probability path ("teleportation") are easier to handle. This explains the difference in sensitivity in the discretization observed in our experiments.

Therefore, the role of the probability path is to determine **determine the range of KFE solutions** that exist. This can, in fact, influence the performance heavily. However, note that there are many KFE solutions for the same probability paths - even for the same Markov class.

| Name | Mixture | CondOT |
|---|---|---|
| Formula | $p_t(\cdot\|z) = \kappa_t \delta_z + (1-\kappa_t)\text{Unif}_{[a_1,a_2]}$ | $p_t(x\|z) = \mathcal{N}(tz, (1-t)^2)$ |
| Flow | $u_t(x\|z) = \frac{\dot{\kappa}_t(a_2-a_1)}{(1-\kappa_t)}(1_{x \le z} - F_0(x))$ | $u_t(x\|z) = \frac{z-x}{1-t}$ |
| Diffusion | $\sigma_t^2(x\|z) = \frac{2\dot{\kappa}t(G_0(z)+[x-z]_+-G_0(x)))}{(1-\kappa_t)p_0(x)}$ | No KFE solution exists. |
| Jump | $\lambda_t(x\|z) = \frac{\dot{\kappa}_t}{1-\kappa_t}, J_t(dx'; x\|z) = \delta_z$ | $\lambda_t(x\|z) = \frac{[k_t(x\|z)]_+}{(1-t)^3}, \quad J_t(x\|z) \propto [-k_t(x\|z)]_+ p_t(x)$ |
| MS | Jump+Diffusion+Flow | Jump+Flow |

Table 4: Systematic combinations of probability paths and KFE solutions introduced in this section. All of these models are written as models on $S = \mathbb{R}$ but that can be extended to arbitary dimensions using proposition 4. A "pure" diffusion solution to the CondOT path is theoretically impossible to exist - meaning a solution with zero drift (we show this in app. E.6). Notation: $k_t(x|z) = x^2 - (t+1)xz - (1-t)^2 + tz^2$. MS: Markov superposition. The derivations for table 4 can be found in the remainder of this section.

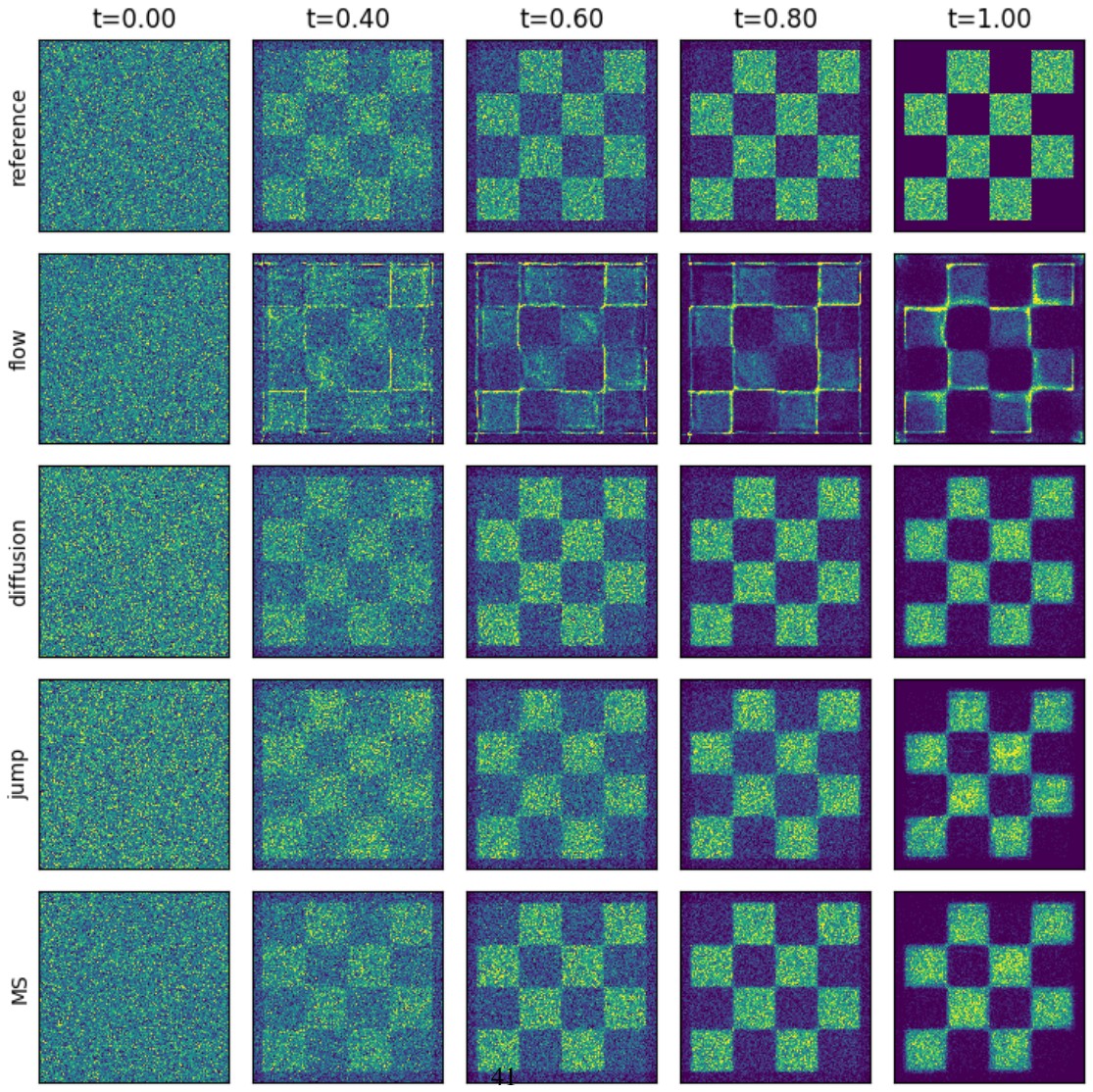

Figure 5: Illustration of Markov models in table 4 trained on a checkerboard pattern data distribution with a *mixture probability path*. As one can see, the flow model does not perform well on this probability path (due to the high Lipschitz constant), while the jump model performs very well because it can "teleport" mass due to its discontinuity.

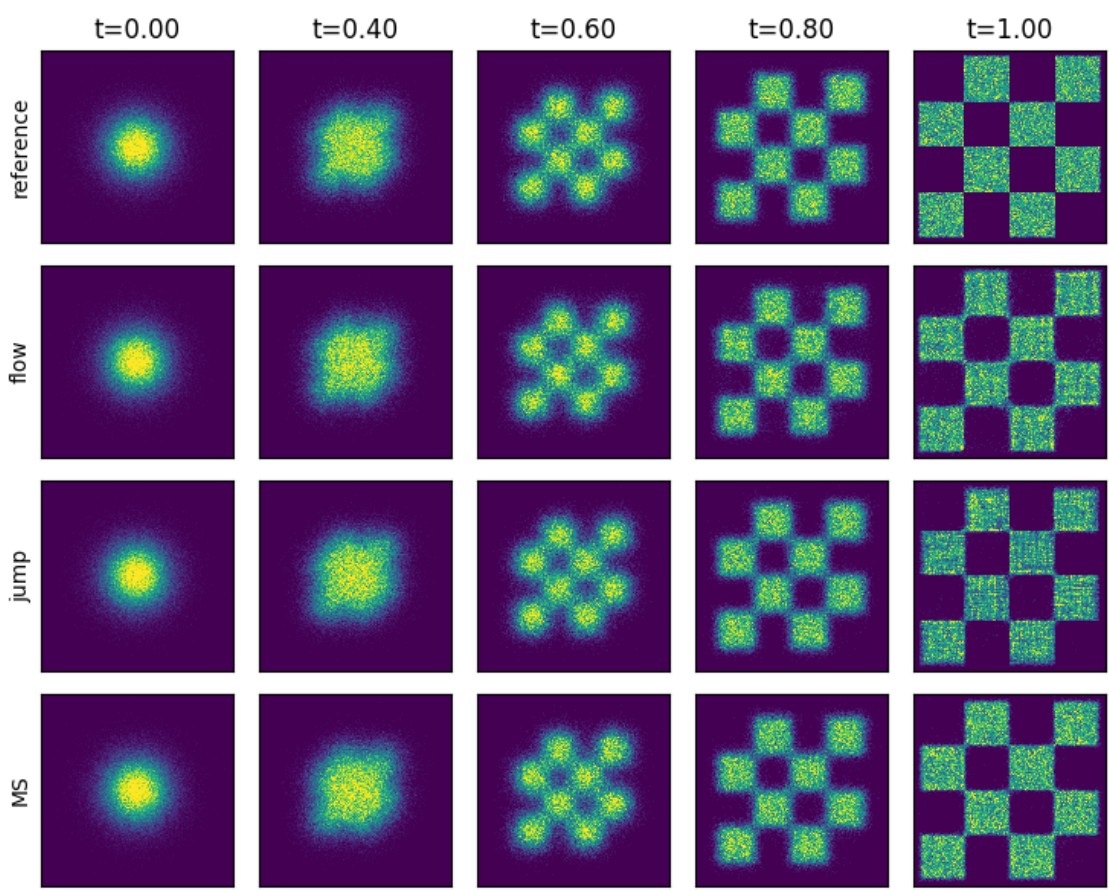

Figure 6: Illustration of Markov models in table 4 trained on a checkerboard pattern data distribution with a *CondOT probability path*. As one can see, the flow model performs better than the jump model on this probability path as suggested by optimal transport theory (Ambrosio, 2004), while the Markov superposition (jump+flow) seems to perform best. We validate these differences in our experiments on CIFAR-10 and ImageNet32 (see sec. 9).

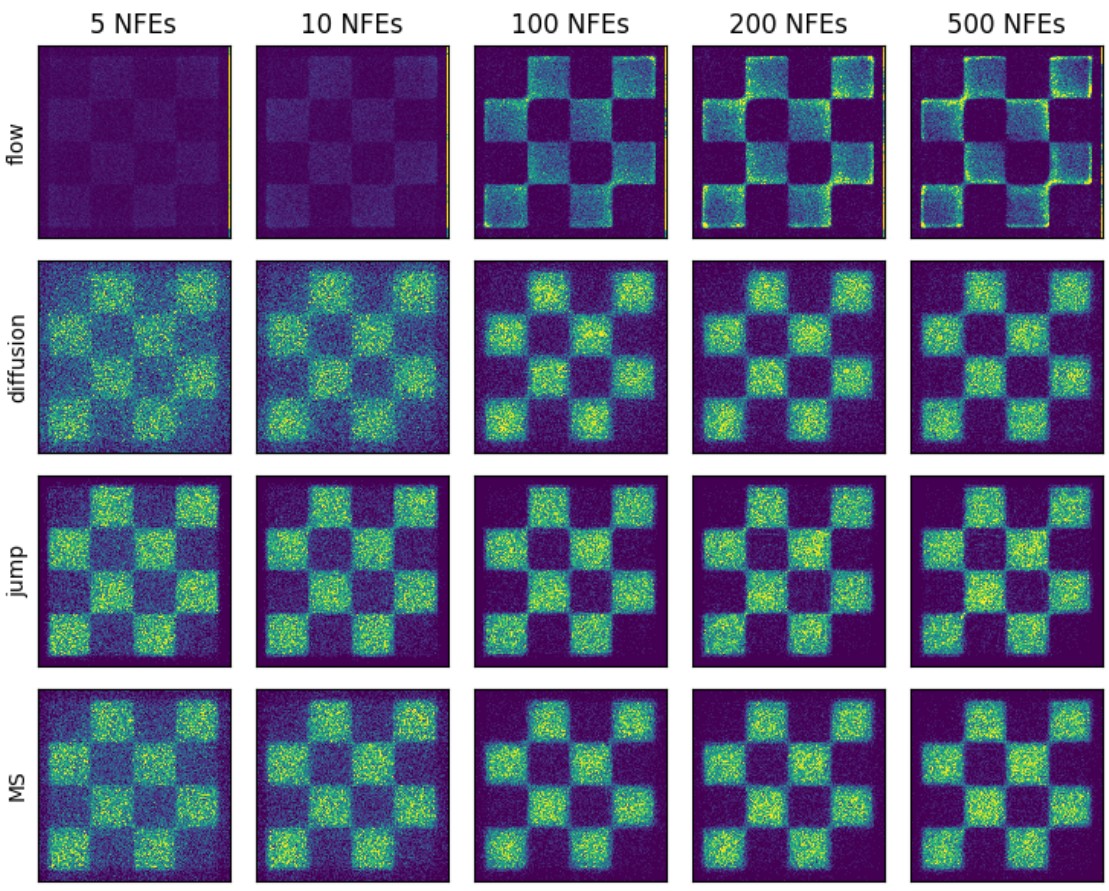

Figure 7: Results from sampling with a various number of NFEs for *mixture path* for various models. As one can see, the jump model is least prone to discretization error for this probability path.

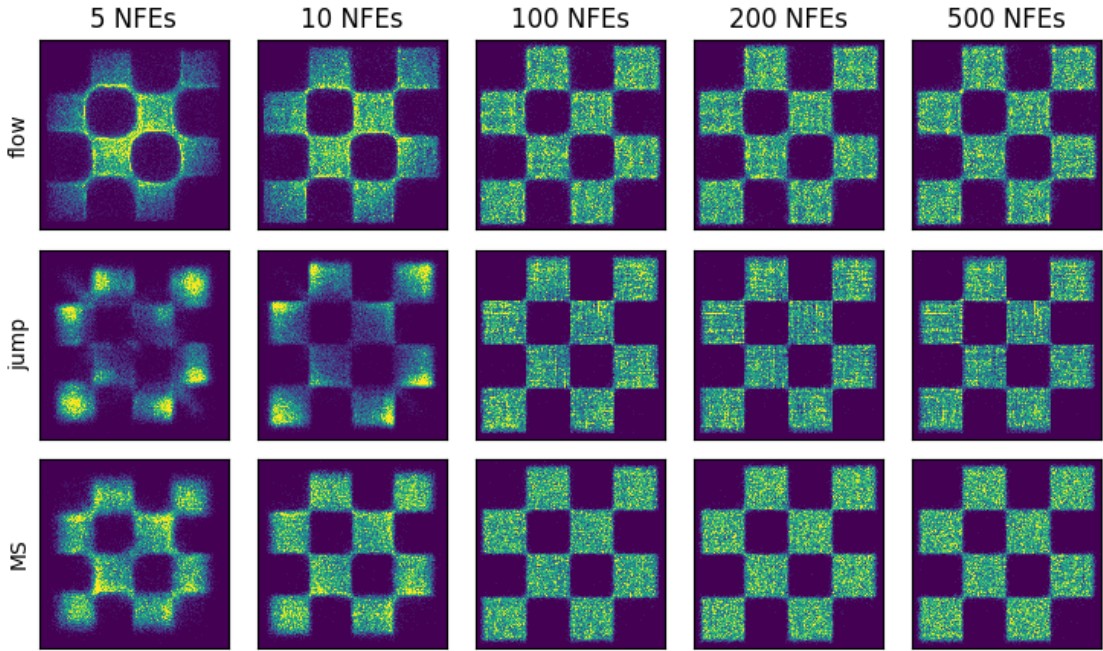

Figure 8: Results from sampling with a various number of NFEs for *CondOT probability path* for various models. As one can see, the flow model is least prone to discretization error for this probability path.

### E.2 DERIVATION OF FLOW MODEL FOR MIXTURE PATH

Let $p_0$ be an arbitrary distribution over $S = \mathbb{R}$. Let $\sigma_{\min} > 0$ a noise parameter. Let's consider a mixture probability path in 1d of the shape:

$$p_t(\cdot|z) = \kappa_t \mathcal{N}(z, \sigma_{\min}^2) + (1 - \kappa_t)p_0$$

Let $F_0$ be the cumulative distribution function of $p_0$ and $F_{0,\sigma}$ be the cumulative distribution function of $\mathcal{N}(x; 0, \sigma_{\min}^2)$. Then our goal is to find a vector field $u_t(x|z) \in \mathbb{R}$ that satisfies the continuity equation, i.e.

$$
\begin{aligned}
\frac{\partial}{\partial t} p_t(x|z) =& \dot{\kappa}_t(\mathcal{N}(x; z, \sigma_{\min}^2) - p_0(x)) \\
=& -\frac{\partial}{\partial x}(\dot{\kappa}_t(F_0(x) - F_{z,\sigma_{\min}}(x))) \\
=& -\frac{\partial}{\partial x}(p_t(x|z)\frac{\dot{\kappa}_t(F_0(x) - F_{z,\sigma_{\min}}(x))}{p_t(x|z)}) \\
=& -\frac{\partial}{\partial x}(p_t(x|z)\underbrace{\frac{\dot{\kappa}_t(F_0(x) - F_{z,\sigma_{\min}}(x))}{\kappa_t\mathcal{N}(x; z, \sigma_{\min}^2) + (1 - \kappa_t)p_0(x)}}_{=:u_t(x|z)}) \\
=& -\frac{\partial}{\partial x}(p_t(x|z)u_t(x|z))
\end{aligned}
$$

Therefore, we can see that $u_t(\cdot|z)$ generates the above probability path.

**Limit for $\sigma_{\min} \to 0$.** Note that for $\sigma_{\min} \to 0$, the conditional vector field becomes:

$$u_t(x|x_1) = \frac{\dot{\kappa}_t}{(1 - \kappa_t)p_0(x)} \cdot \begin{cases} 0 & \text{if } x = x_1 \\ F_0(x) & \text{if } x < x_1 \\ F_0(x) - 1 & \text{if } x > x_1 \end{cases}$$

**Uniform prior.** Let $p_0 = \mathrm{Unif}_{[a_1, a_2]}$. Then the vector field would become:

$$u_t(x|z) = \frac{\dot{\kappa}_t}{(1 - \kappa_t)\frac{1}{a_2 - a_1}} \frac{(a_2 - a_1)1_{x \leq z} - x + a_1}{a_2 - a_1}$$

$$= \frac{\dot{\kappa}_t}{(1 - \kappa_t)}((a_2 - a_1)1_{x \leq z} - x + a_1)$$

Note that the above vector field is not continuous. In particular, its Lipschitz-constant is infinite. However, it still defines a valid flow.

**Remark - Cross-entropy training.** Beyond training a flow model, we could also train a binary classifier model $p_\theta(x) \approx \mathbb{P}[x_t > x_1 | x_t = x]$. The marginal vector field is then given via:

$$u_t(x) = \frac{\dot{\kappa}_t}{(1 - \kappa_t)p_0(x)} \cdot (F_{0,\sigma}(x) - p_\theta(x))$$

Learning $p_\theta(x_t)$ can be done via cross-entropy for every dimension, a very different Bregman divergence and generator parameterization (see app. A.6). We do not consider this possibility in this work but mention it an interesting avenue for future work.

### E.3 DERIVATION OF DIFFUSION SOLUTION TO MIXTURE PATH

In this section, we would like to find a solution based on a *forward* diffusion process that solves for the mixture probability path in $\mathbb{R}$ given by:

$$p_t(x|z) = (1 - \kappa_t) \cdot p_0(dx) + \kappa_t \cdot \mathcal{N}(z, \sigma_{\min}^2) \quad \Leftrightarrow \quad x_t \sim \begin{cases} \sim \mathcal{N}(z, \sigma_{\min}^2) & \text{with prob } \kappa_t \\ \sim p_0 & \text{with prob } (1 - \kappa_t) \end{cases} \quad (277)$$

where $\sigma_{\min} > 0$ is a small value (we will later $\sigma_{\min} \to 0$). Specifically, we search for KFE solutions given by an SDE

$$dX_t = \sigma(X_t, t)dW_t + dL_t \quad (278)$$

where $\sigma(X_t, t)$ is a diffusion coefficient controlling the amount of infinitesimal noise we add and $dL_t$ describes a reflection process. Let $G_0(x) = \int_{-\infty}^{x} \int_{-\infty}^{y} p_0(w)dwdy$ and $G_{z,\sigma_{\min}}(x) = \int_{-\infty}^{x} \int_{-\infty}^{y} \mathcal{N}(w; z, \sigma_{\min}^2)dwdy$. Then:

$$\frac{\partial^2}{\partial^2 x} G_{x_1, \sigma_{\min}}(x) = \mathcal{N}(x; x_1, \sigma_{\min}^2), \quad \frac{\partial^2}{\partial^2 x} G_0(x) = p_0(x) \quad (279)$$

Therefore, for any $a_t, b_t \in \mathbb{R}$ we get

$$\frac{\partial}{\partial t} p_t(x|z) = \dot{\kappa}_t(\mathcal{N}(x; z, \sigma_{\min}^2 I_d) - p_0(x)) \tag{280}$$

$$= \frac{\partial^2}{\partial^2 x}(\dot{\kappa}_t(a_t + b_t x + G_{z,\sigma_{\min}}(x) - G_0(x))) \tag{281}$$

$$= \frac{\partial^2}{\partial^2 x}(p_t(x|z) \frac{\dot{\kappa}_t(a_t + b_t x + G_{z,\sigma_{\min}}(x) - G_{0,\sigma}(x))}{p_t(x|z)}) \tag{282}$$

$$= \frac{1}{2}\frac{\partial^2}{\partial^2 x}(p_t(x|z) \underbrace{\frac{2\dot{\kappa}_t(a_t + b_t x + G_{z,\sigma_{\min}}(x) - G_0(x))}{\kappa_t \mathcal{N}(x; z, \sigma_{\min}^2) + (1 - \kappa_t)p_0(x)}}_{=:\tilde{\sigma}_t^2(x|z)}) \tag{283}$$

$$= \frac{1}{2}\frac{\partial^2}{\partial^2 x}(p_t(x|z)\tilde{\sigma}_t^2(x|z)) \tag{284}$$

Therefore, we can see that $\tilde{\sigma}_t^2(x|z)$ satisfies the Fokker-Planck equation. However, $\tilde{\sigma}^2$ can be negative and therefore it is not a valid diffusion coefficient in general. We have to pick $a_t, b_t$ such that $\tilde{\sigma}_t^2$ is non-negative.

**Choice of $a_t, b_t$.** Specifically, define $a_t = c_t + \frac{1}{2}z$ and $b_t = 0$ for a value $c_t$ that we will define later. Then we get that:

$$\tilde{\sigma}_t^2(x|z) = \frac{2\dot{\kappa}_t(a_t + b_t x + G_{z,\sigma_{\min}}(x) - G_0(x))}{\kappa_t \mathcal{N}(x; z, \sigma_{\min}^2) + (1 - \kappa_t)p_0(x)} = \frac{2\dot{\kappa}_t(c_t + \frac{1}{2}x + \frac{1}{2}(z - x) + G_{z,\sigma_{\min}}(x) - G_0(x))}{\kappa_t \mathcal{N}(x; z, \sigma_{\min}^2) + (1 - \kappa_t)p_0(x)} \tag{285}$$

The value $c_t$ is chosen such that $\tilde{\sigma}_t^2(x|z)$ is non-negative (we will simply define it as the minimum of the residual). For $\sigma_{\min} \to 0$, it holds that $G_{z,\sigma_{\min}}(x) \to [x - z]_+$ and we have:

$$\tilde{\sigma}_t^2(x|z) = \frac{2\dot{\kappa}_t(c_t + \frac{1}{2}x + \frac{1}{2}(z - x) + G_{z,\sigma_{\min}}(x) - G_0(x))}{\kappa_t \mathcal{N}(x; z, \sigma_{\min}^2) + (1 - \kappa_t)p_0(x)} \tag{286}$$

$$\to \frac{2\dot{\kappa}_t(c_t + \frac{1}{2}x + \frac{1}{2}(z - x) + [x - z]_+ - G_0(x)))}{(1 - \kappa_t)p_0(x)} \tag{287}$$

$$= \frac{2\dot{\kappa}_t(c_t + \frac{1}{2}x + \frac{1}{2}|x - z| - G_0(x)))}{(1 - \kappa_t)p_0(x)} \tag{288}$$

We can define

$$c_t := -\min_{x \in \mathbb{R}} g(x), \quad g(x) = \frac{1}{2}x + \frac{1}{2}|x - z| - G_0(x) \tag{289}$$

We know that

$$g'(x) = 1 - F_0(x) < 0, \quad (x \geq z) \tag{290}$$

$$g'(x) = -F_0(x) < 0, \quad (x \leq z) \tag{291}$$

The above implies that the minimum of $g(x)$ is obtained at $z$ and it holds that:

$$c_t = G_0(z) - \frac{1}{2}z \tag{292}$$

This value can be computed numerically. The final function we get is:

$$\tilde{\sigma}_t^2(x|z) = \frac{2\dot{\kappa}_t(G_0(z) - \frac{1}{2}z + \frac{1}{2}x + \frac{1}{2}|x - z| - G_0(x)))}{(1 - \kappa_t)p_0(x)} = \frac{2\dot{\kappa}_t(G_0(z) + [x - z]_+ - G_0(x)))}{(1 - \kappa_t)p_0(x)} \tag{293}$$

If the above prior $p_0$ has no boundaries (i.e. support $\mathbb{R}$) as for a Gaussian, then we are done here. For a prior with compact support, we need to consider reflections.

**Uniform prior.**   Let's consider a uniform prior $p_0 = \text{Unif}_{[a_1, a_2]}$. Then the equation becomes

$$\tilde{\sigma}_t^2(x|z) = (a_2 - a_1) \frac{2\dot{\kappa}_t \left( \frac{1}{2} \frac{(z-a_1)^2}{a_2-a_1} + [x - z]_+ - \frac{1}{2} \frac{(x-a_1)^2}{a_2-a_1} \right)}{(1 - \kappa_t)} \tag{294}$$

The above equation has non-zero boundaries:

$$\tilde{\sigma}_t^2(a_1|z) = \frac{\dot{\kappa}_t(z^2 - a_1^2 - 2a_1[z - a_1])}{(a_2 - a_1)(1 - \kappa_t)} = \frac{\dot{\kappa}_t(z^2 + a_1^2 - 2a_1 z)}{(a_2 - a_1)(1 - \kappa_t)} = \frac{\dot{\kappa}_t(z - a_1)^2}{(a_2 - a_1)(1 - \kappa_t)} \tag{295}$$

$$\tilde{\sigma}_t^2(a_2|z) = \frac{\dot{\kappa}_t(z^2 - a_2^2 + 2a_2[a_2 - z])}{(a_2 - a_1)(1 - \kappa_t)} = \frac{\dot{\kappa}_t(z^2 + a_2^2 - 2a_2 z)}{(a_2 - a_1)(1 - \kappa_t)} = \frac{\dot{\kappa}_t(z - a_2)^2}{(a_2 - a_1)(1 - \kappa_t)} \tag{296}$$

Therefore, we need to consider a **reflected SDE** of the form

$$dX_t = \sigma_t(X_t)dW_t + L_t$$

where $L_t$ describes a reflection on the boundaries $[a_1, a_2]$. We refer to (Pilipenko, 2014) for a rigorous definition on reflected SDEs. Here, we simply note that one can imulate the reflected SDE up to $o(h)$-approximation error with

$$X_{t+h} = R_{a_1, a_2}(X_t + \sqrt{h}\sigma_t(X_t))$$

where $R_{a_1, a_2}$ describes the reflection operator along the boundaries $[a_1, a_2]$. Beyond satisfying the Fokker-Planck equation, it must then also satisfy the Neumann boundary condition that (Pilipenko, 2014; Lou & Ermon, 2023):

$$\frac{\partial}{\partial x}[\tilde{\sigma}_t^2(x|z)p_t(x|z)]_{|x=a_1} = 0 \tag{297}$$

$$\frac{\partial}{\partial x}[\tilde{\sigma}_t^2(x|z)p_t(x|z)]_{|x=a_2} = 0 \tag{298}$$

Note that here, $p_t(x|z)$ is a constant along the boundary (because the only place where it is not constant is around $z$ for $\sigma_{\min} \to 0$). Therefore, the above reads as

$$\frac{\partial}{\partial x}[\tilde{\sigma}_t^2(x|z)]_{|x=a_1} = 0 \tag{299}$$

$$\frac{\partial}{\partial x}[\tilde{\sigma}_t^2(x|z)]_{|x=a_2} = 0 \tag{300}$$

That this is fulfilled can be easily seen. Therefore, in total, we have proven that a **reflected Brownian motion with a uniform initial distribution and $\tilde{\sigma}_t^2$ as defined as in eq. (294) generates the conditional mixture path.**

### E.4   DERIVATION OF JUMP MODEL FOR MIXTURE PATH

Let us consider again the mixture probability path given by:

$$p_t(\cdot|z) = \kappa_t \delta_z + (1 - \kappa_t)p_0$$

We show here that the KFE is satisfied for a jump model with

$$Q_t(dx'; x|z) = \lambda_t(x|z)J_t(dx'; x|z), \quad \lambda_t(x|z) = \frac{\dot{\kappa}_t}{1 - \kappa_t}, \quad J_t(dx'; x|z) = \delta_z(dx')$$

To show this, we show that the above jump process fulfils the KFE. Using the form of the generator $\mathcal{L}_t$ derived in app. A.5.3, we derive:

$$
\begin{aligned}
&= \mathbb{E}_{x \sim p_t(\cdot|z)}[\mathcal{L}_t f(x)] \\
&= \mathbb{E}_{x \sim p_t(\cdot|z)}[\lambda_t(x|z)\mathbb{E}_{x' \sim J_t(dx';x|z)}[f(x') - f(x)]] \\
&= \frac{\dot{\kappa}_t}{1 - \kappa_t}\mathbb{E}_{x \sim p_t(\cdot|z)}[(f(z) - f(x))] \\
&= \frac{\dot{\kappa}_t}{1 - \kappa_t}[f(z) - \mathbb{E}_{x \sim p_t(x)}[f(x)]] \\
&= \frac{\dot{\kappa}_t}{1 - \kappa_t}[f(z) - [(1 - \kappa_t)\mathbb{E}_{x \sim p_0}[f(x)] + \kappa_t f(z)]] \\
&= \dot{\kappa}_t f(z) - \dot{\kappa}_t\mathbb{E}_{x \sim p_0}[f(x)] \\
&= \dot{\kappa}_t f(z) - \dot{\kappa}_t\mathbb{E}_{x \sim p_0}[f(x)] \\
&= \partial_t [\kappa_t f(z) + (1 - \kappa_t)\mathbb{E}_{x \sim p_0}[f(x)]] \\
&= \partial_t\mathbb{E}_{x \sim p_t(x)}[f(x)] \\
&= \partial_t \langle p_t(\cdot|z), f \rangle
\end{aligned}
$$

i.e. we see that the process fulfils the KFE. Therefore, we have established jump model.

### E.5 DERIVATION OF FLOW SOLUTION TO CONDOT PATH

Next, we consider the CondOT probability path $p_t(\cdot|z) = \mathcal{N}(tz, (1-t)^2)$. We consider a Markov process defined via a flow determined by the vector field

$$
u_t(x|z) = \frac{z - x}{1 - t}
$$

as already introduced in (Lipman et al., 2022). For completeness, we show that this generates the probability path (in (Lipman et al., 2022), this is done more generally for Gaussian probability paths). Specifically, one

has to show that it fulfils the continuity equation (the adjoint KFE for flows, see table 1), i.e.

$$\frac{\partial}{\partial t}p_t(x|z)$$

$$=\frac{\partial}{\partial t}\left[\frac{1}{\sqrt{2\pi(1-t)^2}}\exp\left(-\frac{(x-tz)^2}{2(1-t)^2}\right)\right]$$

$$=\frac{\partial}{\partial t}\left[\frac{1}{\sqrt{2\pi(1-t)^2}}\right]\exp\left(-\frac{(x-tz)^2}{2(1-t)^2}\right)+\frac{1}{\sqrt{2\pi(1-t)^2}}\frac{\partial}{\partial t}\left[\exp\left(-\frac{(x-tz)^2}{2(1-t)^2}\right)\right]$$

$$=\frac{\partial}{\partial t}[(1-t)^{-1}](2\pi)^{-1/2}\exp\left(-\frac{(x-tz)^2}{2(1-t)^2}\right)-\mathcal{N}(x;tz,(1-t)^2)\frac{\partial}{\partial t}\left[\frac{(x-tz)^2}{2(1-t)^2}\right]$$

$$=(1-t)^{-2}(2\pi)^{-1/2}\frac{\mathcal{N}(x;tz,(1-t)^2)}{(2\pi(1-t)^2)^{-1/2}}-\mathcal{N}(x;tz,(1-t)^2)\frac{\partial}{\partial t}\left[\frac{(x-tz)^2}{2(1-t)^2}\right]$$

$$=(1-t)^{-2}\frac{\mathcal{N}(x;tz,(1-t)^2)}{(1-t)^{-1}}-\mathcal{N}(x;tz,(1-t)^2)\frac{\partial}{\partial t}\left[\frac{(x-tz)^2}{2(1-t)^2}\right]$$

$$=(1-t)^{-1}\mathcal{N}(x;tz,(1-t)^2)-\mathcal{N}(x;tz,(1-t)^2)\frac{\partial}{\partial t}\left[\frac{(x-tz)^2}{2(1-t)^2}\right]$$

$$=\mathcal{N}(x;tz,(1-t)^2)\left[(1-t)^{-1}-\frac{\partial}{\partial t}\left[\frac{(x-tz)^2}{2(1-t)^2}\right]\right]$$

$$=\mathcal{N}(x;tz,(1-t)^2)\left[(1-t)^{-1}-\frac{2(1-t)^2 2(x-tz)(-z)-(x-tz)^2 4(1-t)(-1)}{4(1-t)^4}\right]$$

$$=\mathcal{N}(x;tz,(1-t)^2)\left[(1-t)^{-1}-\frac{-(1-t)(x-tz)z+(x-tz)^2}{(1-t)^3}\right]$$

$$=\mathcal{N}(x;tz,(1-t)^2)\left[(1-t)^{-1}-\frac{(x-tz)}{(1-t)^3}(-(1-t)z+(x-tz))\right]$$

$$=\mathcal{N}(x;tz,(1-t)^2)\left[(1-t)^{-1}-\frac{(x-tz)}{(1-t)^3}(x-z)\right]$$

$$=\left[\frac{1}{1-t}p_t(x|z)+\frac{z-x}{1-t}p_t(x|z)\frac{x-tz}{(1-t)^2}\right]$$

$$=[-\partial_x u_t(x|z)p_t(x|z)-u_t(x|z)\partial_x p_t(x|z)]$$

$$=-\partial_x[u_t(x|z)p_t(x|z)]$$

where derive the last identity in app. E.7. This proves that $u_t(x|z)$ satisfies the continuity equation. Therefore, the corresponding flow generates the CondOT probability path.

## E.6 DERIVATION OF DIFFUSION SOLUTION TO CONDOT PATH

In this section, we show that there is no KFE solution to the CondOT path given by a Markov process that is "purely" defined via a diffusion process, i.e. via an SDE $dX_t = \sigma_t(X_t)dW_t$ with zero drift. Using the

derivation of $\partial_t p_t(x|z)$ in the previous paragraph, we can compute:

$$
\begin{aligned}
\frac{\partial}{\partial t} p_t(x|z) =& \mathcal{N}(x; tz, (1-t)^2) \left[ (1-t)^{-1} - \frac{(x-tz)}{(1-t)^3}(x-z) \right] \quad \text{(See previous section)} \\
=& \frac{\partial^2}{\partial^2 x} \left[ a_t + b_t x + \int\limits_{-\infty}^{x} \int\limits_{-\infty}^{y} \mathcal{N}(z; tz, (1-t)^2) \left[ (1-t)^{-1} - \frac{(z-tz)}{(1-t)^3}(z-z) \right] dzdy \right] \\
=& \frac{\partial^2}{\partial^2 x} \left[ a_t + b_t x + \int\limits_{-\infty}^{x} \int\limits_{-\infty}^{y-tz} \mathcal{N}(z; 0, (1-t)^2) \left[ (1-t)^{-1} - \frac{z}{(1-t)^3}(z+tz-z) \right] dzdy \right] \\
=& \frac{\partial^2}{\partial^2 x} \left[ a_t + b_t x + \int\limits_{-\infty}^{x} \int\limits_{-\infty}^{\frac{y-tz}{1-t}} (1-t)\mathcal{N}(z; 0, 1) \left[ (1-t)^{-1} - \frac{z}{(1-t)^3}(z+tz-z) \right] dzdy \right] \\
=& \frac{\partial^2}{\partial^2 x} \left[ a_t + b_t x + \int\limits_{-\infty}^{x} \int\limits_{-\infty}^{\frac{y-tz}{1-t}} \mathcal{N}(z; 0, 1) \left[ 1 - \frac{z^2 + z(t-1)z}{(1-t)^2} \right] dzdy \right] \\
=& \frac{\partial^2}{\partial^2 x} \left[ a_t + b_t x + \int\limits_{-\infty}^{\frac{x-tz}{(1-t)}} (1-t) \int\limits_{-\infty}^{y} \mathcal{N}(z; 0, 1) \left[ 1 - \frac{z^2 + z(t-1)z}{(1-t)^2} \right] dzdy \right] \\
=& \frac{\partial^2}{\partial^2 x} \left[ a_t + b_t x + (1-t)G\left(\frac{x-tz}{1-t}\right) - \frac{1}{1-t}K\left(\frac{x-tz}{1-t}\right) + zL\left(\frac{x-tz}{1-t}\right) \right] \\
=& \frac{\partial^2}{\partial^2 x} \left[ \mathcal{N}(x; tz, (1-t)^2) \frac{\tilde{a}_t + \tilde{b}_t \frac{x-tz}{1-t} + (1-t)G\left(\frac{x-tz}{1-t}\right) - \frac{1}{1-t}K\left(\frac{x-tz}{1-t}\right) + zL\left(\frac{x-tz}{1-t}\right)}{\mathcal{N}(x; tz, (1-t)^2)} \right] \\
=& \frac{\partial^2}{\partial^2 x} \left[ \mathcal{N}(x; tz, (1-t)^2) \underbrace{\frac{\tilde{a}_t + \tilde{b}_t \phi(x) + (1-t)G\left(\phi(x)\right) - \frac{1}{1-t}K\left(\phi(x)\right) + zL\left(\phi(x)\right)}{\mathcal{N}(x; tz, (1-t)^2)}}_{=:\sigma_t^2(\tilde{x})} \right]
\end{aligned}
$$

where $\phi(x) = (x-tz)/(1-t)$ and $a_t, b_t, \tilde{a}_t, \tilde{b}_t$ are arbitrary constants (in $x$) - this characterizes the space of solutions to the Fokker-Planck equation fully. However, these constants have to be chosen such that $\sigma$ is

non-negative, which is - as we will show now - impossible. Specifically,

$$G(x) = \int_{-\infty}^{x} \int_{-\infty}^{y} \mathcal{N}(z; 0, 1) dz dy$$

$$L(x) = \int_{-\infty}^{x} \int_{-\infty}^{y} z \mathcal{N}(z; 0, 1) dz dy$$

$$K(x) = \int_{-\infty}^{x} \int_{-\infty}^{y} z^2 \mathcal{N}(z; 0, 1) dz dy$$

For very large $y$, it holds that

$$\int_{-\infty}^{y} \mathcal{N}(z; 0, 1) dz \approx 1 \quad \Rightarrow \quad \lim_{x \to \infty} \frac{G(x)}{x} = 1, \lim_{x \to -\infty} G(x) = 0$$

$$\int_{-\infty}^{y} z \mathcal{N}(z; 0, 1) dz \approx 0 \quad \Rightarrow \quad \lim_{x \to \infty} L(x) = 0, \lim_{x \to -\infty} L(x) = 0$$

$$\int_{-\infty}^{y} z^2 \mathcal{N}(z; 0, 1) dz \approx 1 \quad \Rightarrow \quad \lim_{x \to \infty} \frac{K(x)}{x} = 1, \lim_{x \to -\infty} K(x) = 0$$

Therefore, the asymptotic limit of the numerator is

$$\lim_{x \to \infty} \frac{\tilde{a}_t + \tilde{b}_t \phi(x) + (1-t) G(\phi(x)) - \frac{1}{1-t} K(\phi(x)) + z L(\phi(x))}{x}$$

$$= \lim_{x \to \infty} \frac{\tilde{a}_t + \tilde{b}_t \phi(x) + (1-t) G(\phi(x)) - \frac{1}{1-t} K(\phi(x)) + z L(\phi(x))}{\phi(x)} \frac{\phi(x)}{x}$$

$$= (\tilde{b}_t + (1-t) - \frac{1}{1-t}) \frac{1}{(1-t)}$$

$$= \tilde{b}_t + 1 - \frac{1}{(1-t)^2}$$

$$\lim_{x \to -\infty} \frac{\tilde{a}_t + \tilde{b}_t \phi(x) + (1-t) G(\phi(x)) - \frac{1}{1-t} K(\phi(x)) + z L(\phi(x))}{x}$$

$$= \lim_{x \to -\infty} \frac{\tilde{a}_t + \tilde{b}_t \phi(x) + (1-t) G(\phi(x)) - \frac{1}{1-t} K(\phi(x)) + z L(\phi(x))}{\phi(x)} \frac{\phi(x)}{x}$$

$$= (\tilde{b}_t + 0 + 0 + 0) \frac{-1}{1-t}$$

$$= -\frac{\tilde{b}_t}{1-t}$$

The above poses a problem: we cannot choose $\tilde{b}_t$ such that the asymptotics are both $\geq 0$ for $x \to \pm\infty$. The reason for that is that $1/(1-t)^2 > 1$ for all $0 < t < 1$. Hence, the above cannot have a solution with non-negative $z$. This shows that there is no "pure diffusion" solution to the CondOT path.

### E.7 DERIVATION OF JUMP SOLUTION TO CONDOT PATH

The CondOT probability path in $1d$ with a Gaussian prior is given via:

$$p_t(\cdot|z) = \mathcal{N}(tz, (1-t)^2) \tag{301}$$

for $z \in \mathbb{R}, 0 \le t \le 1$. With a slight abuse of notation, we write $p_t(x|z)$ for its density. The jump continuity equation (see app. A.5.3 for derivation) describes the adjoint KFE in this case given via

$$\frac{\partial}{\partial t}p_t(x|z) = \int Q_t(x;y)p_t(y|z) - Q_t(y;x)p_t(x|z)dy \tag{302}$$

where we omit the dependency of $Q_t$ on $z$ for conciseness and readability. Setting $\lambda_t^z(x) = \int Q_t(y;x)dy \ge 0$ and $J_t(y;x) = Q_t(y;x)/\lambda_t(x)$ (similarly, dropping dependency on $z$ for readability), we get

$$\frac{\partial}{\partial t}p_t(x|z) = \int \lambda_t(y)J_t(x;y)p_t(y|z)dy - \lambda_t(x)p_t(x|z) \tag{303}$$

**Time-derivative $\frac{\partial}{\partial t}p_t$.** Let's compute the left-hand side first:

$$\frac{\partial}{\partial t}p_t(x|z) \tag{304}$$

$$=\frac{\partial}{\partial t}\left[\frac{1}{\sqrt{2\pi(1-t)^2}}\exp\left(-\frac{(x-tz)^2}{2(1-t)^2}\right)\right] \tag{305}$$

$$=\frac{\partial}{\partial t}\left[\frac{1}{\sqrt{2\pi(1-t)^2}}\right]\exp\left(-\frac{(x-tz)^2}{2(1-t)^2}\right) + \frac{1}{\sqrt{2\pi(1-t)^2}}\frac{\partial}{\partial t}\left[\exp\left(-\frac{(x-tz)^2}{2(1-t)^2}\right)\right] \tag{306}$$

$$=\frac{\partial}{\partial t}[(1-t)^{-1}](2\pi)^{-1/2}\exp\left(-\frac{(x-tz)^2}{2(1-t)^2}\right) - \mathcal{N}(x;tz,(1-t)^2)\frac{\partial}{\partial t}\left[\frac{(x-tz)^2}{2(1-t)^2}\right] \tag{307}$$

$$=(1-t)^{-2}(2\pi)^{-1/2}\frac{\mathcal{N}(x;tz,(1-t)^2)}{(2\pi(1-t)^2)^{-1/2}} - \mathcal{N}(x;tz,(1-t)^2)\frac{\partial}{\partial t}\left[\frac{(x-tz)^2}{2(1-t)^2}\right] \tag{308}$$

$$=(1-t)^{-2}\frac{\mathcal{N}(x;tz,(1-t)^2)}{(1-t)^{-1}} - \mathcal{N}(x;tz,(1-t)^2)\frac{\partial}{\partial t}\left[\frac{(x-tz)^2}{2(1-t)^2}\right] \tag{309}$$

$$=(1-t)^{-1}\mathcal{N}(x;tz,(1-t)^2) - \mathcal{N}(x;tz,(1-t)^2)\frac{\partial}{\partial t}\left[\frac{(x-tz)^2}{2(1-t)^2}\right] \tag{310}$$

$$=\mathcal{N}(x;tz,(1-t)^2)\left[(1-t)^{-1} - \frac{\partial}{\partial t}\left[\frac{(x-tz)^2}{2(1-t)^2}\right]\right] \tag{311}$$

$$=\mathcal{N}(x;tz,(1-t)^2)\left[(1-t)^{-1} - \frac{2(1-t)^2 2(x-tz)(-z) - (x-tz)^2 4(1-t)(-1)}{4(1-t)^4}\right] \tag{312}$$

$$=\mathcal{N}(x;tz,(1-t)^2)\left[(1-t)^{-1} - \frac{-(1-t)(x-tz)z + (x-tz)^2}{(1-t)^3}\right] \tag{313}$$

$$=\mathcal{N}(x;tz,(1-t)^2)\left[(1-t)^{-1} - \frac{(x-tz)}{(1-t)^3}(-(1-t)z + (x-tz))\right] \tag{314}$$

$$=\mathcal{N}(x;tz,(1-t)^2)\left[(1-t)^{-1} - \frac{(x-tz)}{(1-t)^3}(x-z)\right] \tag{315}$$

Let's assume that $J_t(x|z) = J_t(x)$ for a state-independent jump distribution $J_t(x)$. Further, let's set $\tilde{\lambda}_t(x) = \lambda_t(x)(1 - t)$. Then jump continuity equation becomes:

$$\mathcal{N}(x; tz, (1-t)^2) \left[ 1 - \frac{(x - tz)(x - z)}{(1-t)^2} + \tilde{\lambda}_t(x) \right] = J_t(x) \int \tilde{\lambda}_t(\tilde{x}) \mathcal{N}(\tilde{x}; tz, (1-t)^2) d\tilde{x} \tag{316}$$

$$\frac{\mathcal{N}(x; tz, (1-t)^2) \left[ 1 - \frac{(x-tz)(x-z)}{(1-t)^2} + \tilde{\lambda}_t(x) \right]}{\int \tilde{\lambda}_t(\tilde{x}) \mathcal{N}(\tilde{x}; tz, (1-t)^2) d\tilde{x}} = J_t(x) \tag{317}$$

To be a valid probability distribution, $J_t$ must fulfill:

$$J_t(x) \geq 0 \Leftrightarrow \tilde{\lambda}_t(x) \geq \max\left( \frac{(x - tz)(x - z)}{(1-t)^2} - 1, 0 \right) \tag{318}$$

$$1 = \int J_t(x) dx \tag{319}$$

$$\Leftrightarrow 0 = \int \mathcal{N}(x; tz, (1-t)^2) \left[ 1 - \frac{(x - tz)(x - z)}{(1-t)^2} \right] dx \tag{320}$$

$$\Leftrightarrow 0 = 1 - \int \mathcal{N}(x; tz, (1-t)^2) \frac{(x - tz)(x - z)}{(1-t)^2} dx \tag{321}$$

$$\Leftrightarrow 0 = 1 - \int \mathcal{N}(x; tz, (1-t)^2) \frac{x^2 - (t + 1)zx + tz^2}{(1-t)^2} dx \tag{322}$$

$$\Leftrightarrow 0 = 1 - \frac{(1-t)^2 + t^2z^2 - (t+1)z^2t + tz^2}{(1-t)^2} \tag{323}$$

$$\Leftrightarrow 0 = \frac{t^2z^2 - (t+1)z^2t + tz^2}{(1-t)^2} \tag{324}$$

$$\Leftrightarrow 0 = \frac{-z^2t + tz^2}{(1-t)^2} \tag{325}$$

$$\Leftrightarrow 0 = 0 \tag{326}$$

Hence, we can see that $J_t(x)$ is indeed a valid probability distribution. Using that $\tilde{\lambda}_t(x) = \lambda_t(x)(1 - t)$, we get the following result:

$$\lambda_t(x) = \frac{\max\left(\frac{(x-tz)(x-z)}{(1-t)^2} - 1, 0\right)}{1 - t} \tag{327}$$

$$= \frac{\max\left((x - tz)(x - z) - (1 - t)^2, 0\right)}{(1 - t)^3} \tag{328}$$

$$= \frac{\max\left(x^2 - (t + 1)xz - (1 - t)^2 + tz^2, 0\right)}{(1 - t)^3} \tag{329}$$

$$J_t(x) = \frac{\mathcal{N}(x; tz, (1 - t)^2)\left[1 - \frac{(x-tz)(x-z)}{(1-t)^2} + (1 - t)\lambda_t(x)\right]}{(1 - t)\int \lambda_t(\tilde{x})\mathcal{N}(\tilde{x}; tz, (1 - t)^2)d\tilde{x}} \tag{330}$$

$$= \frac{\left[1 - \frac{(x-tz)(x-z)}{(1-t)^2}\right]_+ \mathcal{N}(x; tz, (1 - t)^2)}{\int \left[1 - \frac{(\tilde{x}-tz)(\tilde{x}-z)}{(1-t)^2}\right]_+ \mathcal{N}(\tilde{x}; tz, (1 - t)^2)d\tilde{x}} \tag{331}$$

$$= \frac{\left[(1 - t)^2 - (x - tz)(x - z)\right]_+ \mathcal{N}(x; tz, (1 - t)^2)}{\int \left[(1 - t)^2 - (\tilde{x} - tz)(\tilde{x} - z)\right]_+ \mathcal{N}(\tilde{x}; tz, (1 - t)^2)d\tilde{x}} \tag{332}$$

$$= \frac{\left[-x^2 + (t + 1)xz + (1 - t)^2 - tz^2\right]_+ \mathcal{N}(x; tz, (1 - t)^2)}{\int \left[-x^2 + (t + 1)xz + (1 - t)^2 - tz^2\right]_+ \mathcal{N}(\tilde{x}; tz, (1 - t)^2)d\tilde{x}} \tag{333}$$

Let's study the 2nd degree polynomial that is used here:

$$k_t(x) = x^2 - (t + 1)xz - (1 - t)^2 + tz^2 \tag{334}$$

$$x_{1,2} = \frac{(t + 1)z \pm \sqrt{(t + 1)^2 z^2 + 4(1 - t)^2 - 4tz^2}}{2} \tag{335}$$

$$= \frac{(t + 1)z \pm \sqrt{(t^2 + 2t + 1)z^2 + 4 - 8t + 4t^2 - 4tz^2}}{2} \tag{336}$$

$$= \frac{(t + 1)z \pm \sqrt{(t^2 - 2t + 1)z^2 + 4 - 8t + 4t^2}}{2} \tag{337}$$

$$= \frac{(t + 1)z \pm \sqrt{(1 - t)^2 z^2 + 4(1 - t)^2}}{2} \tag{338}$$

$$= \frac{(t + 1)z \pm |1 - t|\sqrt{z^2 + 4}}{2} \tag{339}$$

$$= \frac{tz + z}{2} \pm |1 - t|\sqrt{\frac{z^2}{4} + 1} \tag{340}$$

The above says intuitively that the jump intensity is "most negative" around the area at the arithmetic of the currrent mean $tz$ and the final mean $z$. Note that for $t \approx 1$, it holds that only $x$ with $x$ close to $z$ have $p(x) < 0$, all others must jump.

As an aside, the above allows to know the support of the marginal $J_t$ apriori:

$$\frac{tz+z}{2} \pm |1-t|\sqrt{\frac{z^2}{4}+1} \le c\frac{t+1}{2} + (1-t)\sqrt{\frac{c^2}{4}+1} \tag{341}$$

$$= \left[\frac{c}{2} - \sqrt{\frac{c^2}{4}+1}\right]t + \sqrt{\frac{c^2}{4}+1} + \frac{c}{2} \tag{342}$$

$$\le \sqrt{\frac{c^2}{4}+1} + \frac{c}{2} \tag{343}$$

where $c$ is the upper boundary of the support of the data. A reverse inequality for holds the lower boundary.

**Summary.** The CondOT probability path is generated by a jump process with jump intensity $\lambda_t(x)$ and state-independent jump distribution $J_t$ given by:

$$\lambda_t(x) = \frac{[k_t(x)]_+}{(1-t)^3} \tag{344}$$

$$J_t(x; \tilde{x}) = J_t(x) \propto [-k_t(x)]_+ \mathcal{N}(x, tz, (1-t)^2) \tag{345}$$

$$\text{where} \quad k_t(x) = x^2 - (t+1)xz - (1-t)^2 + tz^2 \tag{346}$$

$$Q_t(y; x) = \lambda_t(x)J_t(y; x) \tag{347}$$

Intuitively, we jump at $x_t$ only if $k(x_t)$ has a positive value. If we jump, we jump to a region of negative $k_t(x)$ proportional to $k_t(x)$ multiplied with the desired density.

## F  DETAILS FOR EUCLIDEAN JUMP MODEL

We use a jump model with $Q_t(y; x) = \lambda_t(x)J_t(y; x)$ where $\lambda_t, J_t$ are described in eqs. (345) and (347).

**Sampling.** For sampling, we can use the fact that $\lambda_t(x) = [k_t(x)]_+/(1-t)^3$ factorizes in a part that is relatively constant across time and a part that is relatively time-dependent. Specifically, we can set:

$$\lambda_{t+s}(x) \approx \frac{[k_t(x)]_+}{(1-t-s)^3} \quad 0 \le s < h \tag{348}$$

Which gives:

$$\mathbb{P}[\text{No Jump in } [t, t+h)] = \exp(-\int_0^h \lambda_{t+s}(x)ds) \tag{349}$$

$$\approx \exp(-[k_t(x)]_+ \int_0^h \frac{1}{(1-t-s)^3}ds) \tag{350}$$

$$= \exp(-[k_t(x)]_+[\frac{1}{2}(1-t-s)^{-2}]_0^h) \tag{351}$$

$$= \exp\left(-\frac{[k_t(x)]_+}{2}\left[\frac{1}{(1-t-h)^2} - \frac{1}{(1-t)^2}\right]\right) \tag{352}$$

$$= \exp\left(\frac{[k_t(x)]_+}{2}\left[\frac{1}{(1-t)^2} - \frac{1}{(1-t-h)^2}\right]\right) \tag{353}$$

$$= \exp\left(\frac{\frac{[k_t(x)]_+}{2}}{(1-t)^3}\left[1-t - \frac{(1-t)^3}{(1-t-h)^2}\right]\right) \tag{354}$$

$$= \exp\left(\frac{1}{2}\lambda_t(x)(1-t)\left[1 - \frac{(1-t)^2}{(1-t-h)^2}\right]\right) \tag{355}$$

$$=: R_{t,t+h}(\lambda_t(x)) \tag{356}$$

Therefore, the above gives us a valid scheduler to decide whether to jump in a time-interval $[t, t+h)$ or not. For us, this modification made a significant difference in the image generation results (e.g. FID 12 vs 4.5 on CIFAR-10).

**Extension to multi-dimensional case.** We assume our data is multi-dimensional and lies in $\mathbb{R}^d$ and we can use proposition 4 to extend the model from $1d$ to multiple dimensions. Specifically, for $x \in \mathbb{R}^d$ our model is given via

$$\lambda_t^d(x) = (\lambda_t^1(x), \ldots, \lambda_t^d(x)) \tag{357}$$

$$J_t^d(x) = (J_t^1(x), \ldots, J_t^d(x)) \tag{358}$$

where $\lambda_t^i(x) \geq 0$ and $J_t^i(x)$ is a categorical distribution (using softmax) over a fixed set of bins in $[-1, 1]$ (support of normalized images). On images, we implement this by using a U-Net architecture with $b + 1$ channels where $b$ describes the number of bins. During sampling, for each time update $t \mapsto t + h$, updates happen independently per dimension. Specifically,

$$X_{t+h} = (X_{t+h}^1, \ldots, X_{t+h}^d) \tag{359}$$

$$m_i \sim \text{Bernoulli}(1 - R_{t,t+h}(\lambda_t(x))) \tag{360}$$

$$X_{t+h}^i = \begin{cases} X_t & \text{if } m = 0 \\ \sim J_t^i(X_t) & \text{if } m = 1 \end{cases} \tag{361}$$

**Loss function.** As a loss function, we use an infinitesimal KL-divergence in $1d$ via

$$Q_t(y; x) = J_t(y; x)\lambda_t(x), \quad Q_t^\theta(y; x) = J_t(y; x)\lambda_t^\theta(x) \tag{362}$$

$$D(Q_t(y; x), Q_t^\theta(y; x)) = \sum_{y \neq x} Q_t^\theta(y; x) - Q_t(y; x) \log Q_t^\theta(y; x) \tag{363}$$

where the sum of $y$'s is here over regularly spaced bin values in $[-1, 1]$. We extend the above loss to the multi-dimensional case via

$$Q_t^i(y^i; x) = J_t^i(y^i; x)\lambda_t^i(x), \quad Q_t^{\theta,i}(y^i; x) = J_t^{\theta,i}(y^i; x)\lambda_t^{\theta,i}(x) \tag{364}$$

$$D(Q_t(y; x), Q_t^\theta(y; x)) = \sum_{i=1}^d D_0(Q_t^i(y^i; x), Q_t^{\theta,i}(y^i; x)) \tag{365}$$

## G  DETAILS FOR PROTEIN JUMP MODEL

### G.1  JUMP SOLUTION TO THE KFE

We first derive a general jump solution to the jump continuity equation. The jump model described in app. F will be a special case of the same construction. Let $p_t(x)$ be a probability density for every $0 \le t \le 1$ on $S$ - where $S$ is an arbitrary state space. For a jump intensity $\lambda_t(x)$ and jump kernel $J_t(\tilde{x}; x)$, the jump continuity equation is given by:

$$\frac{\partial}{\partial t} p_t(x) = \int \lambda_t(\tilde{x}) J_t(x; \tilde{x}) p_t(\tilde{x}) d\tilde{x} - p_t(x) \lambda_t(x) \tag{366}$$

$$\Leftrightarrow \quad p_t(x)[\frac{\partial}{\partial t} \log p_t(x) + \lambda_t(x)] = \int \lambda_t(\tilde{x}) J_t(x; \tilde{x}) p_t(\tilde{x}) d\tilde{x} \tag{367}$$

Making $J_t(x; \tilde{x}) = J_t(x)$ state-independent, we get:

$$p_t(x)[\frac{\partial}{\partial t} \log p_t(x) + \lambda_t(x)] = J_t(x) \int \lambda_t(\tilde{x}) p_t(\tilde{x}) d\tilde{x} \tag{368}$$

$$\Leftrightarrow \quad \frac{p_t(x)[\frac{\partial}{\partial t} \log p_t(x) + \lambda_t(x)]}{\int \lambda_t(\tilde{x}) p_t(\tilde{x}) d\tilde{x}} = J_t(x) \tag{369}$$

We require $J_t(x)$ to be a probability density and $\lambda_t(x) \ge 0$. Therefore, we get the two necessary constraints:

$$\lambda_t(x) \ge [-\frac{\partial}{\partial t} \log p_t(x)]_+ \tag{370}$$

$$1 = \int J_t(x) dx \tag{371}$$

$$\Leftrightarrow \quad \int \lambda_t(x) p_t(x) dx = \int p_t(x) \left[ \frac{\partial}{\partial t} \log p_t(x) + \lambda_t(x) \right]_+ dx \tag{372}$$

$$\Leftrightarrow \quad 0 = \int \frac{\partial}{\partial t} p_t(x) dx \tag{373}$$

$$\Leftrightarrow \quad 0 = \frac{\partial}{\partial t} \int p_t(x) dx \tag{374}$$

$$\Leftrightarrow \quad 0 = \frac{\partial}{\partial t} \int 1 dx \tag{375}$$

$$\Leftrightarrow \quad 0 = 0 \tag{376}$$

Therefore, $\lambda_t(x)$ and $J_t(x)$ defined as above are always a solution to the jump continuity equation for any state space. For minimal jump intensity $\lambda_t$, we get:

$$\lambda_t(x) = [-\frac{\partial}{\partial t} \log p_t(x)]_+ \tag{377}$$

$$= \frac{[-\frac{\partial}{\partial t} p_t(x)]_+}{p_t(x)} \tag{378}$$

$$J_t(x) = \frac{p_t(x)[\frac{\partial}{\partial t} \log p_t(x)]_+}{\int [-\frac{\partial}{\partial t} \log p_t(\tilde{x})]_+ p_t(\tilde{x}) d\tilde{x}} \tag{379}$$

$$= \frac{[\frac{\partial}{\partial t} p_t(x)]_+}{\int [\frac{\partial}{\partial t} p_t(\tilde{x})]_+ d\tilde{x}} \tag{380}$$

$$\tag{381}$$

The above equations are illustrated in app. G.1 for a sphere in $\mathbb{R}^3$. The above in fact represents a general solution to arbitrary state spaces.

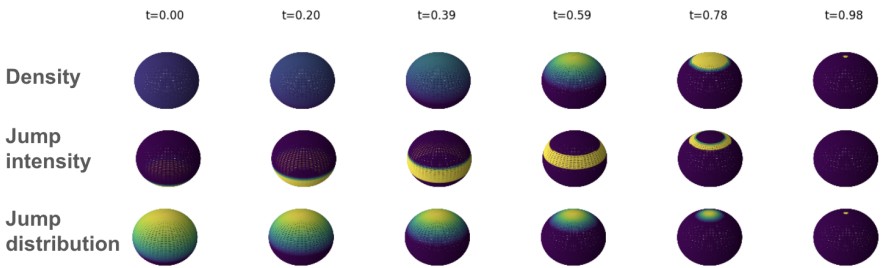

Figure 9: Illustration of jump model on manifolds. Left: Illustration of conditional probability $p_t(dx|z)$ on sphere with $z = (0,0,1)^T$ (North pole). Top: density $p_t(x|z)$. Middle: jump intensity $\lambda_t(x) = \int Q_t(dy;x)$. Bottom: Jump distribution $J_t(dy;x) = Q_t(dy;x)/\int Q_t(dy;x)$.

### G.2 PROBABILITY PATHS AND COMPUTING $\lambda_t, J_t$

We consider the quaternion model for $SO(3)$, namely, each element of $SO(3)$ is represented by a unit vector $x \in \mathcal{S}^3 \subset \mathbb{R}^4$, where

$$\mathcal{S}^3 = \left\{ x \in \mathbb{R}^4 \mid \|x\| = 1 \right\} \tag{382}$$

with $\|x\| = 1$. We consider a probability path of Fisher-von-Mises distributions given by

$$p_t(x|x_1) = C_p(\kappa_t) \exp(\kappa_t x_1^T x) \tag{383}$$

for a scheduler $\kappa_t$ such that $\kappa_0 = 0$ and $\kappa_1 >> 0$ and normalization constant $C_p(\kappa_t)$. We use a custom implementation of the Fisher-von-Mises distribution in a way that makes $p_t(x|x_1)$ differentiable with respect to $t$. We then use automatic differentiation to compute $\partial_t p_t(x|x_1)$. This allows us to compute $\lambda_t$ and $J_t$ (see previous section). To parameterize $\lambda_t$ on $SO(3)$, we simply consider a function on $SO(3)$. To parameterize $J_t$ on $SO(3)$ we place uniform bins over $SO(3)$ and make each bin represent a rotation. We note that the probability path that FrameFlow was trained on is not the probability path above. Rather, it is a probability path constructed via geodesic interpolation of a uniform to a delta function. Computing $\lambda_t, J_t$ on this path was numerically unstable for us (because of sharp boundaries introduced by the uniform distribution). Therefore, we choose a Fisher-von-Mises path as above but selected the scheduler $\kappa_t$ to optimally approximate the geodesic path. This ensured numerical stability and an (approximately) faithful recovery of the probability path. The Fisher-von-Mises path is more numerically stable as its support is all of $SO(3)$ for all $0 \le t \le 1$.

### G.3 SAMPLING

The FrameFlow model predicts a data point $z$ given a state $x_t$ at time $0 \le t \le 1$, i.e. it predicts the conditional expectation $\mathbb{E}_{z \sim p_t(dz|x)}[z]$. The marginal jump rate kernel is given by

$$Q_t(y;x_t) = \mathbb{E}_{z \sim p_t(dz|x_t)}[Q_t^z(y;x_t)] \tag{384}$$

In order to repurpose the flow model, we make the simplifying assumption that

$$Q_t(y;x_t) = \mathbb{E}_{z \sim p_t(dz|x_t)}[Q_t^z(y;x_t)] \approx Q_t^{\mathbb{E}_{z \sim p_t(dz|x_t)}[z]}(y;x_t) \tag{385}$$

Assuming that the distribution $p_t(dz|x)$ is unimodal, this corresponds to high temperature sampling of $z$. We do not employ any temperature to sample from $J_t$ or $\lambda_t$ but use plain Euler sampling as described in table 1.

### G.4 EXPERIMENT DETAILS

We based our implementation off `https://github.com/jasonkyuyim/multiflow` and downloaded pre-trained weights from the same repository. We use Euler-Maruyama integrator and 100 discretization steps for all sampling runs. Each sample uses 100 neural network function evaluations (NFEs). Neural network architecture details can be found in Campbell et al. (2024b). The size of the model is 17.4 million parameters. All MultiFlow hyperparameters, except those for the jump model, use the default ones provided in in the open source code. We found increasing the number of $SO(3)$ jumps bins to improve results and use 2056 bins in our experiments. For our metrics, we deviate slightly from (Yim et al., 2023a) by only reporting diversity and novelty and exclude designability. As noted in (Yim et al., 2024), designability can be artificially high if the generative model samples the same protein repeatedly. Nevertheles, using designability to filter bad samples is still important since protein generative models are prone to hallucination and producing proteins that would never be real. A detailed description of designability can be found in (Yim et al., 2023b). Our metrics already take designability into account by first filtering the sampled proteins to only be the designable proteins then clustering the protein structures to compute diversity and searching against protein datasets to measure novelty. Following the widely adopted benchmark in Watson et al. (2023), we sample 100 proteins for each length 70, 100, 200, 300 for a total of $n = 400$ samples then compute each metric as follows:

1. **Diversity (Div)**: Taking only the samples passing the designability filter, we use MaxCluster (Herbert & Sternberg, 2008) to compute the number of clusters $n_c$. We report $n_n/n$ which is the proportion of designable clusters to number of total samples. The higher this value, the more diverse the generated samples are after the designability filter. Diversity is important in protein design where it is ideal to test many diverse protein candidates in the hopes of having many shots to make a new drug or medicine for instance.

2. **Novelty (Nov)**: Taking only the samples passing the designability filter, we use FoldSeek (Van Kempen et al., 2024) to compute the similarity of each sample to the Protein Data Bank (PDB) (Berman et al., 2000). The similarity score is given as a protein structure alignment score called the TM-score (Xu & Zhang, 2010) where a value of 0.5 or less means the two structures are likely to be distinct. The probability of the two structures being similar with the same biological function increases as the score goes to 1.0. Therefore, the dissimilarity is 1 - (TM-score). We report the average dissimilarity of the outputs of FoldSeek as the novelty since this describes how "novel" the designable samples are on average compared to the known protein structures.

For baselines, we follow the ones described in Campbell et al. (2024b) with the addition of FoldFlow (Bose et al., 2023) since it was recently open sourced. Our full results are provided in Table 5 whereas a truncated version is presented in the main text Table 3. We remove mention of Des. in the main text since this metric can be misleading – it can easily be hacked and achieve 100% if the method repeatedly samples the same designable protein. As we see, our jump modifications lead to the best results in all categories. The ensemble model of jumps and flows performs similarly as the pure jump model.

| Method | Co-design 1 (multi-modal) | | | PMPNN 8 (unimodal) | | |
|---|---|---|---|---|---|---|
| | Des | Div | Nov | Des | Div | Nov |
| RFdiffusion | | | | 0.87 | 0.4 | 0.37 |
| FrameFlow | | | | 0.86 | 0.39 | 0.39 |
| FoldFlow | | | | 0.81 | 0.24 | 0.32 |
| Protpardelle | 0.63 | 0.10 | 0.40 | 0.90 | 0.12 | **0.41** |
| ProteinGenerator | 0.37 | 0.09 | 0.31 | 0.89 | 0.19 | 0.35 |
| MultiFlow | **0.86** | 0.38 | 0.39 | **0.99** | 0.52 | 0.39 |
| w/ $SO(3)$ jumps (ours) | 0.76 | **0.48** | **0.41** | 0.96 | **0.63** | **0.41** |
| w/ $SO(3)$ jumps + flow (ours) | 0.78 | 0.47 | 0.40 | 0.96 | 0.59 | 0.40 |

Table 5: Full protein generation results.

# H    EXTENDED DISCUSSION OF RELATED WORKS

## H.1    FLOW MATCHING

Flow matching and rectified flows (Lipman et al., 2022; Liu et al., 2022) are immediate instances of Generator Matching leveraging the flow-specific versions of the KFE given by the continuity equation (see table 1 and app. A.5.1 for a derivation). We briefly describe here how one can map the propositions from this work to their work. Specifically, flow matching restricts itself to generators of the form $\mathcal{L}_t^\theta f(x) = \nabla f(x)^T u_t^\theta(x)$ for a vector field $u_t^\theta(x)$ parameterized by a neural network with parameters $\theta$. Given a conditional vector field $u_t(x|z)$ and a probability path $p_t(x|z)$, the corresponding marginal vector field in flow matching (see (Lipman et al., 2022, equation (8))) is given by

$$u_t(x) = \int u_t(x|z) \frac{p_t(x|z) p_{\text{data}}(z)}{p_t(x)} dz \tag{386}$$

and corresponds to the marginal generator (see proposition 1). The Bregman divergence used is the mean squared error (MSE) obtained by choosing $\phi(x) = \|x\|^2$ in eq. (15). The conditional flow matching loss (Lipman et al., 2022, Theorem 2 ) is a special case of proposition 2. Therefore, Generator Matching can be seen as a generalization of the principles of flow matching to the space of Markov process generators for arbitrary state spaces.

## H.2    DENOISING DIFFUSION MODELS

From the perspective of Generator Matching, denoising diffusion models (Song et al., 2020) are flow models with two conceptual differences to flow matching: (1) They allow for stochastic sampling via SDEs by adding a divergence-free Langevin component (see proposition 3) and (2) A probability path is defined via forward noising process and a time-reversal of that process serves a solution to the KFE. We explain both differences below.

**Time-reversal to find solutions to the KFE.**    First, let's discuss the idea of time-reversal. In diffusion models, a probability path $p_t(dx|z)$ is constructed via a forward diffusion process that noises data (i.e. here, not only the marginals are specified but a full distribution across all time points). The conceptual idea of time-reversal allows to find solutions to the KFE. We illustrate this here. We derived this already in app. H.2 for general Markov processes and illustrate it here for diffusion models. Specifically, let's consider a Markov noising process $\bar{X}_t$ given by a variance-exploding SDE $d\bar{X}_t = \sigma_t d\bar{W}_t$ that goes from $t = 1$ to $t = 0$

backwards in time[1]. Then we know that the KFE holds in reverse time

$$\partial_t p_t f = -\langle p_t, \bar{\mathcal{L}}_t f\rangle \qquad\qquad \blacktriangleright \text{ KFE in reverse time} \qquad (387)$$

$$= -\int p_t(x)\frac{\sigma_t^2}{2}\Delta f(x)dx \qquad\qquad \blacktriangleright \text{ generator for diffusions, see table 1} \qquad (388)$$

$$= -\int p_t(x)\frac{\sigma_t^2}{2}\nabla\cdot\nabla f(x)dx \qquad\qquad \blacktriangleright \text{ definition of Laplacian} \qquad (389)$$

$$= \int \nabla p_t(x)^T\frac{\sigma_t^2}{2}\nabla f(x)dx \qquad\qquad \blacktriangleright \text{ partial integration} \qquad (390)$$

$$= \int p_t(x)\underbrace{\nabla f(x)^T\left[\frac{\sigma_t^2}{2}\nabla\log p_t(x)\right]}_{=:\mathcal{L}_t f(x)}dx \qquad\qquad \blacktriangleright \text{ derivative of log} \qquad (391)$$

$$= \langle p_t, \mathcal{L}_t f\rangle \qquad (392)$$

As we can see, the operator $\mathcal{L}_t f = \nabla f^T\frac{\sigma_t^2}{2}\nabla\log p_t$ fulfils the KFE in forward time. Further, $\mathcal{L}_t$ corresponds to a generator of a flow with vector $\frac{\sigma_t^2}{2}\nabla\log p_t$. We can see that the corresponding flow must generate the probability path $p_t$. This flow is commonly called the *probability flow ODE* (Song et al., 2020). The conditional Generator Matching loss with a mean squared error then leads to the denoising score matching loss (Vincent, 2011)

$$\mathbb{E}_{z\sim p_{\text{data}},t\sim\text{Unif},x\sim p_t(\cdot|z)}[\|\nabla\log p_t(x|z) - s_\theta(x,t)\|^2] \qquad (393)$$

for a neural network $s_\theta : \mathbb{R}^d \times [0,1] \to \mathbb{R}^d$ approximating the score vector field. We remark that mathematically there is also a stronger notion of time-reversal that also requires the joint distribution (across time points) to the be same - as opposed to just the marginals.

**Stochastic sampling by adding a divergence component.** Second, let's discuss stochastic sampling. For a general probability path $p_t$ with density $p_t(x)$, a general divergence-free component is given via the Langevin generator corresponding to an SDE with drift $\sigma_t^2\nabla\log p_t(x)$ and diffusion coefficient $\sqrt{2\sigma_t}$

$$\mathcal{L}_t^{\text{Langevin}}f(x) = \sigma_t^2\nabla f(x)^T\nabla\log p_t(x) + \sigma_t^2\Delta f(x), \qquad (394)$$

This fact is widely applied in statistical physics and Markov chain Monte Carlo methods in the form of Langevin dynamics (Roberts & Tweedie, 1996). To see that this is divergence-free (in the sense as defined in proposition 3), we can simply use partial integration

$$\left\langle p_t, \mathcal{L}_t^{\text{Langevin}}f\right\rangle \qquad (395)$$

$$= \int p_t(x)\sigma_t^2\nabla f(x)^T\nabla\log p_t(x)dx + \int p_t(x)\sigma_t^2\Delta f(x)dx \qquad \blacktriangleright \text{ by definition} \qquad (396)$$

$$= \int \sigma_t^2\nabla f(x)^T\nabla p_t(x)dx + \int p_t(x)\sigma_t^2\Delta f(x)dx \qquad \blacktriangleright \text{ derivative of log} \qquad (397)$$

$$= -\int \sigma_t^2\Delta f(x)p_t(x)dx + \int p_t(x)\sigma_t^2\Delta f(x)]dx \qquad \blacktriangleright \text{ partial integration} \qquad (398)$$

$$= 0 \qquad (399)$$

---

[1]Note that in (Song et al., 2020) $t = 0$ corresponds to data, while we keep the convention here that $t = 1$ corresponds data

In (Song et al., 2020), they fix a specific weighting and run several iterations of a Langevin sampling iteration at every time step, there called a predictor-corrector scheme (see (Song et al., 2020, Algorithm 1-3)). However, by proposition 3 any positive weighting of the above Langevin component leads to a valid sampling procedure. This observation was already made by Karras et al. (2022, section 4) where the optimal weighting of the divergence-free Langevin component is studied experimentally in more detail.

Finally, we note that many denoising diffusion models use formulations in discrete time (Sohl-Dickstein et al., 2015; Ho et al., 2020). While these formulations do not incur a time discretization error during sampling, they incur an error in the loss formulation. The reason for that is that a similar proposition as in proposition 2 does not hold for discrete time steps (i.e. the linearization only holds for the generator and not the transition kernel). Therefore, a formulation with discrete time steps can only us an approximate loss via an ELBO lower bound using parameterized family of distributions such as a Gaussian (Sohl-Dickstein et al., 2015).

### H.3 STOCHASTIC INTERPOLANTS

Stochastic interpolants (Albergo & Vanden-Eijnden, 2022; Albergo et al., 2023) is a framework for generative modelling that shares many similarities with diffusion models and flow matching. We explain conceptual differences by placing its findings within the Generator Matching framework.

**General flow solution based on interpolant.** A difference in the perspective of stochastic interpolants is that they take a sample-based perspective, i.e. they specific an initial distribution $x_0 \sim p_0$ and a probability path $x_t \sim p_t(dx)$ is constructed implicitly via a simulator function $I$ (see (Albergo et al., 2023, definition 2.1))

$$x_t := \Phi(t, x_0, z, \epsilon) := I(t, x_0, z) + \gamma(t)\epsilon \tag{400}$$

where $\epsilon \sim \mathcal{N}(0, I_d)$ and we impose the condition that $I(0, x_0, z) = x_0$, $I(1, x_0, z) = z$ and $\gamma(0) = 0, \gamma(1) = 0$. A general flow-based solution can be derived by using the KFE (see (Albergo et al., 2023, theorem 2.6)):

$$\frac{\partial}{\partial t} \langle p_t, f \rangle = \partial_t \mathbb{E}[f(\Phi(t, x_0, z, \epsilon))] = \mathbb{E}[\nabla f(\Phi(t, x_0, z, \epsilon))^T \partial_t \Phi(t, x_0, z, \epsilon)] \tag{401}$$

$$= \mathbb{E}_{x_t \sim p_t} [\underbrace{\nabla f(x_t)^T \mathbb{E}[\partial_t \Phi(t, x_0, z, \epsilon) | x_t]}_{=: \mathcal{L}_t f(x_t)}] \tag{402}$$

$$= \mathbb{E}_{x_t \sim p_t} [\mathcal{L}_t f(x_t)] \tag{403}$$

The operator $\mathcal{L}_t f$ is a generator of a flow with vector field given by the conditional expectation of the velocity $u(x_t, t) = \mathbb{E}[\partial_t \Phi(t, x_0, z, \epsilon) | x_t]$. Hence, we see that the above vector field generates the probability path $p_t$. This can be trained in the same way with a mean-squared error as for denoising diffusion models and flow matching.

**Stochastic sampling.** Another difference of stochastic interpolants is a generalization of the stochastic sampling procedure for denoising diffusion models (see app. H.2). One advantage of denoising diffusion models is that one gets - informally - "2 advantages for 1": Specifically, both for learning a flow-based solution to the KFE and for stochastic sampling, we only need to learn the function $\nabla \log p_t$ commonly called the *score function* (see app. H.2). This works because the process is constructed as a diffusion process. However, in the general case, one can still learn the score $\nabla \log p_t$ separately from the flow and then add a divergence-free Langevin component during sampling (see eq. (394)). As pointed out in (Albergo et al., 2023, theorem 2.8), the special shape of the stochastic interpolant (see eq. (400)) allows to derive a simple denoising score matching loss. In (Albergo et al., 2023, theorem 2.23), it is further shown that adding stochastic sampling leads to the ability to control the KL-divergence between the target distribution and the distribution generated by the model. This highlights an advantage of adding a noise (SDE) term because bounding the KL-divergence is in general not possible with a pure flow model (although bounds in Wasserstein distance exist for flow models, see (Benton et al., 2023)).

### H.4 DISCRETE SPACES VIA CONTINUOUS-TIME MARKOV CHAINS ("DISCRETE DIFFUSION")

For discrete state spaces $S$, the generator of a Markov process $X_t \in S$ is given by a rate transition matrix $Q_t \in \mathbb{R}^{S \times S}$ (see app. A.5.4 for derivations). Therefore, if we restrict ourselves to generators on discrete state spaces, we recover the framework developed by (Campbell et al., 2022) as an instance of Generator Matching. Proposition 1 in Campbell et al. (2022) corresponds to proposition 1 showing that the marginal generator/rate matrix corresponds a conditional generator weighted by the posterior. Proposition 2 in Campbell et al. (2022) shows that a continuous-time ELBO can be derived via a Bregman divergence. We derive a similar, slightly simpler, Bregman divergence loss in app. D.2 that also corresponds to an ELBO lower bound. Further, they use a predictor-corrector scheme (see Campbell et al. (2022, Proposition 4)) as outlined here for the general case in proposition 3. In the discrete setting, this leads to significant improvements (Gat et al., 2024).

In many applications such as language modelling, the state space decomposes into dimensions, i.e. is given via $S = \{1, \ldots, N\}^d =: [N]^d$ where $N$ is the vocabulary size. This state space is usually too large that one cannot store a full rate transition matrix $Q_t$ (and not even a single row of it) in a computer. However, using a factorized probability path, we can use proposition 4 to see that we can learn a rates $Q_t$ that update each dimension independently (i.e. it has a block structure). This reduces the dimension significantly. This was shown for discrete spaces also in Campbell et al. (2022, Proposition 3) and has since then the de facto standard for discrete diffusion models (Lou et al., 2024a; Gat et al., 2024). Instead of parameterizing the generator directly, proposition 4 also shows that one can also only learns the marginals $p_t(z_i|x)$ for each $z = (z_1, \ldots, z_n) \in [N]^d$ independently. One can then train the marginals of the posterior $p_t(z_i|x)$ via the cross-entropy loss (Gat et al., 2024; Campbell et al., 2024b).

Campbell et al. (2022) adapt the idea of time-reversal from diffusion models to find solutions of the KFE for a given probability path. Specifically, the time-reversal $Q_t$ of a process with rate matrix $\bar{Q}_t$ running in backwards-time is given via

$$Q_t(x; y) = \bar{Q}_t(y; x) \frac{p_t(x)}{p_t(y)} \tag{404}$$

In more recent works (Lou et al., 2024a), it has been shown that parmeterizing the generator by the ratio $p(y)/p(x)$ that is needed to time-reverse the process - called the "discrete score" - leads to improved results. As this is a linear parameterization of the generator, they can use the same Bregman divergence as we derive in app. D.2 to train the discrete score.

**Blackout diffusion.** As another example of a discrete Markov model, we discuss *blackout diffusion* (Santos et al., 2023). This model also considers a model on discrete state spaces $S$ with rate matrix $Q_t$ (there, denoted as $L_{mm'}$). The KFE in discrete spaces corresponds to (Santos et al., 2023, Equation (2)). A probability path is constructed via a "forward process" or noising process that allows for transitions to neighboring states, in the specific case for the blackout diffusion characterization by a decay process representing by $m \to m - 1$ (see app. A.4 for a discussion of how probability paths and time-reversal relate). The loss function in (Santos et al., 2023, Equation (11) and (12)) correspond to the ELBO likelihood loss derived in app. D.2 (up to constants). Further, Santos et al. (2023) show that this design of the probability path allows to simplify several equations during training (Santos et al., 2023, Algorithm 1) and allows to make finite-time approximations reducing the discretization error during sampling (Santos et al., 2023, Algorithm 2).

### H.5 GEOMETRIC DATA AND MANIFOLDS

We next describe how Generator Matching generalizes models on Riemannian manifolds. In the following, let $S = \mathcal{M}$ be a smooth Riemannian manifold with metric $g$. For $x \in \mathcal{M}$, let $T_x\mathcal{M}$ be the tangent space of $x$ and let $T\mathcal{M} = \bigsqcup_{x \in \mathcal{M}} T_x\mathcal{M}$ be the tangent bundle.

**Riemannian Flow Matching (Chen & Lipman, 2024).** First, we consider flows on $\mathcal{M}$ showing how we can recover Riemannian Flow Matching (Chen & Lipman, 2024). A flow on $\mathcal{M}$ is defined via a vector field $u : \mathcal{M} \times [0,1] \to T\mathcal{M}$ such that $u_t(x) \in T_x\mathcal{M}$ for all $x \in \mathcal{M}$. As every tangent space is a vector space, the space of vector fields is a vector space again. Every vector fields defines a flow $\phi_{t|s}$ that fulfils

$$\phi_{t|t}(x) = x \text{ for all } 0 \leq t \leq 1 \tag{405}$$

$$\frac{d}{dt}\phi_{t|s}(x) = u_t(\phi_{t|s}(x)) \text{ for all } 0 \leq s \leq t \leq 1 \tag{406}$$

Next, we derive the generator. Let $f : \mathcal{M} \to \mathbb{R}$ be a smooth function. Then the generator is given via

$$\mathcal{L}_t f(x) = \lim_{h \to 0} \frac{f(\phi_{t+h|t}(x)) - f(x)}{h} = \langle \nabla f(x), u_t(x) \rangle_g \tag{407}$$

where $\langle \cdot, \cdot \rangle_g$ describes the dot product defining the Riemannian metric $g$ and $\nabla f$ describes the gradient of $f$ with respect to $g$. This coincides with the Lie derivative of a function (Jost & Jost, 2008), a fundamental concept in differential geometry. Therefore, we see that $u_t(x)$ is a linear parameterization of the generator. We can then use an arbitrary Bregman divergence on $T_x\mathcal{M}$, e.g. the mean-squared error. The CGM loss then recovers the Riemannian Conditional Flow Matching loss (see (Chen & Lipman, 2024, equation (8))). This shows that Riemannian Flow Matching is a specific instance of Generator Matching with Markov processes on manifolds restricted to flows.

**Diffusion models on manifolds (De Bortoli et al., 2022; Huang et al., 2022).** Riemannian score-based generative modeling can equally be seen as an instance of GM. Similar to Euclidean diffusion models, a probability path is constructed via a forward noising process and a solution to the KFE is found via a time-reversal of the process (see app. H.2 as an example on Euclidean space). Specifically, a forward-time SDE that generates data is obtained via time-reversal (see (De Bortoli et al., 2022, theorem 3.1.)) and has the shape

$$dY_t = [-b(Y_t) + \nabla \log p_t(Y_t)]dt + dB_t^{\mathcal{M}} \tag{408}$$

where $b : \mathcal{M} \to T\mathcal{M}$ describes a drift, $\nabla \log p_t : \mathcal{M} \to T\mathcal{M}$ describe the score vector field and $\mathcal{B}_t^{\mathcal{M}}$ describes a Brownian motion on $\mathcal{M}$. The generator of the above SDE is given via

$$\mathcal{L}_t f(x) = -\langle b(x), \nabla f(x) \rangle_g + \langle \nabla f(x), \nabla \log p_t(x) \rangle_g + \frac{1}{2}\Delta_{\mathcal{M}} f(x) \tag{409}$$

where $\Delta_g f$ describes the Laplace-Beltrami operator on manifolds (Elworthy, 1998). Note that $b$ is fixed as a hyperparameter. Therefore, like in the Euclidean case, a linear parameterization of the generator is again given via a score network $s_\theta : \mathcal{M} \times [0,1] \to T\mathcal{M}$. Choosing the mean squared error as a Bregman divergence, one recovers Riemannian denoising score matching (see (De Bortoli et al., 2022, section 3.2)). However, the fact that the noising process is not analytically tractable does not give us an analytically tractable formula for $p_t(x|z)$, i.e. in GM language the conditional solution to the KFE is not known analytically. Therefore, this approach requires iterative simulation of a noising process during training even for geometries with analytic geodesic formulas and requires approximations of score functions as a training target (see (De Bortoli et al., 2022, table 2)).

### H.6 MULTIMODAL SPACES

Markovian multimodal generative models have been previously described for specific spaces and model classes (Anand & Achim, 2022; Campbell et al., 2024b).

**MultiFlow (Campbell et al., 2024b).**    As an example, we illustrate here how the work by Campbell et al. (2024b) for multimodal protein generation fits into the GM framework. Campbell et al. (2024b) first construct a generative model on discrete spaces representing amino acids of a protein using the recipe outlined in (Campbell et al., 2022) (see app. H.4 for explanations how this fits into GM framework). Further, they use a Euclidean flow model (Lipman et al., 2022) for the translation components of the protein and a Riemannian Flow model (Yim et al., 2024) is used for frames represented via elements on $SO(3)$. Using a factorized probability, one can use the recipe outlined in proposition 4 to build a multimodal generative model. This corresponds to Proposition 4.1. and Proposition 4.2. in (Campbell et al., 2024b). As outlined in proposition 4, loss functions for individual modalities can be summed up, as done in (Campbell et al., 2024b, equation (16)). Therefore, the work by Campbell et al. (2024b) is a direct example of the power of the recipe outlined in proposition 4 to build a Markovian generative model.

## H.7    Jump Models on non-discrete state spaces

We discuss a selection of models relying on (and including) jump models. In app. H.4, we discussed already continuous-time Markov processes, which are jump models on discrete state spaces. Hence, we focus here on other state spaces.

**Piecewise deterministic generative models (Bertazzi et al., 2024).**    The combination of a flow and jump Markov process is commonly called a piecewise-deterministic Markov process (PDMP) (Davis, 1984; Del Moral & Penev, 2017). Recently, PDMPs have been applied by Bertazzi et al. (2024) for generative modeling in the form of "piecewise-deterministic generative models". Specifically, they consider models in phase space, i.e. where the states is given via a tuple $(x, v)$ of a location $x$ and a velocity $v$. Similar to diffusion models, they construct a probability path via a forward/noising process and use its time-reversal as a solution to the KFE of the corresponding probability path. The jump intensity and jump kernels of the time-reversal can be linearly parameterized via likelihood ratios (see (Bertazzi et al., 2024, Equation (4) and (5))) - this corresponds to a linear parameterization of the generator (see app. A.6) and is related to the linear parameterization used by (Lou et al., 2024a) on discrete state spaces.

**Score-based generative models with Levy Processes (Yoon et al., 2023).**    A general class of stochastic processes is given by Lévy processes that are characterized by independent and stationary increments. Lévy processes include certain flows, diffusion, and jump processes and were recently used for generative modeling purposes by Yoon et al. (2023). Here, Yoon et al. (2023) construct SDEs driven by Lévy processes as a forward process - implicitly defining a probability path (see app. A.4) - and the time-reversal is giving a solution to the KFE (see app. A.4). There, the time-reversal depends on the *fractional score function* that determines the flow component of the backwards process (the other parts are hyperparameters and don't have to be learned). The fractional score serves as a linear parameterization of the generator (see app. A.6). To learn this vector field, we can employ the conditional Generator Matching loss with the Bregman divergence given by the MSE - this recovers the fractional denoising score matching loss (i.e. (Yoon et al., 2023, Theorem 4.3.) directly corresponds to proposition 2). While Lévy processes in principle include jump processes (and also their SDEs), we note that the work by (Yoon et al., 2023) only consider learning the flow component of the Markov processes (the rest are hyperparameters), while here we show how to also learn the jump kernel or the diffusion coefficient.

**Trans-dimensional jump diffusion (Campbell et al., 2024a).**    Recently, jump processes have recently been leveraged to model generative modeling tasks over state-spaces that includes a various numbers of dimensions (Campbell et al., 2024a). From the perspective of GM, we can consider this to be on state space $S = \cup_{k=1}^{d} \{d\} \times \mathbb{R}^d$. A conditional probability path is constructed via a forward noising process and a time-reversal serve as a solution to the KFE ((Campbell et al., 2024a, Proposition 1)). The loss in (Campbell et al., 2024a, Proposition 2) corresponds to a combined loss of two Bregman divergences: (1) A mean-squared

error and (2) The Bregman divergence considered in app. D.2 (the jump measure $Q_t$ is denoted there as the product $\lambda_t(x) A_t^\theta(y; x)$ and uses the fact that $\log(\lambda_t(x) A_t^\theta(y; x)) = \log(\lambda_t(x)) + \log(A_t^\theta(y; x)))$. The second term has an additional time-reversal step similar to (Campbell et al., 2022).

## I  ADDITIONAL EXPERIMENTS ON LOSS FUNCTIONS

To illustrate the utility of Bregman divergences for GM models, we ran additional experiments using a flow model on image generation. Specifically, we use the following two functions $\phi : \mathbb{R} \to \mathbb{R}$:

$$\phi(x) = \frac{\exp(\alpha x) + \exp(-\alpha x)}{2} = \cosh(\alpha x) \qquad \blacktriangleright \text{Cosh}$$

$$\phi(x) = \exp(\alpha x) \qquad \blacktriangleright \text{Exponential}$$

We construct Bregman divergences with these where we give weight $0.5$ to the MSE and weight $0.5$ to the above $\phi(x)$. We then train a flow model on CIFAR-10 using the resulting Bregman divergences. We ablate over a few options of $\alpha \in \mathbb{R}$. In fig. 10, one can see the training dynamics are significantly more stable and the final results are better by modifying the Bregman divergence loss function. For example, the MSE model achieves an FID score of $2.62$ (existing flow matching or diffusion models), while the Bregman divergence achieves a score of $2.54$. Note that this was achieved by modifying one line in an existing codebase. Note that the results slightly differ from fig. 4 because those were obtained with a joint flow and jump model trained together. Of course, further exploration in the future is needed to explore the design space of Bregman divergences. Here, **this shows the practical utility of the space of loss functions given by Bregman divergences**.

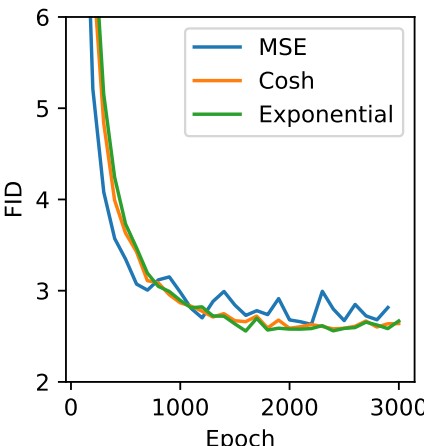

Figure 10: Training evolution of a flow model trained on CIFAR-10 with different Bregman divergences. As one can see, using Bregman divergences other than the MSE leads to improved training stability and final performance (achieved with changing a single line of code). Results are reported using a dopri5 sampler.

