# OpenReview forum: "Generator Matching: Generative modeling with arbitrary Markov processes"
_ICLR.cc/2025/Conference — ICLR 2025 Oral_

### Official Review · Reviewer_RJkg · 2024-10-30

**Soundness:** 4
**Presentation:** 4
**Contribution:** 4
**Rating:** 8
**Confidence:** 4

**Summary:**

The paper introduces a generalized framework called *generator matching* to train diffusion-based models for continuous-time Markov processes, encompassing various well-known models as special cases. In particular, the proposed framework provides a consistent theory for training generative models of continuous-time Markov processes within the class of Feller processes. Feller processes include various types of continuous-time Markov processes, such as Ito diffusions, continuous-time Markov chains (CTMCs), jump processes, and continuous-time normalizing flows.

Specifically, this paper first introduces some general properties of the continuous-time Markov processes to motivate the proposed method. Many continuous-time Markov processes of interest exhibit two main characteristics. First, their transition probability follows the Markov property. Second, sample paths created by Markov transitions exhibit some form of continuity (right-continuous or left-continuous). These two characteristics allow the averaged behavior of the transition probability at an infinitesimal time scale to be expressed by a gradient operator on the state space $S$. This operator is known as the (infinitesimal) generator of a Markov process and is uniquely defined for each transition probability. Notably, this generator is not simply a gradient evaluated at a point in the state space $S$ but an operator that yields the gradient when applied to a function $f$ on the process. It is essential to remember that a Markov process is uniquely determined by its transition probability (and initial distribution). Therefore, this averaged gradient operator, or generator, along with the initial distribution, uniquely determines the Markov process. The paper proposes a training method for a model generator based on sample paths of a reference Markov process, in particular, ones to bridge or connect two distributions, as seen in bridge matching or flow matching.

While similar works on generative modeling for Feller processes exist [1], this paper proposes a novel parameterization that directly models the generator of Markov processes. This enables direct training of the generator, in contrast to other parameterizations such as score-based training in Euclidean spaces or probability ratio training in discrete state spaces. Recall that the (infinitesimal) generator of a continuous-time Markov process is a type of averaged gradient operator on the stochastic process. When this gradient operator is applied to a function $f$ on the state space, it typically comprises a combination of the gradient, Hessian, divergence, or difference of $f$ and the transition probability-related parameters. The specific form of the generator depends on the chosen Markov process. However, the paper emphasizes that the generator for most Markov processes of interest can be expressed as $\langle \mathcal{K}f, F \rangle$, where $\mathcal{K}$ is a linear operator independent of the Markov process’s transition probability, making it parameter-independent. Thus, $F_t$ uniquely determines the generator, and the paper proposes to train $F_t$ directly—a technique called *generator matching*. The paper specifically employs a Bregman divergence-based objective for this approach.

This theory applies to all Feller processes, including continuous-time Markov processes like Ito diffusions, continuous-time Markov chains, jump processes, and continuous-time normalizing flows. Moreover, since generators are linear operators, defining a Markov process with a transition probability that combines various types of transition probabilities becomes straightforward: the generator can simply be represented as the sum of the original generators. This approach enables an intuitive understanding of combined Markov processes by summing their respective generators.

Unfortunately, generator matching faces the same challenges as (explicit) score matching, particularly since the exact $F_t$ of the reference processes is generally unknown. To address this, the authors show that $F_t$ at time $t$ can be reformulated as the expected value of an initial value-conditioned $F_t$ (with the expectation taken with respect to the posterior distribution of $X_0$ given $X_t$). Using this insight, the paper proposes *conditional generator matching*, which serves a similar function to denoising score matching in Euclidean diffusion-based models.

Finally, the paper demonstrates the effectiveness of the proposed method through various experiments.

[1] Joe Benton, Yuyang Shi, Valentin De Bortoli, George Deligiannidis, and Arnaud Doucet. From denoising diffusions to denoising markov models. Journal of the Royal Statistical Society Series B: Statistical Methodology, 86(2):286–301, 2024.

----
I have updated the overall rating from 6 to 8 and the soundness & presentation from 3 to 4 after the authors' rebuttal.

**Strengths:**

In my understanding, the paper's contributions are clear, and I consider that the results are essential for several reasons.

First and foremost, the authors introduce a unified framework that combines a unified theory to explain how distinct Markov processes, defined across various state spaces, can be represented within a consistent language. While this framework may be not be novel itself, the paper exploits this framework to propose a novel method for training these processes in the context of generative modeling. Specifically, the paper’s new generator-based parameterization leverages this unified theory, offering a streamlined and theoretically grounded approach to model various Markov processes in a consistent fashion.

Second, beyond recovering existing diffusion-based models within the unified framework, the paper uses this framework to establish innovative classes of generative models. For example, one of new class straightforwardly combines different types of Markov processes, paving the way for flexible yet rigorous generative modeling approaches.

I believe this paper provides invaluable insights for comprehensively understanding various diffusion-based models (including flow matching, bridge matching, and others). It offers a cohesive view that can unify these models within a broader framework. This paper could very well become a must-read for those interested in this topic, given the clarity it brings to understanding and advancing diffusion-based generative models.

**Weaknesses:**

I find the paper original and significant overall. However, the presentation should be improved. In particular, there exists some significant notational issue over the paper.

First, the author suggests that $ p_t $ can be interpreted both as a measure function (a set function on a $ \sigma $-algebra of the state space $ S $) and as a function on $ S $. This dual interpretation creates confusion in the derivations and leads to inconsistency in the notation. Additionally, it introduces an unnecessary operator $ \odot $, which is intended as an inner product. For the proofs in this submission to be correct, several notational revisions are necessary. Specifically, $ p_t $ should be defined as a density function, represented as a Radon-Nikodym (RN) derivative with respect to a reference measure $ \mu $ in a general setting.

Here, $ \mu $ denotes a reference measure on the state space $ S $, such as the Lebesgue measure for $ \mathbb{R}^n $ or the counting measure for a countable set. Consequently, $ p_t(x) = \frac{d\mathbb{P}_t}{d\mu}(x) $ represents the RN-derivative with respect to $ \mu $, often referred to as a probability density function in the Euclidean setting or as a probability mass function in a discrete (particularly countable) state space.

As a result, all equations involving $ \odot $ can be rewritten in terms of an inner product. For example, Equation (2) can be expressed as
$$
\mathbb{E}[f(X_t)] = \int f(x) p_t(x) d\mu(x) = \langle p_t, f \rangle.
$$
Another example is Equation (6), which becomes
$$
\partial_t \langle p_t, f \rangle = \langle p_t, \mathcal{L}_t f \rangle.
$$

Second, the notational issue discussed above introduces further complications, which undermine the motivation for using the $ F $-parameterization rather than directly parameterizing the generator $ \mathcal{L} $. In my understanding, for the continuous-time Markov processes mentioned, the generator $ \mathcal{L}f $ can be represented as $ \langle \mathcal{K}f, F \rangle $, where $ \mathcal{K} $ is a linear operator independent of the transition probability of the Markov process (thus parameter-independent). This formulation, in my opinion, effectively motivates the $ F $-parameterization. For example, in the case of a continuous-time Markov chain (CTMC),
$$
\sum_{y} ( f(y) - f(x) ) Q(y; x) = \Delta f(x)^\top Q(x),
$$
where $ \Delta f(x) = (f(x_1) - f(x), f(x_2) - f(x), \dots) $ and $ Q(x) = (Q(x_1; x), Q(x_2; x), \dots) $. With this notation, transitioning from matching the generator to matching the $ F $-parameterization becomes straightforward. Moreover, it allows the use of well-known objective functions, such as mean-squared error minimization, alongside the Bregman divergence-based loss. This expands the flexibility and applicability of the model training approach.

As previously mentioned, I consider this paper to be an excellent work that provides valuable insights into unifying various diffusion-based models (including flow matching, bridge matching, etc.). For those interested in this topic, I believe that it could very well become essential reading. However, due to the significance of the notational issues, there is a risk of widespread misunderstanding if these inconsistencies persist. While I believe that reviewers should respect the authors’ narrative, including their notations, I also believe that it is essential to avoid reinventing standard notations with new one, especially when these are well-established in standard textbooks. This is particularly important since once published, changing the notation would be extremely challenging. Therefore, I am inclined to assign a lower score at this stage. I am, however, open to raising the score if the paper is revised to address these issues.

Despite my willingness to increase the score following revisions, I must express a concern based on recent experience: I have frequently reviewed papers with similar notational issues where authors promised corrections but ultimately did not make the necessary changes. As a result, this time, I will not consider adjusting the score unless these notational issues are thoroughly addressed.

**Questions:**

N/A

---

> ### Author Response · Authors · 2024-11-22
>
> We want to thank the reviewer for considering our paper as "original and significant overall" and that it "could very well become a must-read for those interested in \[generative modeling\]". We understand that the main concern of the reviewer is notation and understandability of our work that we are happy to address.
>
> ## Densities vs. measures
>
> **A. Why do we use probability measures and not restrict ourselves to densities?** It was a deliberate choice for us to build a generative modeling framework that is not restricted to probability measures that admit a density with respect to a common reference measure. Let us give several examples where this is relevant:
>
> - *Discrete support:* The probability path $p\_t=\\delta\_{tz}$ on $S=\\mathbb{R}$ is a path of point measures corresponding to a line connecting $0$ and $z\\in\\mathbb{R}$.  This probability path does not admit a density with respect to the Lebesgue measure. However, this is not an artificially constructed example but such paths actually appear in cases where one conditions probability paths on start and end points (see e.g. Tong et al (2023)). There are other examples of data distributions with discrete support, e.g. when using latent space encodings with a VQ-VAE.
> - *Complex state spaces:* Generator Matching (GM) would also work for generative modeling over infinite-dimensional spaces, e.g. path space $C(\[0,1\])$ for trajectory modelling. However, measures over paths of stochastic processes in general do not have a density with respect to a common reference measure (they often only have relative densities, e.g. see Girsanov's theorem). Another example is trans-dimensional generative modeling (see e.g. section H.7).
> - *Universal characterization:* The universal characterization in theorem 1 only holds for general probability measures and jump measures (if we restrict ourselves, we would lose this result).
>
> **B. Notational changes to make density vs measure clearer to the reader** Thank you for pointing out a potential confusion regarding the notation of probability measures vs probability densities. We made several changes:
>
> - *Different notation for density and measure:* As you suggested, we introduce now formally the reference measure $\\nu$. Further, we now distinguish in the notation between a probability measure $p\_t$ and its Radon-Nikodym derivative $\\frac{dp\_t}{d\\nu}$ with respect to a reference measure $\\nu$ throughout our entire work. When operating on Euclidean space with the Lebesgue measure, we suggest to keep the old notation $p\_t(x)$ to denote a density - this is to avoid confusion with readers who are familiar with the notation of densities in Euclidean space but not with general Radon-Nikodym derivatives. However, we always state this clearly then (see e.g. A.5.1 or description of table 1).
> - *Assumptions stated explicitly:* Thank you for pointing out that it should be highlighted more clearly when we assume that a density exists. Whenever we assume that a density exists, we state this assumption now more explicitly. For example, in table 1, we state explicitly which rows assume a density exists. Similarly, we modified the appendix to make it clear.
>
> This should dissolve the confusion between the density and a measure.
>
> ## Dot product
>
> We are adapting your notation and writing the action of $p\_t$ on a test function in the dot product notation: $$ \\langle p\_t,f\\rangle :=\\mathbb{E}\_{x\\sim p\_t}\[f(x)\]. $$ We note that if $p\_t$ is a probability measure without a density, then the above is a "symbolic" dot product (i.e. it is not a dot product defined on a Hilbert space) and is simply a definition. However, it always is a bounded linear functional motivating this notation.
>
> ## F-parameterization of generator
>
> Thank you for suggesting to rewrite the generator and the $F$-parameterization in the form $\\langle \\mathcal{K}f,F\\rangle$ for a linear operator $\\mathcal{K}$ independent of the parameters. We call this now a *linear parameterization* and added a section in A.6. that explains this. We further show that all propositions of our work also hold for these linear parameterizations.
>
> **Discrete difference operator:** We note that for discrete state spaces, it is standard parameterizes the jump kernel as a rate matrix $Q$ for which it holds $$ Q\_t(x;x)=-\\sum\\limits\_{y\\neq x}Q(y;x),\\quad Q(y;x)\\geq\\text{for all }y\\neq x $$ (see section A.5.4). In particular, columns sum to $0$. Therefore, in your suggested example, it would hold that $$\\sum\\limits\_{y\\in S}(f(y)-f(x))Q(y;x)= \\sum\\limits\_{y\\in S}f(y)Q(y;x)=f^TQ(\\cdot;x)$$ Therefore, the generator application can be expressed as a simple matrix multiplication. This is the notation we use in table 1.
>
> **We have adapted all changes in our updated revised manuscript that you can view on OpenReview**. Would you consider this notation as clear? **We are very happy to integrate further feedback.** Thank you again for your helpful response.

---

> > ### Comment · Reviewer_RJkg · 2024-11-25
> >
> > Thank you for addressing my concerns in your revised manuscript. I acknowledge the improvements made, particularly in enhancing the clarity of the paper's presentation, including the notations. I sincerely appreciate your efforts and am pleased to reflect these revisions in my updated evaluation.

---

### Official Review · Reviewer_MFMu · 2024-10-31

**Soundness:** 3
**Presentation:** 3
**Contribution:** 3
**Rating:** 8
**Confidence:** 2

**Summary:**

Generator matching is introduced as an ambitious unification of multiple generative modeling methods, both in continuous and discrete spaces. This unified design space encompasses jump processes and in general any Markov process. Experiments on protein modeling and generative image modeling.

**Strengths:**

1. This paper has the ambitious goal of presenting a unifying framework for many (currently popular) generative modeling paradigms. Even unifying continuous and discrete models, and jump processes. This can be very relevant for the community, and the theory seems adequate and well supported.

2. Experiments are performed on diverse applications: proteins and images

**Weaknesses:**

1. It is unclear to me, and difficult to gauge, in how far the GM framework has benefits over *simple, naive* combinations of different modeling paradigms. I.e. how will the GM framework help future researchers in practice to build better models, other than only providing a theoretical unificiation?

**Questions:**

1. What are the limitations of GM? Is the linearity of the operators a limiting factor in the modeling capacity?
2. Line 128. What exactly is meant with *"simple"*. If I understand correctly, it needs to be linear, why not use that word?
3. Figure 2. What is *ME* in the legend? This figure could benefit some extra attention. Larger text size, match the naming in the legends with the vertical namings of the rows.

## Minor remarks/typo's:

- The use of $dx$ is somewhat confusing to me. I guess it is not the product of $d$ and $x$, is it supposed to signify $\mathrm{d}x$?
- On line 124: (1) and (2) hyperlink to seemingly random equations in the appendix. The same on line 401.
- Line 429: is *the* probability kernel
- Line 447: the informal use of *just increase number of channels* can be distracting to readers. In general the Experiments section could benefit from some additional refining.
- Equation numbering is inconsistent. Absent for large parts of the appendix. Also in the main text numbering is sometimes absent (e.g. lines 223, 251)

---

> ### Author Response · Authors · 2024-11-22
>
> **Practical benefits of GM.**
>
>     It is unclear to me, and difficult to gauge, in how far the GM framework has benefits over simple, naive combinations of different modeling paradigms. I.e. how will the GM framework help future researchers in practice to build better models, other than only providing a theoretical unificiation?
>
> Indeed, one contribution of GM lies in a theoretical unification of many existing generative models and also a universal characterization of the design space of Markovian generative models. However, there is a whole **variety of practical benefits for future machine learning introduced via GM**. We are happy to provide several examples:
>
> 1. **Markov superpositions:** GM enables the construction of Markov superpositions of different classes of Markov processes generating the same probability path (section 7.1).  Without GM, there would be no basis for combining for example a jump model and a flow model at the same time and achieving the improvement that we report in our experiments would not be possible.
> 2. **Multimodal constructions:** GM enables the construction of multimodal models. Such a rigorous construction for arbitrary modalities does not exist to the best of our knowledge and naturally requires the GM abstraction. **To address your comment, we added an explicit "recipe" to build a multimodal model out of unimodal models in section C.6. in detail**. This will help future researchers to construct models for arbitrary combinations of modalities. **We also included new experiments for multimodal protein generation improving state-of-the-art and highlighting the practical utility of the GM framework.**
> 3. **Jump models:** In our work, we introduced a scalable way of training jump models on Euclidean spaces in this work. As we show, this model is a powerful model achieving performance that rivals first versions of GANs or diffusion models. We believe that with more engineering and optimization efforts of the jump kernel and sampling, this model could well become equally or more powerful than existing state-of-the-art models, while having potentially other benefits such as increased diversity of samples (see e.g. the protein generation results).
> 4. **Modality-agnostic generative models:** The GM framework also allows to build generative models agnostic of the modality. In section 5.1., we derive a general solution of a jump process on arbitrary state spaces (that we then end up using for proteins). Therefore, any modality could be modelled with that process.
> 5. **Loss functions:** The GM enables a whole diversity of loss functions (i.e. any Bregman divergence works). This gives future researchers a whole variety of loss functions to use. To illustrate the utility of this, we did additional experiments to show one can improve existing flow and diffusion models with a minimal change in the code. We explain this in the general response.
>
> **Questions:**
>
>     Is the linearity of the operators a limiting factor in the modeling capacity?
>
> If a generator of a Markov process exists, then it must necessarily be a linear operator \- in the same way as the "common" derivative of a function is always a linear operation if it exists. While generators are linear as operators on function spaces, the neural network we use is still highly non-linear. Therefore, the linearity of the generator is rather a feature that allows us to develop the GM framework.
>
>     What are the limitations of GM?
>
> GM considers Markovian generative models and ones that are in continuous time. For this reason, GANs or VAEs are not instantiations of GM. Further, sampling with a model from GM requires iterative simulation of a Markov process. This can require a number of steps for sampling (i.e. one might require distillation as done for diffusion models).
>
>     Line 128. What exactly is meant with "simple". If I understand correctly, it needs to be linear, why not use that word?
>
> We did not use the word "linear" because if one considers $p\_{t+h|t}$ as an operator (as explained in section 4 in the paragraph on "test functions"), it is always a linear operator \- even for "large" h. We choose the word "simple" to motivate the decomposition in equation (1) that is then rigorously defined later in the section.
>
>     Figure 2. What is ME in the legend? This figure could benefit some extra attention. Larger text size, match the naming in the legends with the vertical namings of the rows.
>
> Thank you for pointing this out. "ME" was a typo and should stand for "MS" (Markov superposition). We also increased the font size.
>
> **Minor comments.** Thank you for the suggested edits. We integrated them into the new version.
>
> Together with our general response, we hope that the above addresses all of your comments. We welcome any additional discussions. Thank you again for your helpful feedback and response.

---

> > ### Comment · Reviewer_MFMu · 2024-11-26
> >
> > I thank the authors for the very clear answers to my questions, and the edits to the paper.

---

### Official Review · Reviewer_rGQU · 2024-11-04

**Soundness:** 3
**Presentation:** 4
**Contribution:** 3
**Rating:** 8
**Confidence:** 4

**Summary:**

This paper introduces the generator-matching (GM) framework for modeling arbitrary Markov processes in both continuous (i.e., $\mathbb{R}^d$) and discrete (i.e., $\\{1,...,N\\}^D$) domains. GM generalizes the (continuous and discrete) flow-matching (FM) approach, extending its concept of transforming the complex task of learning *marginal* distributions/scores/flows/generators to a simpler task of specifying *data-conditional* distributions/scores/flows/generators, and approximating the marginal ones through a loss based on conditional expectations. The framework not only unifies a variety of existing methods across continuous and discrete generative modeling, but also holds promise for multimodal applications, where the state space comprises the product of multiple different state spaces.

**Strengths:**

As discussed above, The GM framework extends the FM approach, building on the well-established relationships between marginal distributions/scores/flows and their data-conditional counterparts. From my persepective, this framework presents several notable innovations:

1. The introduction of jump, a type of stochastic process in the continuous domain with non-continuous trajectories, which largely unexplored within the generative AI community. Particularly compelling is the result showing that the widely-used conditional path $p_ {t|1}(x|z)={\cal N}(x|tz,(1-t)^2)$ can also be achieved through jumps, highlighting its potential utility for generative tasks.

2. An elegant theorem establishing that the generators of interest in $\mathbb{R}^d$ for applications can be decomposed to three basic types: flow (ODE), diffusion (scaled Brownian motion), and jump. This decomposition underscores the advantages of using the generator perspective in modeling.

The paper also introduces methodologies for: (a) combining models that use different generators, and (b) modeling multimodal data distributions. While not entirely novel, these contributions effectively systematize existing techniques through the GM perspective. Prior works have addressed these topics, for example: the equivalence of backward SDE and backward probability-flow ODE, the detailed balance condition for CTMC (Campbell et al., 2024), and the predictor-corrector for CTMC (Gat et al., 2024) are all examples of model combination; similarly, multimodal flows introduced in Campbell et al. (2024) and earlier models such as FrameFlow and FrameDiff have laid groundwork in multimodal data modeling, potentially inspiring aspects of this work.

Finally, the paper is well-written, featuring clear notations, a strong motivation, and a rigorous mathematical foundation.

**Weaknesses:**

1. The writing style in some part of the paper, especially section 6, feels overly general and abstract. While the GM framework is indeed comprehensive, it primarily applies to only 4 specific examples: flow, diffusion, jump in $\mathbb{R}^d$, and CTMC in finite state spaces. Although the theoretical concepts like target-affine (TA) loss and Bregman divergences are elegant, they lack practical utility in these contexts. This abstraction, without sufficient illustrative examples, limits the reader's ability to grasp the practical impact of the framework. I suggest exploring these theoretical constructs further to enrich the paper. For instance, theoretically or numerically comparing the performance of the TA-loss and discussing generator choice in more general state spaces would provide valuable insights. Additionally, a discussion on how TA-loss may facilitate the development of better training losses for generative tasks would be appreciated.

2. The literature review for jump-based models is insufficient. Jump is related to piecewise deterministic process (flow + jump) and piecewise diffusion process (flow + diffusion + jump) in the stochastic process literature (see, e.g., Del Moral & Penev, 2014), and there have been several works leveraging such process for generative modeling, e.g., Campbell et al. (2023) and Bertazzi et al. (2024), which the author(s) have not mentioned. Including these citations would better position GM within the broader literature and highlight its contributions.

3. The experiments are not convincing enough and lack the design details necessary for fair comparisons.

- For $\mathbb{R}^d$, the paper introduces two constructions of the conditional probability path (mixtures and geometric averages), each with realizations based on flow, diffusion, jump, or a mixture of these (Markov superposition). The toy example in figure 2 is worth further explorations: (1) Which conditional probability path is recommended for better empirical performance? Comparing the performance on more complex target distributions, such as $6\times6$ checkerboard patterns (Lou et al., 2023), would be informative. (2) Although these processes all generate the same marginal probability distribution in theory, practical implementations face approximation and discretization errors. Which realization provides the best sampling quality? (3) One potential advantage of jump over flow and diffusion is the ability of exploring complicated landscape of target distribution due to its non-continuous nature, but this benefit isn’t evident in the experiments. Therefore, it remains unclear whether the new algorithms offer practical advantages over existing methods, such as FM.

- The details of image generation experiment are lacking, such as the model size, neural network structure, and sampling schedule. The baselines (DDPM, VPSDE, and EDM) are outdated, and the datasets (CIFAR10 and ImageNet32) are less challenging than those commonly used today, such as ImageNet256.

- Despite emphasizing the GM framework’s potential for multimodal data, there are no numerical results on joint multimodal generation. For example, instead of considering protein backbone generation or pure image generation (which are unimodal tasks where existing algorithms already achieved amazing performance), evaluating GM on protein structure-sequence co-generation or image-text joint generation (which has been mentioned in section 7.2) would better showcase its advantage in multimodal settings.

- For the protein experiments, critical experimental details are absent, such as sampling NFE, training time, and model size. The target task is restricted to protein with length less than $128$, a relatively unchallenging setting. Designability, as an important metric for evaluating protein backbone, is not presented. Additionally, more up-to-date work like FoldFlow (Bose et al., 2023) and FoldFlow-2 (Huguet et al., 2024) are missing from the benchmarks. The performance gain achieved by the proposed approach seems to be minimal at the cost of training a separate network with more computational resources. Thus, its efficacy for protein generation remains unclear, especially in comparison to existing methods. It would certainly be more compelling to apply GM to multimodal tasks in protein design by incorporating additional sequence information.

**Questions:**

1. In appendix D.1, line 1319 suggests that the solution $\lambda_ t(x)$ is not unique. The derivation selects the form that achieves equality on line 1319 (see line 1334). Are there alternative choices of $\lambda_ t(x)$ that also satisfy the constraint? If so, are all choices of $\lambda_ t(x)$ equivalent in some sense, or do they have different characteristics in practical uses (given that they all correspond to the same conditional path $p_ {t|1}(x|z)$)?

2. I don't fully understand the argument in appendix D.2, starting from line 1473. Specifically, can you have a detailed elaboratation on the reason why the prior has compact support poses a potential issue (i.e., lines 1473-1474)? What implications do the non-zero boundaries of $\tilde\sigma_ t^2(\cdot|z)$ have? Moreover, I hope that the author(s) could have a review of the reflected SDE (the process ${\rm d}L_ t$) so that the readers can better understand the intuition behind how this pure diffusion solution to mixture path works.

3. Minor comments:

- I personally don't quite appreciate the unconventional notation $p_ t\odot f$ and $p_ {t+h|t}\odot f$ in this paper. Typically, $p_ t\odot f$ would be denoted as $\mathbb{E}_ {p_ t}f$; in standard textbooks of Markov semigroup (e.g., Bakry et al., 2014), $p_ {t+h|t}\odot f$ in this paper would appear as $P_ {t,t+h}f$, where the semigroup $(P_ {t,s})_ {s\in[t,\infty)}$ is defined as $P_ {t,s}f=\mathbb{E}(f(X_ s)|X_ t=\cdot)$. However, I recognize these non-standard notations may be acceptable within the machine learning community.
- Line 274, *no* diffusion coefficient? Besides, the $x_ 1$ in equation (9) seems to be a typo.
- Line 667: $\mathbb{E}_ {x\sim\mu_ 1}[f(x)]=\mathbb{E}_ {x\sim\mu_ 2}[f(x)]$.
- Line 669: $(X_ t)_ {0\le t\le1}$.
- Line 725: $p_ {t+h|t}(A_ {t+h}|x)$.
- Line 755: the limit operation is missing.
- Line 863: should mention the definition of matrix inner product $A\cdot B=\operatorname{tr}(AB)$ for positive definite $A,B$.
- Lines 901 and 968-978: should be "with probability $...+o(h)$ when $h\to0$".
- Lines 917, 1001-1017, 1050, 1060, 1067, 1072: the parentheses are missing: should be $\int_ {y\ne x}(f(y)-f(x))Q_ t({\rm d}y;x)$.
- Lines 1112-1125: $B(F_ t^\theta(x))$ is missing.
- Line 1155: the first generator is $\bar{\cal L}_ t$, not ${\cal L}'_ t$.
- Appendix C.6, proof for conditional generator: I suggest deriving the equation by directly applying derivative by parts to line 1178, which simplifies the proof and reduces its length.
- Line 1264, should mention that $\lambda_ t(x)$, $Q_ t(y;x)$ and $J_ t(y;x)$ may depend on $z$, and here this dependence is omitted for conciseness. Similarly, in line 1426, should mention that $a_ t,b_ t$ may depend on $z$ but are independent of $x$.
- Appendix D.1, time-derivative $\frac{\partial}{\partial t}p_ t$: I suggest rewriting the proof by directly calculating $\partial_ t\log{\cal N}(x|tz,(1-t)^2)$, which would simplify the presentation.
- Appendix D.2, I suggest replacing all $p_ {\rm simple}$ with $p_ 0$.
- Line 1421: the definitions of $G_ 0(x)$ and $G_ {z,\sigma_ {\min}}(x)$ have typos.
- Line 1476: "Let's *consider* a ..."

I will be happy to raise the score if the author(s) could address my concern.

**References**

- Bakry et al. Analysis and Geometry of Markov Diffusion Operators. Springer 2014.
- Bertazzi et al. Piecewise deterministic generative models. Arxiv 2407.19448, 2024.
- Bose et al. SE(3)-Stochastic Flow Matching for Protein Backbone Generation. ArXiv 2310.02391, 2023.
- Campbell et al. Trans-Dimensional Generative Modeling via Jump Diffusion Models. NeurIPS 2023.
- Campbell et al. Generative Flows on Discrete State-Spaces: Enabling Multimodal Flows with Applications to Protein Co-Design. ICML 2024.
- Del Moral & Penev. Stochastic Processes: From Applications to Theory. Chapman & Hall 2014.
- Gat et al. Discrete Flow Matching. NeurIPS 2024.
- Huguet et al. Sequence-Augmented SE (3)-Flow Matching For Conditional Protein Backbone Generation. ArXiv 2405.20313, 2024.
- Lou et al. Scaling Riemannian Diffusion Models. NeurIPS 2023.

---

> ### Author Response · Authors · 2024-11-22
>
> We thank the reviewer for insightful comments including the extensive feedback on the supplementary material.
>
> **Generator abstraction.**
>
>     GM primarily applies to only 4 specific examples: flow, diffusion, jump in [Euclidean space], and CTMC in finite state spaces.
>
> We stress that GM applies much more broadly (table 1 just lists *examples*). For example, as shown in our experiments, we also include Markov processes on manifolds (see also section H.5 on related work). Taking pairwise multimodal combinations gives us $\~80$ different classes of Markov processes. Further, there are different linear parameterizations of a generator and we can take Markov superpositions of all mentioned models. This leads to an explosive number of modelling options and design choices for GM.
>
> **Loss abstraction.**
>
>     Although the theoretical concepts like target-affine (TA) loss and Bregman divergences are elegant, they lack practical utility [...].
>
> We have done additional experiments showing that certain Bregman divergences improve training stability. New theoretical contributions allow us to drop the definition of TA-losses altogether such that we can simplify the manuscript. See general response for details.
>
> **Multimodal data.**
>
>     It would certainly be more compelling to apply GM to multimodal tasks in protein design [...].
>
> We have added multimodal protein experiments (see general response and the updated version of the paper) showcasing that GM improves the state-of-the-art.
>
> **Performance gain vs computational cost of training network.**
>
>     The performance gain seems to be minimal at the cost of training a separate network [...].
>
> In all of our protein experiments, we did not need to re-train any model for this result but repurposed an existing model (also for the new multimodal experiments). We made this clearer now. We note that even for re-training, the computational cost is essentially the same (only the final layer of the network has more channels).
>
> **Details for protein experiments.** We have added the requested details and more in Appendix G.4.
>
>     The target task is restricted to protein with length less than 128, a relatively unchallenging setting.
>
> We have used a standard benchmark which goes up to proteins with length 300.
>
>     Designability, as an important metric for evaluating protein backbone, is not presented.
>
> We have included a comprehensive table with designability, diversity, and novelty in Appendix G.4. We don't discuss designability in the main paper because as noted in (Yim et al., 2024), "designability can be artificially high if the generative model samples the same *designable* protein repeatedly." Diversity measures the number of *unique* proteins that pass the designability criteria.
>
>     [...] More up-to-date work like FoldFlow (Bose et al., 2023) and FoldFlow-2 (Huguet et al., 2024) are missing from the benchmarks.
>
> We have included FoldFlow in our new results (see general response ). FoldFlow2 code is not open sourced at time of writing, so we are unable to compare to them. We note that these newly reported FoldFlow results are worse than reported in their manuscript. We have reached out to the authors to verify we have ran the code from their github correctly.
>
> **Literature review for jump models:** We included a very extended discussion of all related work in the supplementary material (section H) including a subsection on jump models in section H.7 discussing the suggested references.
>
> **Reflected SDEs.**
>
>     I don't fully understand the argument in appendix D.2, starting from line 1473.
>
> We added references and a brief explanation of reflected SDEs to make it more understandable. Adding the mentioned boundary conditions is necessary for a uniform distribution because of the following: To use the Fokker-Planck equation, it is necessary that the density is differentiable. Here, the density of a uniform distribution is not differentiable (see section A.3.). It might be helpful to look at the derivation in A.5.2 of the adjoint Fokker-Planck and see that the partial integration (equation (72)-(74)) has to be adapted.
>
>     Are there alternative choices of lambda_t that also satisfy the constraint?
>
> Indeed, there are alternative choices for the jump intensity $\\lambda\_t$. Our derivation shows that the $\\lambda\_t$ is the minimal jump intensity with a state-independent jump kernel $J\_t$ that fulfils the jump continuity equation. However, we can artificially increase the number of jumps. For example, choosing
>
> $$\\begin{align\*} \\lambda\_t(x)=\\frac{1}{(1-t)^3}\\left(z-\\frac{(t+1)z}{2}\\right)^2,\\quad J\_t(x;x')=p\_t(x|z) \\end{align\*}$$
>
> also gives a solution. We tested this option but the version with minimal jump intensity gave better results.
>
> **Questions/Comments**: Thank you for the suggested edits. We integrated them into the new version and we also simplified the notation adapting the notation requested by RJkg.
>
> Thank you again for your helpful feedback.

---

> > ### Comment · Reviewer_rGQU · 2024-11-24
> >
> > I would like to sincerely thank the authors for their thoughtful effort in addressing my concerns. I am pleased to see that significant modifications have been made to the paper, including replacing the abstract TA-losses with more concrete examples, implementing comparisons between different Bregman divergences, adding an extensive literature review in the appendix that highlights the universality of the GM framework, providing additional experimental details, and presenting comparisons of experimental results on modeling multimodal protein data.
> >
> > That said, some of my concerns and questions remain partially unresolved. For example, regarding Weakness 3 (1), I inquired about the impact of the choice of the conditional probability path and its realization on empirical performance, particularly in the presence of errors from approximation and discretization. This appears to be related to your response about the choice of $\lambda _ t(x)$, described in the rebuttal as "the minimal jump intensity with a state-independent jump kernel that fulfills the jump continuity equation." Intuitively, more jumps along the trajectory may exacerbate discretization errors, akin to the phenomenon observed in score-based generative models, where increasing noise levels in the backward SDE degrades generation quality when sampling hyperparameters are fixed.
> >
> > I believe that exploring the conditional probability path, its practical realization, and developing improved numerical schemes for sampling are valuable directions for future work. Additionally, the task of image-text joint generation mentioned in Section 7.2 also seems highly promising for future exploration.
> >
> > Thank you once again for your detailed modifications and responses. I will raise the rating as promised. While I deeply appreciate the novelty of your framework, I remain hesitant to assign a rating as high as 8 due to the unresolved issues mentioned above.

---

> > > ### Author Response · Authors · 2024-11-26
> > >
> > > We thank the reviewer for the helpful feedback. We address the remaining concerns with additional theoretical and experimental results.
> > >
> > > ## Novel results
> > >
> > > We added an extended section E (page 40-55) to address your question, where we introduce and study several new GM models for various probability paths. This results in a GM model for every combination of {mixture path, CondOT path} x {flow, diffusion, jump, Markov superposition} (see table 4). This provides a **systematic study of the design space of GM, in particular ablating the effect of the probability path and the Markov model**. As suggested by the reviewer, we implemented the model on a 2d checkerboard distribution answering the questions raised by the reviewer:
> > >
> > > 1. **Influence of probability path on performance:** In fig. 5 and fig. 6, we show that the flow model performs better on the CondOT path but the jump performs better on the mixture path. **This shows that different probability paths are better or worse depending on the Markov model used and that there is no single best probability path.** There is an intuitive explanation for our results: the CondOT path geometrically moves probability mass, while the mixture path only up- and down-weighs mass but does not move the support of the distribution. In turn, the flow model "transports" probability mass, while the jump model "teleports" probability mass due to its discontinuous evolution.
> > > 2. **Discretization error:** In fig. 7 and fig. 8, we show that the flow model is highly prone to discretization errors for the mixture probability path, while showing low discretization error for the CondOT path. For a jump model, the opposite is true. This can be explained by the high Lipschitz constant of the learned vector field necessary to learn the transport of mass via a flow in the mixture path. In the context of density sampling, this is a known problem (see e.g. \[1\] Máté, and Fleuret (2023)). In contrast, for the jump model, larger movements of probability mass ("teleportation") are easier to handle. This explains the difference in sensitivity in discretization observed in our experiments.
> > >
> > > **Therefore, there is not one best probability path but rather an optimal combination of many design choices in the GM that will lead to good performance.**
> > >
> > > ## Experiments
> > >
> > > We highlight that we have heavily explored the design space of probability paths, models and Markov models in our existing experiments:
> > >
> > > - **Jump model:** We propose and extensively benchmark a whole range of novel KFE solutions to the CondOT probability path, e.g. finding the one with minimal jump intensity to work best. As noted by the reviewer, this is in line with previous results for diffusion models.
> > > - **Markov superpositions:** Every Markov superposition gives a solution to the KFE and therefore allows us to explore the design space of KFE solutions for a given probability path. We show that the optimal weighting in a Markov superposition improves the SOTA on image generation.
> > > - **Manifold experiments:** In our experiments on manifolds, we use a probability path on SO(3) constructed via a von-Mises distribution. This probability path has not been considered beforehand and leads to SOTA results on multimodal protein generation.
> > >
> > > ## Other Concerns
> > >
> > >        Developing improved numerical schemes for sampling are valuable directions for future work
> > >
> > > We highlight that we have developed better solvers for jump models in our existing work (we might have not highlighted this enough in the previous version). In section F, we describe a solver for a specific jump model developed in our work (equation 359-361). This sampling scheme improves the performance massively (without this solver, FID \> 10 on CIFAR10 for the jump model). We note that developing optimal numerical schemes for general sampling of a Markov process is a fundamental unsolved question of stochastic process theory that is outside the scope of our submission.
> > >
> > >          The task of image-text joint generation [...] seems highly promising for future exploration.
> > >
> > > We have presented additional multimodal protein generation results in the rebuttals that **improve the state-of-the-art in multimodal protein generation**, a convincing illustration of the power of GM for multimodal models. Joint image-text generation is usually addressed by separate papers with modality-specific innovations, e.g. see [3] \- one of the highest-ranking papers at ICLR 2025 that focuses solely on multimodal text-image generation. We agree that joint text-image generation with GM as an interesting direction for future work but consider it clearly outside the scope of the current submission.
> > >
> > > [1] Máté, and Fleuret (2023), [https://arxiv.org/abs/2301.07388](https://arxiv.org/abs/2301.07388)
> > > [2] Anonymous authors. [https://openreview.net/forum?id=SI2hI0frk6](https://openreview.net/forum?id=SI2hI0frk6)
> > >
> > > Thank you again for the extensive feedback on our work and for raising such interesting questions.

---

> > > > ### Comment · Reviewer_rGQU · 2024-11-28
> > > >
> > > > I would like to sincerely thank the authors for the additional theoretical and experimental results, and I believe that they have successfully resolved my concerns. I am happy to raise my rating to 8.

---

### Official Review · Reviewer_oGRr · 2024-11-04

**Soundness:** 4
**Presentation:** 3
**Contribution:** 4
**Rating:** 8
**Confidence:** 3

**Summary:**

The paper introduces a framework for creating generative models using arbitrary Feller processes. The framework is based on the concept of a generator that describes the infinitesimal change of distribution of a Feller process and on a possibility to learn that generator through minimizing a training objective. Existence of such a framework allows to design novel generative approaches and perform model combinations for building multimodal generative models.

**Strengths:**

**Significance and originality.** The main result of the paper lies in developing a generalization of the existing generative models into one theoretical framework. For example, the universal characterization of generators (provided by Theorem 1) appears for the first time in machine learning literature. Moreover, the result is stated for time-inhomogeneous Markov processes while the vast majority of mathematical literature focuses on studying properties of time-homogeneous ones. The paper also demonstrates immediate benefits of having such a theoretical framework by designing two new generative models based on jump and diffusion processes.

**Clarity and quality.** Overall, the paper is well-executed and well-written. It provides necessary background into Feller processes. The authors usually provide a motivation and an intuition behind introduced mathematical concepts. The paper gives guidance principles to build new generative models in a concise manner. The proposed generator matching framework is both mathematically sound and computationally feasible. The authors provided the empirical results on protein and image generation to demonstrate validity of the approach.

**Weaknesses:**

The main weakness of the paper is that the authors ignore previous results that have been done for jump processes [1, 2]. For example, [1] introduces a theoretical framework for constructing Diffusion Models in discrete-state spaces for an arbitrary Markov process, which can be either discrete time or continuous-time in nature. [1] uses generators and adjoint generators together with Chapman–Kolmogorov equations. Moreover, [1] constructs a generative model using the pure-death process, which, in its turn, is the example of jump processes. [2] focuses on the Levy process.  I believe the authors should provide a very detailed comparison with these papers in section 8 (Related Work).

I believe that the exposition of the framework can be improved. It still remains unclear from the paper why the generator $L_t$ can be used in forward and reverse processes. It remains unclear why using generators and adjoint KFE leads to “correct” generated samples.

[1] Santos, J. E., Fox, Z. R., Lubbers, N., & Lin, Y. T. (2023, July). Blackout diffusion: generative diffusion models in discrete-state spaces. In International Conference on Machine Learning (pp. 9034-9059). PMLR.
[2] Eunbi BI Yoon, Keehun Park, Sungwoong Kim, and Sungbin Lim. Score-based generative
models with Lévy processes. In Advances in Neural Information Processing Systems, volume 36 pages 40694–40707. Curran Associates, Inc., 2023.

**Questions:**

1. The authors skip a discussion of Anderson’s result [3], which was an important brick in developing generative diffusion models. Why is it not needed in the general framework? Why does focusing on matching of probability paths remove the need for Anderson's result for generators and adjoint generators?
2. Why does regularity assumption 2 (finite number of discontinuities) is needed? I see that the authors discard one of the terms in the proof of theorem 1. What does this term break in the framework if we do not discard it?
3. The authors state several times through the main part of the paper that generator matching works with arbitrary Markov processes while in reality they restrict the family of Markov processes to Feller processes. Do the authors expect that restricting to Feller processes may not be needed and the whole framework can be applied to any process with just Markovian property?
4. In proposition 1, the object “conditional generator” appears without any definition. I also believe that $X^z_t$ appears for the first time. Could the authors provide a definition of these objects and notation before using it in the proposition?
5. Could the authors provide a motivation behind introducing three regularity assumptions on Markov process under consideration?

*Minor:*

6. Page 3, line 120. “Starting with the right initial distribution $X_0 \sim p_0$.” What is the meaning (or definition) of “right” in the sentence?
7. Page 3. Line 096-099. The meaning of $\delta_z(dx)$ is neither defined before nor in the formula.
8. What does the authors mean under the words “an informal” in lines 129-130?
9. Page 17, lines 762-763. “... the following two regularity … ” -> “... the following three regularity…”

[3] Anderson, B. D. Reverse-time diffusion equation models. Stochastic Processes and their Applications, 12(3):313–326, 1982. ISSN 0304-4149.

---

> ### Author Response · Authors · 2024-11-22
>
> We thank the reviewer for insightful comments and for taking the time to review our work.
>
> **Time-reversals.**
>
>     Why does focusing on matching of probability paths remove the need for Anderson's result?
>
> Indeed, the idea of time-reversal is not needed in our framework. We added a detailed explanation in section A.4. that we summarize here. In diffusion models, one "noises" data via a "forward process". Every such "forward process" also defines a unique probability path but not vice versa. Our framework aims to train a model to approximate a solution to the KFE following a desired probability path $p\_t$. If we have defined $p\_t$ via a noising Markov process ("forward process"), then this is equivalent to finding a Markov process in reverse time that has the same marginals. This is a *weaker condition* than to find a true time-versal process in the sense used by Anderson (1982) \- the true time-reversal does not only concern marginals $X\_t\\sim p\_t$ but the joint distribution across time points (see A.4). However, for generative modeling, we often only use the final point $X\_1$ of the Markov process as a sample. Therefore, it only matters whether $X\_t$ has the desired marginals. For example, consider the *probability flow ODE* in diffusion models (Song et al, 2020\) which does *not* constitute a time-reversal of a diffusion process in the sense of Anderson (1982) but it fulfils the KFE in our sense. **This shows that restricting oneself to proper time-reversals imposes unnecessary mathematical complexity and can even lead to suboptimal solutions.** Still, a "true" time-reversal can still be used as *one* solution to the KFE for the probability path $p\_t$. See section A.4. for details.
>
> **Related work on jump models.** We have added an extended related work section including a subsection dedicated to jump processes (section H) - here discussing the 2 mentioned references:
>
> - **Blackout diffusion:** Thank you for pointing out this reference. We reference it now in the related work section and added an extended discussion in H.4. of how "blackout diffusion" is an interesting instantiation of GM on discrete spaces. We note that every continuous-time Markov process on discrete spaces is a jump model (see section A.5.4) and that we had discussed a variety of such models in the related work section of the submission. Hence, one can find a detailed discussion of blackout diffusion in section H.4.
> - **Levy processes:** Yoon et al (2023) construct SDEs driven by Lévy processes as a forward process. The fractional score serves as a linear parameterization of the generator (see appendix A.6) and the fractional denoising score matching loss (i.e. Theorem 4.3. in Yoon et al (2023)) is a special case of the CGM loss (proposition 2 in our work). While Levy processes in principle include jump processes, we note that the work by Yoon et al (2023) only considers learning the flow component of the Markov processes (the rest are hyperparameters), while here we show how to also learn the jump kernel or the diffusion coefficient. See section H.7. for details.
>
> **Generators and KFE.**
>
>     It remains unclear why using generators and adjoint KFE leads to “correct” generated samples.
>
> For all state spaces and Markov processes of interest, the KFE is both a sufficient and necessary criterion for a Markov process to generate a probability path. We made this clearer adding a comment on page 5 (below equation (6)) and adding an additional section A.2.
>
> **Regularity assumptions.** We made the regularity assumptions again clear in A.2. and also listed our motivation for choosing them.
>
>     Do the authors expect that [...] the whole framework can be applied to any process with just Markovian property?
>
> In order for a generator of a Markov process to be defined, property 2 in the definition of a Feller process must hold (see section A.1.2). Property 1 in the definition of a Feller process, might not be strictly necessary. However, we use it because (i) most of the mathematical literature (incl. the one we cite) uses it and (ii) any function on a computer is defined on a compact set. We stress this in the main paper now.
>
>     Why does regularity assumption 2 (finite number of discontinuities) is needed?
>
> While our framework would not break, simulating an infinite number of discontinuities would necessarily induce uncontrollable sampling error. We therefore came to the conclusion that such a model would therefore most likely be not interesting for the purposes of generative modeling.
>
> **Other questions**
> - Page 3, line 120: This refers to sampling the initial state of the Markov process.
> - $\\delta\_z$ refers to a Dirac delta measure.
> - Line 129-130: “Informal” refers to the fact that a Taylor approximation is only defined if a derivative of a function is defined \- this definition is then introduced formally later in the section.
>
> Thank you again for your helpful feedback and response. We welcome any additional discussions.

---

> > ### Author Response · Authors · 2024-11-27
> >
> > We appreciate the reviewer's valuable feedback and questions. Since our initial response to reviewer oGRr (see above), we have implemented several updates to enhance our work. To ensure the reviewer can assess all recent developments, we briefly summarize the most important *additional* updates below:
> >
> > **Systematic Study of Probability Paths and Markov Models:**
> > - We introduce and systematically analyze new GM models to evaluate the effects of various probability paths and Markov models.
> > - Specifically, we examine GM models for all combinations of {mixture path, CondOT path} x {flow, diffusion, jump, Markov superposition} (Table 4).
> > - Detailed descriptions can be found in section E (pages 40–55).
> >
> > **New Multimodal Experiments.**
> > - We present a new model and novel experimental results for GM applied to multimodal protein generation, **improving state-of-the-art results.**
> > - For details, see the general response and section 9 in the updated draft.
> >
> > **New loss functions:**
> > - We experimentally explore the space of loss functions for GM showing **improved training stability and final performance**. See section I in the updated draft.
> > - We universally characterize the space of loss functions via Bregman divergences, providing both a sufficient and necessary condition. See section 6 in the updated draft.
> >
> > We hope that this brief additional update helps the reviewer make a full assessment. Thank you again for the great feedback and for the opportunity to enhance our work. We welcome any additional discussions.

---

> ### Comment · Reviewer_oGRr · 2024-11-27
>
> Dear authors,
>
> Thank you for addressing my concerns in your revised manuscript and providing answers on my questions. I read revised version and the new version is easier to understand. Overall, the exposition of the framework has been improved. I agree with other reviewer that this paper will become a must-read for those interested in flow/sde generative models. I increased my score.

---

### Author Response · Authors · 2024-11-22

We would like to thank all reviewers for their constructive and positive feedback. We are pleased to see that our work is considered to potentially become a "must-read for those interested in \[generative modeling\]" (reviewer RJkg), and that it is "well-written, featuring clear notations, a strong motivation, and a rigorous mathematical foundation" (reviewer rGQU). We highlight that most comments have been addressed in the updated manuscript that you can find on OpenReview. Below, we’ve compiled rebuttal points asked by several reviewers while we address individual questions in the individual rebuttals.

**Multimodal data experiments.** Generator Matching (GM) allows one to design generative models for arbitrary modalities, including multimodal state spaces. To showcase this, we ran additional experiments on multimodal protein generation leveraging the jump model derived in our work. As one can see in the updated version of our work, **our multimodal jump-flow model improves the state-of-the-art result for multimodal protein generation**. We do the same as we did for the protein structure generation experiments and repurpose an existing model without re-training (given the time frame, it was not possible for us to re-train from scratch). Specifically, we switched to using MultiFlow instead of FrameFlow to accommodate multi-modal generation. Overall, we hope that these results highlight how versatile and practically useful the GM framework is.

**Loss functions.** Beyond giving rise to many new generative models and universally characterising them, Generator Matching also gives rise to new classes of loss functions. In fact, we universally characterise the space of loss functions. Several reviewers expressed interest in expanding this point. We made several changes to the manuscript to make this clearer. We did more experiments on this, too. Specifically:

- **Drop TA-loss:** We simplified the exposition of the paper by proving that any target-affine loss (TA-loss) is a Bregman divergence \- up to constants not affecting training gradients. We therefore dropped the concept of TA-loss altogether.
- **Necessary condition:** We have shown that the loss function has to necessarily be  a Bregman divergence in order for proposition 2 to hold, i.e. that the CGM loss has the same gradient as the GM loss. This shows that Bregman divergences are indeed the natural family of loss functions and this universally characterises the space of loss functions.  See Section C.4.2.
- **Additional image generation experiments showcasing the utility of new loss functions:** We ran additional image generation experiments with a flow model to illustrate the practical utility of Bregman divergences. We use different functions $\\phi:\\mathbb{R}\\to\\mathbb{R}$ for the Bregman divergences (within the limited time frame). Note that this corresponds to changing one line of code in a standard codebase for training a flow/diffusion model. However, this simple change of code led to an improvement of both the training dynamics and the overall performance. Specifically, we tested various functions $\\phi(x)$ and trained a flow model on CIFAR-10 with the respective Bregman divergences. We explain this in section I in the appendix. As one can see in figure 6, the **training dynamics of the models trained with the new Bregman divergences are significantly better (more stable training) and the final FID is also lower**. We could not explore this more given the limited time frame of the rebuttals and this is not main focus of our work. However, this illustrates however that the concept of a Bregman divergence as a loss function for GM introduced in our work has immediate practical benefit and opens up a large design space that can be leveraged in future work.

**Additions to the paper.** We highlight to the reviewers that we made several additions to the paper. Specifically,
- Section H: A (very) extended discussion of related work explaining in detail how previous works fall under the GM framework.
- Section A.4.: A rigorous definition of a linear parameterization of a generator (now also referenced in the main text)
- Section C.6: Making multimodal constructions more explicit using definition of linear parameterization.
- Section C.3.: Added a proof that marginal generator shape also holds for linear parameterization.
- Section D: Derivation of an ELBO loss for the jump model.

**Notational changes:** To address the concerns of reviewers rGQU and RJkg, we made several changes in the notation in the paper, e.g. writing the action $p_t\odot f$ as a dot product $\langle p_t, f \rangle$ and more clearly distinguishing between a measure $p_t$ and its density $\frac{dp_t}{d\nu}$ w.r.t. a reference measure $\nu$.

We would like to thank the reviewers again for their helpful comments and we welcome additional discussions.

---

### Public Comment · ~Adam_Gosztolai1 · 2025-02-06
**'Pure diffusion'**

I want to thank the authors for the interesting papers.

Upon trying to reproduce the results 'Example2-Purediffusionsolutiontomixturepath.' I encountered several difficulties. I believe this is partly caused by several inaccuracies in the section and by incomplete information to facilitate easy reproduction of the results.

1. As one reviewer pointed out below, the use of a uniform initial distribution is not motivated. This is an unusual assumption in papers on generative modelling, which typically either assume a Gaussian initial distribution or an arbitrarily complex initial distribution.

2. The probability path chosen is p_t(dx|z)= \kappa_t\delta_z + (1-\kappa_t)Unif[a1,a2]. However, in Appendix E.3, this changes to pt(x|z)=(1-\kappa_t) p_0(dx)+ \kappa_t N(z, \sigma_min). It would be nice to relate the two.

3. Starting with the Uniform distribution, the authors include 'an additional reflection term at the boundaries of the data support'. I assume the boundaries are at a1 and a2. Is this correct? It would be nice to define this in arbitrary dimensions, not just in R. Does this mean that only data distributions inside these boundaries can be generated? Does this example generalise the case where boundaries are not included?

4. In table 1, \sigma^2_t(x) is a d x d positive semidefinite matrix. Yet, in Eq. (14), \sigma^2_t(x|z) seems to have the same dimensions as the data, d x 1. How is this consistent? Do you consider only a diagonal matrix?

5. The authors say that this example has 'no drift (no vector field)'. However, this seems incorrect because diffusion generators can be expressed as a flow involving the score, \Delta p = \nabla\cdot(\nabla\log p).

6. What is the reflection process dL_t in (278)? This seems not defined anywhere.

7. On line 2110, the arguments of integration in the cumulative distributions are strange, in particular, there is no argument y.

---

> ### Public Comment · ~Peter_Holderrieth2 · 2025-02-06
>
> Thank you for your interest in this example.  All information you requested is in the paper and we happily point you to the places where the information can be found that you request:
>
> 1. **Use of uniform distribution**: This example illustrates the flexibility of the generator matching framework, also with respect to the prior distribution. We could also use a Gaussian distribution in principle (the derivations in section E.3) are general. However, the function $G_0(x)$ cannot analytically be computed for a Gaussian (there is no closed formula) but for a uniform it can be (see section E.3).
>
> 2. **Relation between two paths:** One is a special case of the other. So the relation is to plugin the uniform distribution for $p_0$.
>
> 3. **Arbitrary dimensions:** We state in line 261 that we do everything for $d=1$ as proposition 4 easily allows to extend it to arbitrary dimensions.
>
> 4. **Diagonal matrix:** This is a common convention in the notation of SDEs. The term $\sigma_t(x|z)$ can be considered a diagonal matrix.
>
> 5. **Diffusion generators:** In fact, in general SDE generators include a drift (not necessarily a score). However, as stated in the paper, we make the choice of setting the drift to be zero in this example (line 274, 275). Further, as stated in the description of table 1, we most of the time make diffusion processes have zero drift in the work because we can use a Markov ensemble of a flow and a diffusion to include arbitrary drifts (i.e. this is no way a restriction). However, you can also just assume that there is a drift term throughout our work. The theory checks out the same way. This was done for brevity.
>
> 6. **Reflection process:** As stated in line 2177, we point to the literature for a definition of reflection SDEs. Explaining this mathematical background is out of scope for our work.
>
> 7. **Cumulative distribution functions:** They are correct as written. We integrate twice (the arguments appear as limits in the integrand).
>
> Let us know if you have any further questions.

---

> > ### Public Comment · ~Adam_Gosztolai1 · 2025-02-06
> > **Response to Authors**
> >
> > Thank you for clarifying my confusion. I will give another go at reproducing the results with my increased understanding. I hope the code will be made openly available so the theory can be easily tested.

---

### Meta-Review · Area_Chair_uuTH · 2024-12-19

**Metareview:**

This paper introduces a generator-matching framework for modelling arbitrary Markov processes in both continuous and discrete domains. It generalizes the flow-matching approach, extending its concept of transforming the complex task of learning marginal distribution generators to a simpler task of specifying data-conditional distributions. The paper was reviewed by four reviewers who all recommend it to be accepted (scores 8+8+8+8).

**Additional Comments On Reviewer Discussion:**

During the rebuttal phase, this paper was subject to rather active discussion. The interaction between the authors and reviewers was helpful. The average score increased from 6.25 -> 8.0 during the discussion.

---

### Decision · Program_Chairs · 2025-01-22

Accept (Oral)